# Distinct contributions of prefrontal, parietal, and cingulate signals to exploratory decisions
Victor K. S. Chan [1] ✉, Nicole H. L. Wong [1], Tsz-Fung Woo[2,3], Kei Watanabe [4,5], Masahiko Haruno [4,5], Chun-Kit Law [1] & Bolton K. H. Chau [1,6,7] ✉

Exploratory decisions could be motivated by multiple reasons, such as to reduce uncertainty of existing options (i.e. internal exploration) or to identify new alternatives (i.e. external exploration). Exploratory behaviours are related to a wide range of brain regions including the intraparietal cortex (IPS), anterior cingulate cortex (ACC), and medial prefrontal cortex (mPFC). However, whether these regions play unique roles signalling different aspects of exploration remain unclear. Here, we identified unique internal and external exploratory signals in the IPS and ACC, respectively. Crucially, these signals were invariant to the impending decisions. In contrast, the mPFC flexibly encoded the value of internal exploration, external exploration or collecting a reward based on the type of impending decision. This is consistent with the neural common currency hypothesis suggesting the mPFC's role in signalling general decision value. Our findings contribute to the understanding of the unique roles of IPS, ACC and mPFC during exploratory decision-making.

Understanding the brain mechanisms underlying how people explore in an uncertain environment has been the focus of many cognitive scientists, neuroscientists and computer scientists[1–8]. Depending on the investigators, cognitive processes related to exploration has been described as search, foraging, prospection, and information sampling[3,9–12], and these processes are implicated in the medial prefrontal cortex (mPFC)[1,5,6], intraparietal sulcus (IPS)[3,13], and anterior cingulate cortex (ACC)[7,8]. However, there is little direct comparisons in a single study showing the dissociable roles of these three regions during exploratory decision making. Intriguingly, these three brain regions are not only related to exploration, but also the expected reward associated with accepting an existing option[14–17]. Hence, it is critical to test directly if mPFC, IPS and ACC play dissociable roles in signalling exploration.

Based on related studies, we suggest exploratory decisions can be classified into two types – internal and external exploration. Here, internal exploration refers to the exploration that aims to reduce uncertainty of the existing choices. In many real-life situations, the outcome of a choice can be uncertain and it is often necessary to explore by more information to reduce the uncertainty. For example, when a new mobile phone is launched to the

market, consumers often explore information from product descriptions and review articles before deciding to buy the phone or not. Alternatively, instead of exploring information of the same product, consumers often also explore in the environment for other alternative products. Here, we refer this as external exploration. Despite these two actions involve exploration processes, they differ fundamentally by whether or not information is obtained from existing options or from the external environment.

Previously, exploratory decision for reducing uncertainty is implicated in the IPS[3,18,19]. For example, Horan and colleagues performed electrophysiological recording in macaque IPS neurons and showed that these neurons were more active when there were greater reductions in choice uncertainty. Their data also suggested that the activity of these IPS neurons was insensitive to the amount of reward associated with the choice. These findings were intriguing because they provide an alternative account of the widely reported role of IPS in signalling reward magnitude[16,20–23].

Similarly, the ACC has been the focus of exploratory decisions[7,8]. One reason why an individual would search for new options in the environment (i.e., external exploration) is because the options in the general environment are more rewarding than the existing options

[1]Department of Rehabilitation Sciences, The Hong Kong Polytechnic University, Hong Kong, China. [2]Danish Research Institute of Translational Neuroscience - DANDRITE, Nordic-EMBL Partnership for Molecular Medicine, Aarhus, Denmark. [3]Department of Molecular Biology and Genetics, Aarhus University, Aarhus, Denmark. [4]Graduate School of Frontier Biosciences, University of Osaka, Osaka, Japan. [5]Center for Information and Neural Networks, National Institute of Information and Communications Technology (NICT), Osaka, Japan. [6]University Research Facility in Behavioral and Systems Neuroscience, The Hong Kong Polytechnic University, Hong Kong, China. [7]Mental Health Research Centre, The Hong Kong Polytechnic University, Hong Kong, China. ✉e-mail: troncks1603@yahoo.com.hk; boltonchau@gmail.com

**Fig. 1 | Decision-making task ($n = 24$). A** Task structure. Each option was composed of four dials, the coloured area of each dial indicates the possible number of points that could be obtained. Only one dial, identity unknown to participants, was related to the actual points earned by each option, such that options containing more variable dials had greater uncertainties. In the decision phase, participants should decide between three possible responses. *Accept*, to end the trial by selecting an option and earning its actual points. *Internal exploration*, to randomly remove one dial of a selected option to reduce its uncertainty. *External exploration*, to reveal a new option from one of the black boxes. Each decision phase terminated at the time of participants' decision. Each internal or external exploration costed one point and the cumulated cost incurred by the end of a trial was displayed in real time at the top left-hand corner. The top right corner indicated the average point of all (hidden and revealed) options of the same trial, which was important for guiding external exploration decisions. **B** Task performances. The proportion of *accept*, *internal exploration* and *external exploration* made by participants throughout the task (top panel). The proportion of trials with different total number of internal exploration and external exploration (bottom panel). Distribution of the number of exploration decisions per trial (0 = no exploration, 1 = one exploratory decision, etc.).

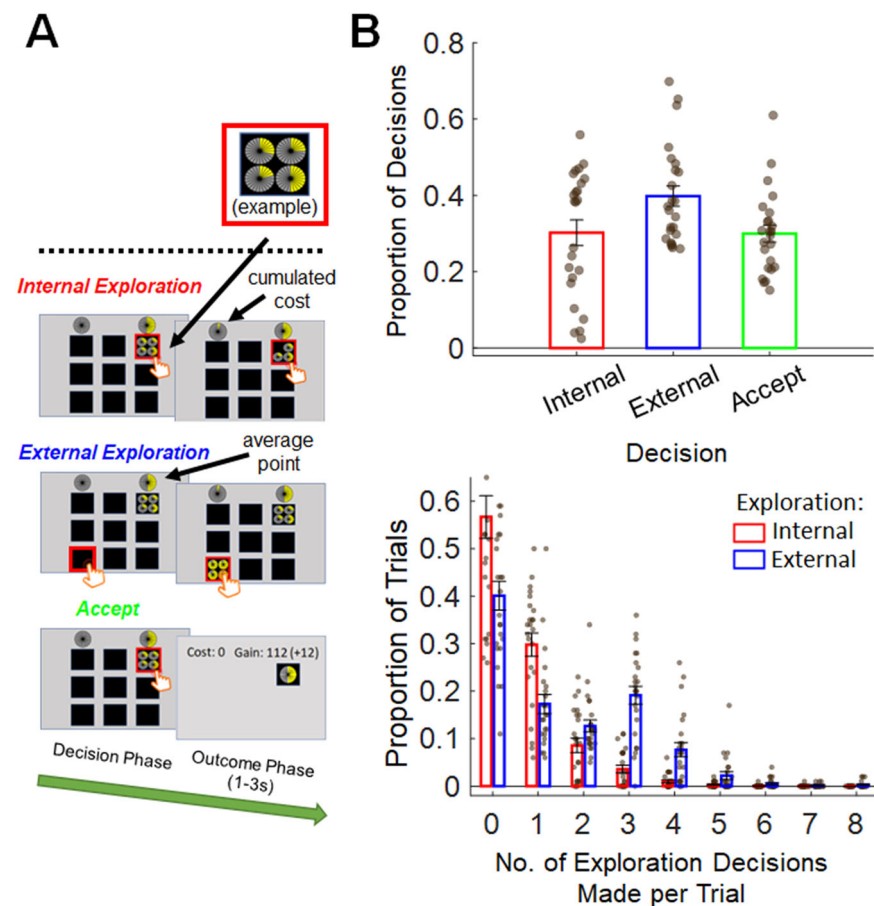

available. Neuroimaging data in human[7,8] and electrophysiology data in macaque[24,25] show that the ACC activity varies as a function of more rewarding environments and less rewarding options available in the choice set. In addition, when an individual experiences a volatile environment, the ACC also forecasts reward changes in the environment. Such information in the ACC is important for guiding whether or not it is advantageous to explore the environment for new options.

Unlike the IPS and ACC, the mPFC, especially the ventral part of the mPFC (vmPFC), has considered to have little role in exploration. Most studies have focused on its role in representing options that are available[15,17,26–28]. However, Trudel and colleagues scrutinized the vmPFC's activity more carefully across different stages of exploration[5]. They showed that the vmPFC's activity correlated positively with the uncertainty level of a choice when an individual was in an exploration state. Intriguingly, when the individual shifted to an exploitation state, the polarity of such vmPFC signal also changed from a positive to a negative correlation. Choices with greater uncertainty levels should also be more valuable during exploration, but less valuable during exploitation. Hence, one way of understanding these results is that the mPFC signals the general value of a decision, which is advocated in the neural common currency hypothesis[27]. Based on these, we should expect that when people are allowed to take actions between internal exploration, external exploration, or to accept an available option, the mPFC should flexibly encode the decision parameters that are relevant to the action.

In the current study, we contrasted the neural mechanisms underlying internal exploration, external exploration, and choice acceptance (i.e., to halt exploration and accept an option). This was achieved by testing human participants on a multiple-option decision making task that incorporates all three types of decisions (i.e., internal exploration, external exploration, acceptance) whilst undergoing functional magnetic resonance imaging

(fMRI). In a whole-brain analysis, we identified unique sets of neural correlates related to accept, internal exploration, and external exploration, such as the IPS, ACC, and mPFC, respectively. Further fine-grained analyses by focusing on these regions showed that the IPS's internal exploration signal and ACC's external exploration signal were invariant to whether an individual was exploring or not. In contrast, the mPFC signal varied as a function of the type of decision being made. It encoded the value of accept when the individual was about to accept an option, but also shifted to encode the value of internal exploration or the value of external exploration when the individual was about to take these actions. Overall, our results showed that the mPFC is better described as encoding the general decision value, whereas the IPS encodes the internal exploration value and the ACC encodes the external exploration value.

## Results

In the multiple-option decision-making task (Fig. 1A), participants were first offered an option that could lead to one of four possible rewards displayed. Participants had to choose between accepting the option, an internal exploration decision, or an external exploration decision. External exploration was done by revealing a new option from one of the remaining locations. Internal exploration was done by removing one possible reward of an option, such that the option's actual outcome would potentially become more certain. Each trial lasted until the participants made an accept decision (see Methods). During the task, participants made 30.19% internal exploration, 39.82% external exploration, and 29.99% accept decisions (Fig. 1B, top panel). When the number of internal exploration decisions was counted on each trial, 56.68% trials involved no internal exploration decisions and there was a general trend that there were smaller proportion of trials with greater number of internal exploration decisions (Fig. 1B, middle panel). A similar analysis on external exploration decisions showed that

**Fig. 2 | Behavioural results ($n = 24$). A** A multinomial logistic regression showing the effects of internal exploration, external exploration, and accept values on the ratios of internal exploration / accept decision (top panel) and external exploration / accept decision (bottom panel). **B** Illustrations of proportions of decision types varied as a function of internal exploration, external exploration, and accept values. A participant-specific analysis was conducted by ranking internal exploration, external exploration, and accept values into 30 probability classes, which were then consolidated into 20 levels using an 11-rank sliding window approach. This transformation increases the number of trials per level, including at extreme values, so the estimates are more stable and less noisy. The resulting smooth curves show the decision threshold, the point at which exploration drops off and acceptance becomes more likely. The choice ratios between decision pairs were analyzed across these levels, excluding trials involving the third decision type. The final results represented the averaged behavioral patterns across all participants. * denotes $p < 0.05$; ** denotes $p < 0.01$; *** denotes $p \leq 0.001$. Error bars represent ± standard error of the mean (SEM).

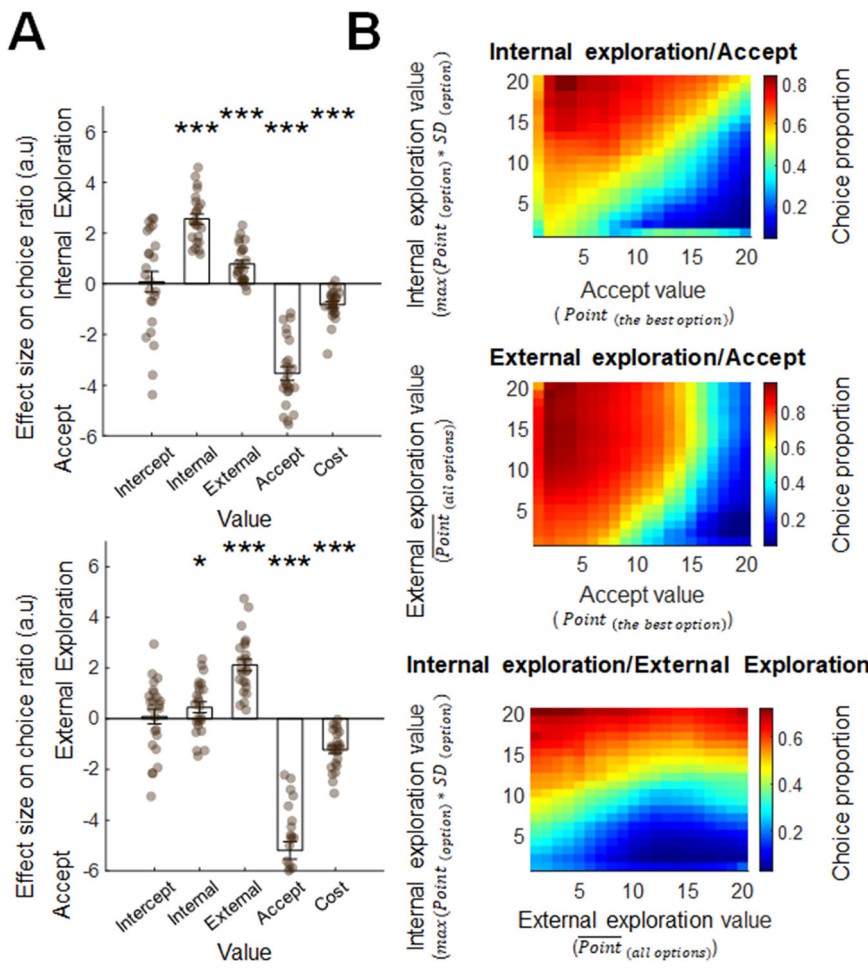

40.08% trials involved no external exploration and there was a general trend that there were smaller proportion of trials with greater number of external exploration (Fig. 1B, lower panel).

## Modelling the utility functions for describing exploration decisions

Prior to the analysis of the fMRI data, it is essential to identify value functions that best describe participants' internal exploration, external exploration, and accept decisions. We proposed 120 variants of a general linear model using different combinations of value functions and evaluated the goodness-of-fit of each variant and the predictive power using the Bayesian information criterion and prediction accuracy, respectively (BIC, Supplementary Fig. S1)[29]. The robustness of our model selection was further validated through Bayesian model selection analysis, which showed that our best-fit model had the highest exceedance probability among all candidate models (Supplementary Fig. S2). Additionally, at the individual level, this model best described the greatest proportion of participants' behaviors (20.83% of participants; Supplementary Fig. S3). To ensure the reliability of our model fitting procedure, we conducted parameter recovery analyses, which demonstrated that all ten parameters could be successfully recovered with high precision (all correlations: rs > 0.801, ps < 0.001; Supplementary Fig. S4).

The optimal model identified involved the following value functions.

$$Value_{Internal\ exploration} = \max(Point_{(option)} * StandardDeviation_{(option)})$$

$$Value_{External\ exploration} = \overline{Point}_{(all\ options)}$$

$$Value_{Accept} = Point_{(the\ best\ option)}$$

We also considered whether the selection of a specific model could drastically affect the results of our subsequent analyses. This was done by applying either the second-best fit model or individual participants' best fit model (instead of the group's best fit model, see Supplementary Fig. S5) and re-run behavioural and fMRI analyses in Supplementary Fig. S6 and S7, respectively. Our results showed that these alternative models generate comparable results.

Next, to illustrate that participants used internal exploration, external exploration and accept values to guide their choices, we performed a multinomial logistic regression. This approach estimated the effects of these value terms on the odds-ratio of internal exploration versus accept decisions and their effects on external exploration versus accept decisions in a single analysis. The results suggested that the choice ratio between internal exploration and accept decisions was positively associated with the internal exploration value (Fig. 2A, top panel; $\beta = 2.558$, $t_{23} = 12.920$, $p < 0.001$), negatively associated with the accept value ($\beta = -3.540$, $t_{23} = -13.115$, $p < 0.001$), and negatively associated with the cumulated cost ($\beta = -0.819$, $t_{23} = -6.597$, $p < 0.001$). The same analysis also showed that the choice ratio between external exploration and accept decisions was positively associated with the external exploration value (Fig. 2A, bottom panel; $\beta = 2.125$, $t_{23} = 9.092$, $p < 0.001$), negatively associated with the accept value ($\beta = -5.193$, $t_{23} = -15.150$, $p < 0.001$), and negatively associated with the cumulated cost ($\beta = -1.222$, $t_{23} = -7.832$, $p < 0.001$). These results confirmed that participants were more likely to internally explore, externally explore or accept when their corresponding values were greater.

We noticed, surprisingly, that the internal exploration/accept choice ratio was positively associated with the seemingly irrelevant external exploration value (Fig. 2A, top panel; $\beta = 0.784$, $t_{23} = 5.282$, $p < 0.001$), and the external exploration/accept choice ratio was positively associated with

**Fig. 3 | A whole brain analysis identified dissociable brain regions encoding the decision values of internal exploration, external exploration, and accept (n = 24).** A whole-brain analysis, three distinct sets of brain regions activated as a function of the values of internal exploration (left), external exploration (middle) and accept (right) were identified. This was achieved by contrasting the effect of one type of value (e.g., internal exploration value) with the average effect of the remaining two types of value (e.g., the average effect of external exploration and accept values). Cluster-based threshold z > 3.10, p < 0.05. See more details in the Methods section, "fMRI data analysis: whole-brain analysis".

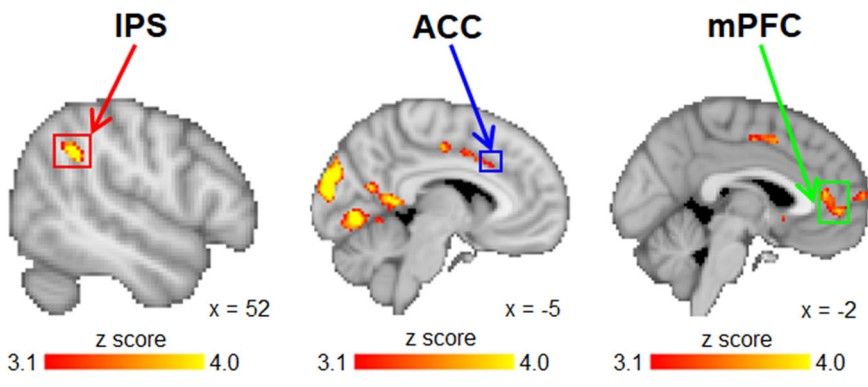

the seemingly irrelevant internal exploration value (Fig. 2A, bottom panel; $\beta = 0.450$, $t_{23} = 2.088$, $p = 0.0481$). These suggested that participants were also generally more inclined to explore when they were dissatisfied with the current offers (Fig. 2B, middle panel). Next, we investigated the neural mechanisms of exploration via a series of whole-brain analyses, and time-course analyses with specific regions-of-interest (ROIs).

A whole-brain analysis was performed to identify neural signals that were related to the values of accept, internal exploration and external exploration. First, we identified the IPS encoded an internal exploration signal to track the option with a great overall value that also came with a large level of uncertainty (Fig. 3, left panel; cluster-based threshold z > 3.10, p < 0.05), which is similar to previous reports suggesting the IPS activity reflects the level of information gain that follows a decision[3,19]. Second, we identified the sulcus of the ACC encoded an external exploration signal to track the overall value in the environment (Fig. 3, middle panel; cluster-based threshold z > 3.10, p < 0.05), which is similar to previous reports suggesting the ACC sulcus reflects the recent reward history or the environment's overall reward value[7,8,30]. We notice that the cluster had an elongated shape that lies along the cingulate sulcus. In subsequent analyses, we focused on the anterior part of the cluster, which is most related to reward processing in the cingulate cortex (Cluster 3 of the cingulate parcellation map by Beckmann et al.[31]. Third, we identified that the mPFC encoded an accept signal to track the value of the best option that was presented in the choice set (Fig. 3, right panel; cluster-based threshold z > 3.10, p < 0.05)[14,27,28,32–35]. So far, we found that different sets of brain regions (mainly the IPS, ACC and mPFC) could seemingly be matched to the three types of exploration values. These findings are summarized in Supplementary Table S2. Next, we focused on IPS, ACC and mPFC as ROIs (Fig. 4A) to scrutinize their signals more closely using time-course analyses.

**Exploration regions encoded multiple exploration parameters**

Performing a time-course analysis to examine the ROIs' signals has several advantages. First, it illustrates the dynamics of the signals across time without the need of assuming the shape of their waveforms (as in typical whole-brain analyses). Second, because it is free from any waveform assumptions, it allows identification of signals that deviate from a canonical haemodynamic response function. Third, it verifies that the whole-brain contrasts in Fig. 3 reflect positive value signals in the target regions rather than relative deactivation in the compared conditions (see Fig. 3 for contrast definitions and Fig. 4B for independent verification).

We began our analysis by focusing on the IPS and examined how its signal varied as a function of exploration values, as in parameters used in the whole-brain analysis. We found, again, that the IPS showed a positive internal exploration signal that peaked at 6.42 s after the onset of the decision phase (Fig. 4B, left panel; $t_{23} = 3.275$, $p = 0.003$), suggesting that the IPS became more active as the internal exploration value was greater (i.e., in the presence of options that were both rewarding and uncertain). In contrast, there was an absence of an external exploration signal

(Fig. 4B, left panel; $t_{23} = 1.858$, $p = 0.076$) and an absence of an accept signal (Fig. 4B, left panel; $t_{23} = -1.463$, $p = 0.157$). Next, we further validated whether the observed brain activity was functionally relevant to the decisions made. We tested the relationship between individual variabilities in the proportion of the internal exploration, external exploration and accept signals and the proportion of their corresponding choice in a between-participant analysis. In other words, we asked whether a stronger neural signal of a certain kind (e.g., stronger internal exploration signal) was associated with more frequent choices of the same kind (e.g., more frequent internal exploration decisions). We extracted the peak of each signal in Fig. 4B for each participant and correlated with each participant's proportions of internal exploration, external exploration, and accept decision made throughout the task. The results showed that those with stronger internal exploration signals also had greater proportions of internal exploration decision (Fig. 4C, left panel; $r = 0.438$, $p = 0.042$). In contrast, the strength of IPS's external exploration and accept signals was unrelated to the proportion of external exploration (Fig. 4C, left panel; $r = 0.027$, $p = 0.904$) and accept (Fig. 4C, left panel; $r = -0.038$, $p = 0.866$) decisions, respectively.

The ACC showed a different pattern of results when a similar set of time course analyses were performed. There was a significant external exploration signal that peaked at multiple time points (Fig. 4B, middle panel; peaked at -2.52 s: $t_{23} = 2.114$, $p = 0.046$, peaked at 0.71 s: $t_{23} = 2.790$, $p = 0.01$, peaked at 8.26 s: $t_{23} = 2.525$, $p = 0.019$, peaked at 12.36 s: $t_{23} = 2.074$, $p = 0.049$). There was a negative internal exploration signal peaked at 3.903 s (Fig. 4B, middle panel; $t_{23} = -2.628$, $p = 0.015$) and an absence of accept signal (Fig. 4B, middle panel; $t_{23} = -1.291$, $p = 0.209$). Further between-participant analysis suggested that participants showing stronger ACC external exploration signals also tended to make more external exploration decisions (Fig. 4C, middle panel; $r = 0.469$, $p = 0.028$). However, for internal exploration and accept signals, there was no relationship between the corresponding ACC 'signal' strength and decision proportion (Fig. 4C, middle panel; internal exploration: $r = -0.376$, $p = 0.084$; accept: $r = 0.148$, $p = 0.509$).

Finally, the mPFC time course analyses showed yet another signal pattern compared to that of IPS and ACC. The mPFC showed a positive accept signal (Fig. 4B, right panel; peak at 5.35 s: $t_{23} = 3.337$, $p = 0.003$, peak at 9.70 s: $t_{23} = 3.379$, $p = 0.003$). The same region also showed a negative internal exploration signal that peaked at 4.69 s (Fig. 4B, right panel; $t_{23} = -4.537$, $p < 0.001$), in which the mPFC became less active when the choice had a more uncertain outcome. This is consistent with previous reports suggesting the additional role of mPFC signalling choice uncertainty or confidence[5,36–39]. Surprisingly, there was also an external exploration signal that peaked at 12.69 s (Fig. 4B, right panel; $t_{23} = 3.322$, $p = 0.004$). Finally, a between-participant analysis showed that those with stronger accept signal had a greater proportion of accept decisions (Fig. 4C, right panel; $r = 0.618$, $p = 0.022$), those with stronger internal exploration signals also had greater proportions of internal exploration decisions (Fig. 4C, right panel; $r = 0.431$, $p = 0.045$). However, the

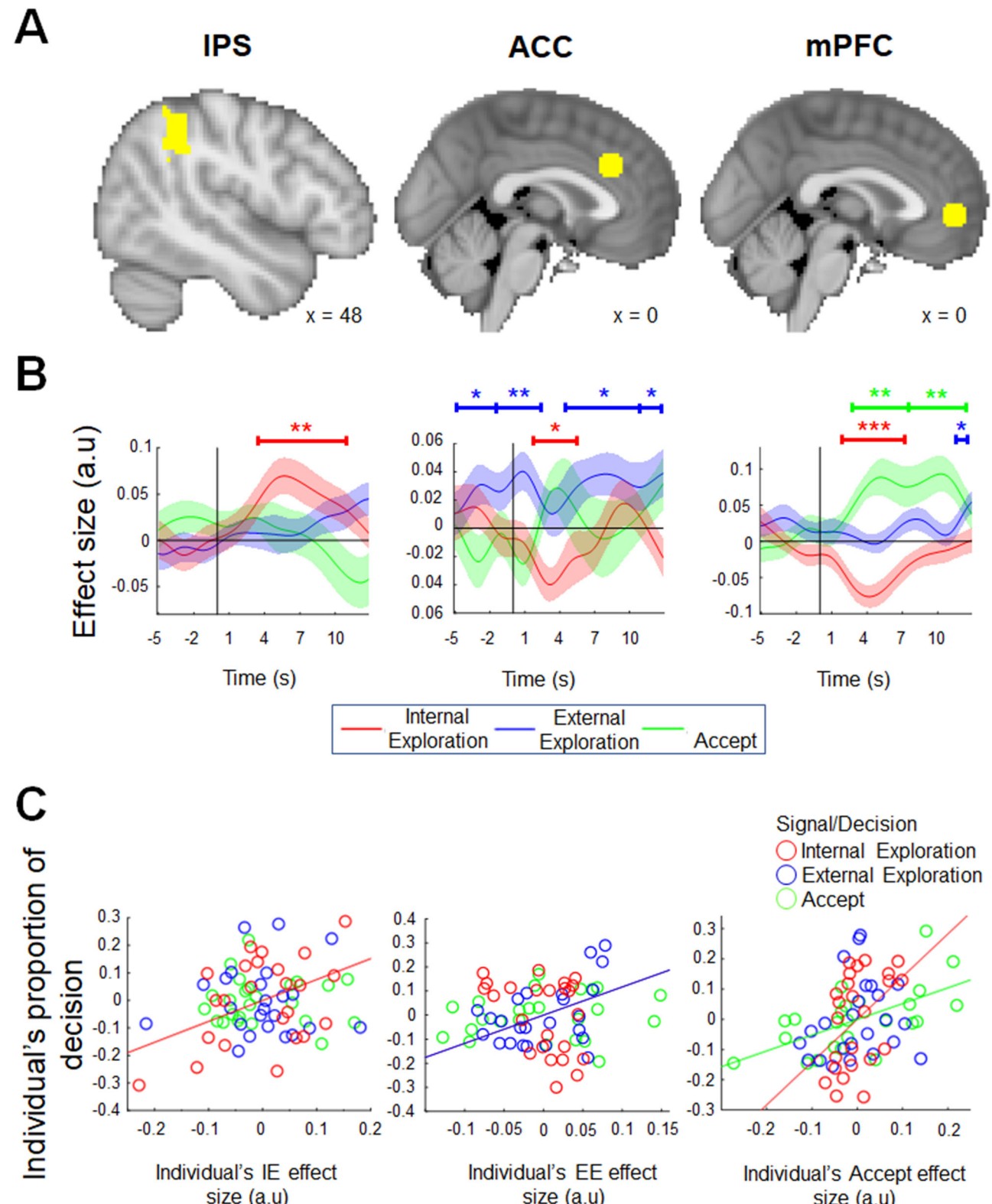

strength of the internal exploration and external exploration signals were unrelated to external exploration (Fig. 4C, right panel; $r = 0.199$, $p = 0.375$) decisions. These results showed that the mPFC was mainly involved in signalling accept decisions, however, additional findings (Fig. 4B, blue line and Fig. 4C, red line) suggest that mPFC may also be involved in signalling exploration to some extent. Next, we performed further analysis to investigate this more closely.

**Further analysis confirmed that IPS and ACC signalled internal exploration and external exploration, respectively**

So far, our results seem to suggest an appealing one-to-one matching between brain regions and value signal types. In other words, the internal exploration, external exploration and accept values were encoded by the IPS, ACC and mPFC, respectively. However, to support this notion, we consider that these signals should exhibit one important characteristic which is that

**Fig. 4 | Signals in the IPS, ACC and mPFC ($n$ = 24). A** The ROIs taken for the following time course analyses. **B**. An illustration of the time course of the internal exploration, external exploration and accept values in the IPS (left panel), ACC (middle panel) and mPFC (right panel), time-locked to the onset of the decision phase (solid black line), which could occur at the start of each trial or after each internal/external exploration. The IPS showed a positive internal exploration signal, but the absence of both an external exploration signal and an accept signal. The ACC showed a positive external exploration signal, a negative internal exploration signal and an absence of accept signal. The mPFC showed a positive external exploration signal, a positive accept signal, and a negative internal exploration signal. Together,

these positive signals verified that the whole-brain contrasts (Fig. 3) reflect positive tracking in the highlighted ROIs (IPS: internal exploration; ACC: external exploration; mPFC: Accept), rather than deactivation in the comparison conditions. **C** Individuals showing stronger internal exploration signals in the IPS tended to make more internal exploration decisions. Similarly, individuals with stronger external exploration signals in the ACC make more external exploration decisions. Individuals showing stronger mPFC accept signal tended to make more accept decisions, and those with stronger mPFC internal exploration signals made more internal exploration decisions. * denotes $p < 0.05$, ** denotes $p < 0.01$. Shaded areas represent ± SEM.

the signal should be invariant to the actual decision made subsequently. For example, a brain region that signals internal exploration value should remain encoding the same value regardless of whether an internal exploration, external exploration, or accept decision is made subsequently. This should be similar for a brain region that signals external exploration or accept value. Alternatively, a brain region may serve other functions while encoding these exploration parameters, such as taking a general role of representing decision value. As such, this brain region should flexibly encode the type of decision value according to the type of decision being made by showing an internal exploration signal during internal exploration, showing an external exploration signal during external exploration, and showing an accept signal during an accept decision. In short, a brain region that involves an exploration code should signal exploration value independent of the decision that is made. However, a brain region that involves a general decision code should flexibly signal value parameters depending on the type of decision made subsequently.

To test whether each brain region signals exploration value or general decision value, next we ran time course analyses within each trial subset defined by the impending decision (i.e., the internal exploration, external exploration or accept decision to be executed), and the three value regressors (internal exploration, external exploration and accept values) were entered simultaneously. Thus, all effects were reported from the same trials in that decision context. We then extracted the peak signal for each of the three decision types and carried out three one sample $t$ tests against zero. $p$ values reported for Fig. 5 are false discovery rate corrected (FDR-adjusted p; FDR procedure described in the Multiple-comparisons section in Methods).

In line with the literature which predominantly reports positive scaling of value signals[27,35,40–42], we focus on positive signals here to facilitate visualization of the signal patterns and cross-ROI comparisons. While all negative signals are reported in Supplementary Fig. S8. The analysis of the IPS and ACC signals confirmed their roles in signalling internal exploration value and external exploration value, respectively. In particular, we found that, in the IPS, the internal exploration signal remained significant in all three types of decision (Fig. 5A, top panel; internal exploration: $\beta = 0.103$, $t_{23} = 2.253$, FDR-adjusted $p = 0.034$, external exploration: $\beta = 0.080$, $t_{23} = 2.285$, FDR-adjusted $p = 0.034$; accept: $\beta = 0.075$, $t_{23} = 2.872$, FDR-adjusted $p = 0.026$).

The presence of a positive internal exploration signal that was independent of the type of impending decision confirmed that the IPS has a general role in reflecting internal exploration value. Given that the IPS was critical for signalling the demand for information, we further scrutinised the role of IPS in information gain by running an additional time course analysis. Information gain and internal exploration value were included as regressors (Supplementary Fig. S10). Our results suggested that the IPS would encode information gain and signal for an extra internal exploration decision if the results of the previous internal exploration decision were not satisfactory.

Similar to the signalling pattern of the IPS, the ACC reflected the value of external exploration regardless of the type of impending decision. There was a positive external exploration signal when participants made external exploration (Fig. 5B, middle panel; $\beta = 0.039$, $t_{23} = 2.094$, FDR-adjusted $p = 0.047$) or accept decisions ($\beta = 0.058$, $t_{23} = 2.102$, FDR-adjusted $p = 0.047$). During internal exploration, the effect of external exploration

signal was initially non-significant (Supplementary Fig. S8b, the inset in the top panel; $\beta = 0.091$, $t_{23} = 1.555$, FDR-adjusted $p = 0.134$).

Previous studies suggest that when individuals repeated the same exploitative decision before switching to an exploratory decision, the ACC signal also ramps up gradually[7,24]. In our study, we also noticed that participants often made internal exploration decisions repeatedly with the same option. Hence, we ran an additional analysis while the trials were defined by leaving out the first occasion when participants made an internal exploration decision with an option, and we hypothesized that the ACC signal should be more robust. Indeed, we found a significant external exploration signal (Fig. 5B, middle panel; $\beta = 0.209$, $t_{23} = 3.126$, FDR-adjusted $p = 0.014$)[40].

To confirm that the three brain regions involved a different code, we performed a three-way ANOVA with factors of Brain Regions (IPS, ACC and mPFC), Decision (internal exploration, external exploration and accept decisions) and Signal type (internal exploration, external exploration and accept signals), and the results showed a significant three-way interaction ($F(8, 621) = 5.79$, $p < 0.001$; FDR corrected post-hoc contrasts and for each Region × Decision × Signal cell are provided in Supplementary Fig. S9). Overall, these findings support the view that the IPS and ACC play a role in signalling the value of internal exploration and external exploration, respectively.

**mPFC showed general decision signal, instead of accept signal**
Interestingly, during internal exploration, the mPFC showed a significant internal exploration signal (Fig. 5c, top panel; $\beta = 0.166$, $t_{23} = 2.666$, FDR-adjusted $p = 0.014$). Furthermore, during external exploration, the mPFC showed external exploration signal (Fig. 5C, middle panel; $\beta = 0.071$, $t_{23} = 2.872$, FDR-adjusted $p = 0.026$). Based on the results in Figs. 2 and 3, we expected that the mPFC should also pass this test by showing an accept signal in all three types of decision. In the mPFC, we could only find positive accept signals during internal exploration decisions ($\beta = 0.155$, $t_{23} = 2.838$, FDR-adjusted $p = 0.014$), and accept decisions (Fig. 5C, bottom panel; $\beta = 0.179$, $t_{23} = 4.133$, FDR-adjusted $p = 0.001$.

Studies in reward-based decision making suggest that mPFC signals the reward value of the chosen option[35,40,43]. In Fig. 5C, we also observed that the nature of the mPFC's signal varied according to the type of impending decision (i.e., encoding accept value during an accept decision, internal exploration value during internal exploration, and external exploration value during external exploration). Together, these suggest that the mPFC may play a general role in encoding the decision value of the impending choice, regardless of the nature of the decision. Hence, we changed our frame of analysis by considering the value of the chosen option and that of the best alternative that was unchosen, regardless of the nature of the decision. By doing so, we found that the mPFC consistently showed a positive signal related to the chosen value during internal exploration (Fig. 6, left panel; $t_{23} = 2.954$, $p = 0.013$), external exploration (Fig. 6, middle panel; $t_{23} = 3.410$, $p = 0.006$), and accept (Fig. 6, right panel; $t_{23} = 2.273$, $p = 0.044$) decision. It also showed a negative signal related to the value of the best unchosen alternative during accept decision (Fig. 6, right panel; $t_{23} = -2.216$, $p = 0.049$), and a similar negative signal, despite not reaching significance, during internal (Fig. 6, left panel; $t_{23} = -1.257$, $p = 0.235$) and external (Fig. 6, middle panel; $t_{23} = -1.895$, $p = 0.085$) explorations. Taken together, these results support the influential neural common currency hypothesis

**Fig. 5 | The roles of the IPS, ACC and mPFC in internal exploration, external exploration and accept decisions (n = 24).** The bar charts illustrate the sizes of internal exploration, external exploration and accept signals extracted from the time course analysis in Supplementary Fig. S8. A Benjamini and Hochberg false-discovery-rate procedure (FDR, q = 0.05) was applied to the three decision-wise tests within each combination of region and signal, and the stars reflect these corrected p values (FDR-adjusted p). **A** In the IPS, the internal exploration signal remained significant in all three types of decision, suggesting its general role in signalling internal exploration value (**B**). In the ACC, the external exploration signal remained significant in all three types of decisions, suggesting its general role in signalling external exploration. Note that we left out the first occasion when participants made repetitive internal exploration decisions with the same option (see Supplementary Fig. S8). **C** Unlike the IPS and ACC that signaled a specific value independent of the impending decision, the mPFC flexibly adapted its signalling framework. Specifically, the mPFC showed a significant internal exploration signal, external exploration signal, and accept signal when the same type of decision was made. Signals that were not significant or significantly negative are presented as hollow bars. * denotes FDR-adjusted p < 0.05, ** denotes FDR-adjusted p < 0.01, *** denotes FDR-adjusted p ≤ 0.001. Error bars represent ± SEM.

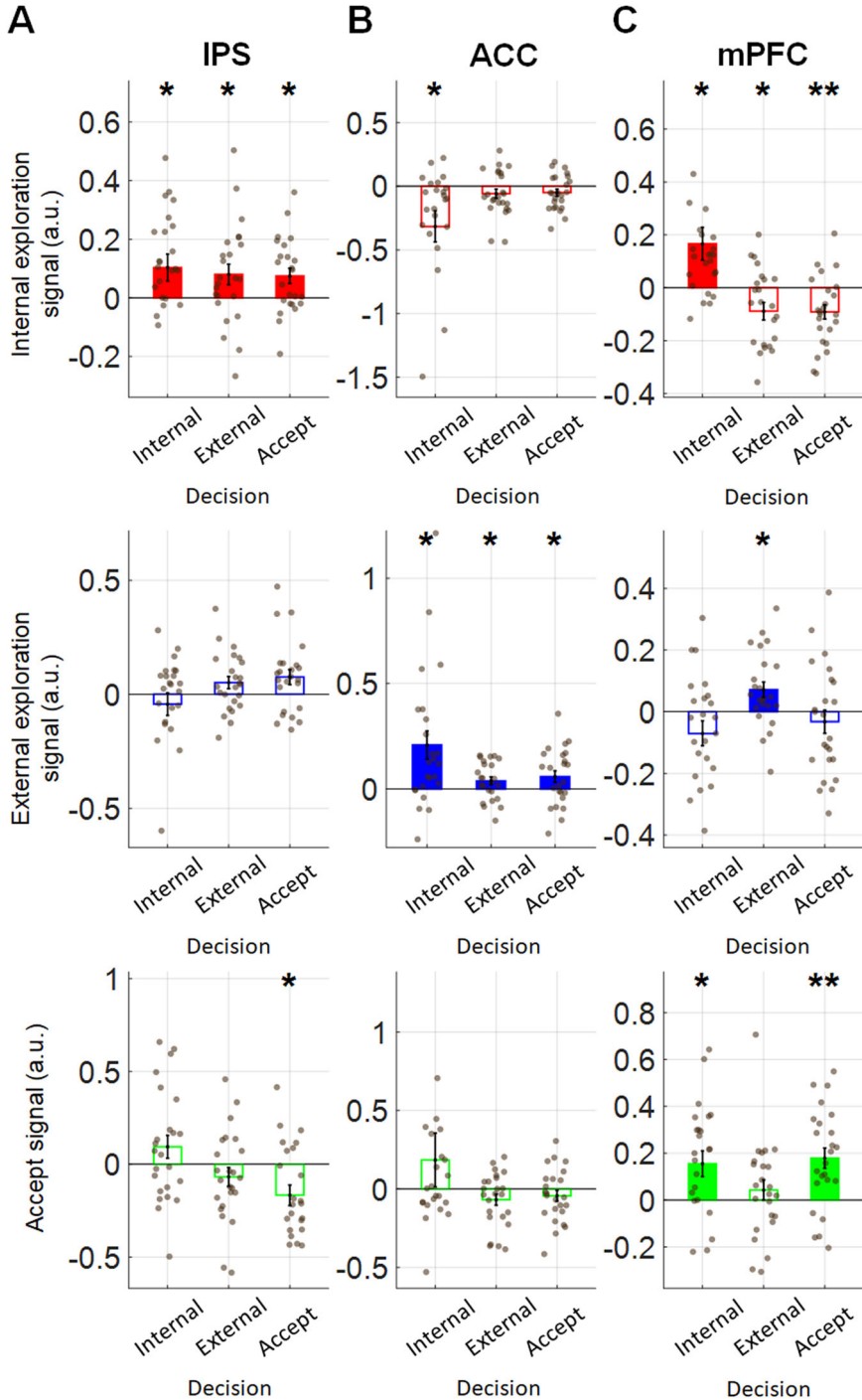

suggesting that a single brain region, often considered to be the mPFC or the orbitofrontal cortex[27,41], can signal general utility of any kinds of item. Our current findings extend this view by showing that the 'neural common currency' in mPFC can be generalized even to decisions of different nature.

## Discussion

Exploration in an uncertain environment empowers people to make informed decisions and manage risks. Much attention has been caught to understand the roles of IPS, ACC and mPFC during exploration. Despite their similarities in signalling exploration, it is broadly unclear whether and how they are specialised in different phases of exploration. Therefore, it is important to have direct comparisons of the activities of these three regions. We designed a multiple-option decision making task and identified the

unique patterns of activity in IPS, ACC and mPFC during exploration. In particular, we found that the IPS encodes the internal exploration value, the ACC encodes the external exploration value, and the mPFC encodes the general decision value.

There is a recent interest in disentangling whether the IPS is involved in signalling option value and exploratory information. In the past, considerable amount of studies demonstrated in macaques that the IPS, especially the lateral intraparietal region (LIP), signals reward value. These studies often involved options with a single dimension, such as the number of dots moving coherently to a certain direction[42,44], the amount of fluid/food received[16,45], or the probability of receiving a reward[16]. Recently, it has been shown, at least in humans, that the parietal cortex is only sensitive to the reward's quantity alone, but insensitive to the overall integrated value when

**Fig. 6 | General decision value signal in the mPFC (*n* = 24).** Time course of the chosen value and unchosen value signals in the mPFC in internal exploration, external exploration and accept decisions. Time-locked to the stimulus onset (solid black line). The dotted vertical line indicates the average reaction time (RT) of each decision type: Internal exploration (M = 3.76 s, SD = 1.56 s), external exploration (M = 2.75 s, SD = 1.19 s), and Accept (M = 3.06 s, SD = 1.50 s). The chosen value defined according to the type of impending decision (i.e., internal exploration value during internal exploration, external exploration value during external exploration and accept value during an accept decision), while the unchosen value was the best alternative. * denotes $p < 0.05$, ** denotes $p < 0.01$. Error bars represent ± SEM.

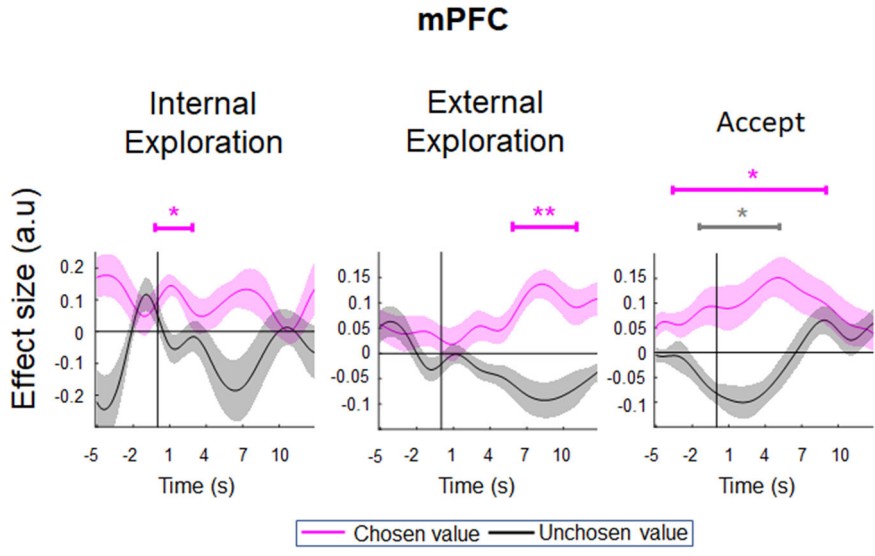

the reward is defined as by multiple dimensions (such as a combination of quantity and quality)[46]. Similarly, our current findings showed that there was an absence of correlation between the IPS activity and option's reward value (Figs. 4B, 5A).

There is an alternative view suggesting IPS is involved in signalling exploratory behaviours in uncertain environments. Horan and colleagues showed in the macaque IPS, activity of LIP neurons is modulated by the gain of information. In other words, during exploration, the LIP neurons responded when new information greatly reduced uncertainties. Similarly, human neuroimaging data showed that the IPS positively encoded the value of exploration[47]. Our current study advocates two different modes of exploration: internal exploration that aims to reduce uncertainties of existing options and external exploration that aims to identify new options. Our results further contribute to our understanding of the precise nature of the exploration signal in the IPS. We showed that the IPS guides internal exploration specifically via monitoring the uncertainty of an option (Fig. 5a, top panel) and the magnitude of information gain (Supplementary Fig. S10).

Our results are broadly consistent with previous studies reporting that ACC plays a key role in external exploration (Fig. 4B) and signals the need to switch to an alternative course of action[15,24,48]. Previous neuroimaging work reported a dissociation between subregions of ACC in which dorsal ACC activity decreased during exploitation while rostral ACC activity increased during exploitation and exploration[47]. Critically, our work expands on the role of ACC during repetitive choice. Our findings suggest that during internal exploration, ACC encoded the external exploration value (Fig. 5B, middle panel). This implicates that ACC constantly tracks the overall point in the environment and signals for behaviour change (e.g., choose another option), which was particularly the case when results of previous internal explorations are unsatisfactory. This is consistent with studies reporting an increase in ACC activity as a function of time spent in the same foraging patch[24], and ACC's role in encoding value of the environment[7]. Crucially, as ACC activity crosses a threshold, the animal switches to an alternative patch.

Aside from its well-established role in mediating exploratory-exploitative decisions, the ACC seems to be part of a network that deals with future-oriented decisions. Our results indicate that the human ACC tracks the overall point of the environment, which is a piece of information especially useful for strategizing rather than guiding a choice of short-term consequences. Along the same vein, macaque ACC was implicated in the encoding of the best current alternative, a piece of information that is useful for guiding future decision switches[49]. Critically, counterfactual choices did not modulate future switching behaviour after temporary disruption of ACC functioning, suggesting that ACC participates in generating decision

strategies for the future. In accordance with this view, Jahn et al. recently reported that the macaque pregenual anterior cingulate cortex (pgACC) was more active when the information is useful for the future[50], and the anterior and mid-cingulate cortex (MCC) was active only when information about the chosen and unchosen alternative was provided. Moving forward, it is important to examine whether a similar anatomical dichotomy exists in humans when dealing with long-term decisions. This is particularly interesting because exploratory behaviours in humans are not only related to the ACC, but additionally involves the lateral frontopolar cortex (FPl)[40,47]. Neuroanatomical studies showed that the FPl is greatly expanded in the human brain and a homolog is arguably absent in the macaque brain[51,52], however, the function of this additional activity in the FPl is broadly unclear. Recently, Law and colleagues suggested that the FPl is involved in decomposing complex decision information and such ability may be particularly useful during exploration[35]. Our preliminary analysis on FPl also showed that it encoded the external exploration value, providing additional insights into its functional role during exploration decisions (see Supplementary Fig. S11). Future work may also address the unique functions of ACC and FPl for guiding exploratory decisions in humans.

Expanding on the neural common currency hypothesis, we propose that the mPFC signals a general decision value. The neural common hypothesis posits that the mPFC, sometimes more specifically the vmPFC, is involved in all kinds of decisions and represents values of almost all reward types on a common scale[27]. Here, we further speculate that the mPFC encodes decision values that are relevant to the current goal. Our results indicated that the mPFC encodes the value of internal exploration, external exploration, and accept decisions when these decisions were made accordingly (Fig. 5c). For example, the mPFC showed a positive internal exploration value during internal exploration and a negative internal exploration value when making external exploration or accept decisions. In the same vein, the mPFC signalled a positive external exploration value exclusively when making external exploration decisions, and a positive accept value exclusively when making accept decisions. The role of mPFC in encoding the average reward of all possible items in the environment (i.e., external exploration value) further corroborates with the view that mPFC carries goal relevant signals but does not exclusively carry the value of an accept decision[5,7,53].

Like the IPS, the mPFC encodes the internal exploration value positively during internal exploration. In accordance with the view that mPFC represents value of an option, one may expect a negative internal exploration value that is discounted by the uncertainty of an option, or a positive internal exploration value which deems uncertain options to be worth exploring. Based on the positive mPFC signal during internal exploration observed in

this study, we speculate that the mPFC sees choices with greater uncertainty levels as more relevant to the current goal. Although this is broadly in line with the neural common currency hypothesis, it is worth noting that the hypothesis may not be generalised universally to all kinds of decisions. In decision making that involves complex information, the FPl was implicated in decomposing complex and abstract information to speculate the long-term consequences of the choices[35]. In contrast, the vmPFC signals for simpler decisions. Here, although individuals had to speculate which option to explore would bring about the best results, individuals were allowed to eliminate uncertainty step by step, which involves lower levels of abstraction and prospection relative to that of complex choice. Taken together, the mPFC is critical for encoding decision values relevant to the current goal when information complexity is low.

Our work revealed the dissociable roles of the IPS, ACC and mPFC in exploratory decision making. Broadly consistent with prior works, the IPS signals internal exploration value whereas the ACC signals external exploration value. The mPFC signals a general decision value and actively participates in different phases of exploratory decision making. Although exploration empowers people to make informed decisions, excessive exploration may impede optimal outcomes. For example, patients with psychological disorders such as depression and obsessive compulsive disorders are associated with ACC dysfunction and atypical information sampling behaviour[54,55]. Future work may look into the relationship between ACC and atypical information sampling strategies.

## Methods

### Participants

Healthy participants provided written informed consent in line with procedures approved by the ethics committees of The Hong Kong Polytechnic University and National Institute of Information and Communications Technology, Japan, and methods conformed to the relevant guidelines and regulations. All participants were screened for MRI contraindications, and had normal or corrected-to-normal vision. A total of 26 participants completed this experiment (age: M = 25.923, SD = 3.236; males = 12). Two participants were excluded from data analysis due to excessive movement during MRI scanning. All ethical regulations relevant to human research participants were followed.

### Task

**Multiple-option decision-making task**. On each trial, participants were first offered an option in one of the nine possible positions (black boxes; Fig. 1A). Each option was composed of four dials, the coloured area of each dial indicates the possible number of points that could be obtained. Only one dial, identity unbeknown to participants, was related to the actual points earned by each option, such that options containing more variable dials had greater uncertainties. Participants should decide between three possible responses. (1) *Accept*, to end the trial by selecting an option and earning its actual points, followed by an outcome phrase that varied randomly between 1000 and 3000 ms, and an intertrial interval (ITI) that varied randomly between 3000 and 6000 ms. (2) *Internal exploration*, to randomly remove one dial of a selected option to reduce its uncertainty (i.e., a maximum of three internal explorations could be made for each option). (3) *External exploration*, to reveal a new option from one of the remaining black boxes (i.e., a maximum of eight external explorations could be made for each trial). Each internal or external exploration costed one point and the cumulated cost incurred by the end of a trial was displayed in real time at the top left-hand corner. A variable interstimulus interval (ISI) ranging from 1000 to 3000 ms was implemented after each internal or external exploration. The top right corner indicated the average point of all (hidden and revealed) options of the same trial, which was important for guiding external exploration decisions. The task consisted of 100 trials. Participants completed the decision-making task while undergoing fMRI. Prior to MRI scanning, participants completed a practice session to familiarise with the experimental task.

## Behavioural analysis - parameter optimisation and model comparison

We developed a General Linear Model (GLM) to aid fitting of value functions for describing participants' internal exploration, external exploration, and accept decisions. The GLM involved a multinomial logistic regression analysis to estimate the probability of each decision type that follows two logit equations:

$$
\begin{aligned}
In\left(\frac{P(y = Internal\ exploration)}{P(y = Accept)}\right) &= \beta_1 + \beta_2 Value_{Internal\ exploration} \\
&+ \beta_3 Value_{External\ exploration} + \beta_4 Value_{Accept} + \beta_5 Cost \\
In\left(\frac{P(y = External\ exploration)}{P(y = Accept)}\right) &= \beta_6 + \beta_7 Value_{Internal\ exploration} \\
&+ \beta_8 Value_{External\ exploration} + \beta_9 Value_{Accept} + \beta_{10} Cost
\end{aligned}
\tag{1}
$$

where $y$ is the decision, and $P(y = Accept)$, $P(y = Internal\ exploration)$, and $P(y = External\ exploration)$ are the choice probability of accept, internal exploration, and external exploration decisions, respectively. With three choice probabilities that sum to one, only two independent comparisons are identifiable that a reference category is fixed for estimation. We set accept as the reference, which provides a non-exploratory baseline common to both exploratory branches. The model estimates two log-odds contrasts (internal exploration versus accept and external exploration versus accept). Coefficients indicate how a one-unit increase in a predictor changes the odds of selecting the corresponding exploratory action rather than accepting. The choice of reference does not affect fitted probabilities or overall model fit; it only alters coefficient labeling and interpretation. The model incorporated $Value_{Internal\ exploration}$, $Value_{External\ exploration}$, $Value_{Accept}$ and $Cost$ as predictors, where the three values are values of their corresponding decisions, and the cost is cumulated by each internal or external exploration within a trial.

Based on the GLM, we applied different operational definitions of each decision value to develop 120 variants of the GLM (Supplementary Fig. S1). We identified the best model by comparing the goodness-of-fit of the variants using the Bayesian Information Criterion (BIC, see Supplementary Table S1 for detailed statistics). The functions of the accept, internal exploration and external exploration values identified in the best fitting model are as follows:

$$
Value_{Internal\ exploration} = \max(Point_{(option)} * StandardDeviation_{(option)})
\tag{2}
$$

$$
Value_{External\ exploration} = \overline{Point}_{(all\ options)}
\tag{3}
$$

$$
Value_{Accept} = Point_{(the\ best\ option)}
\tag{4}
$$

where $Value_{Accept}$ was defined as the average point of the best option. $Value_{Internal\ exploration}$ was defined as the greatest product of the average point and the standard deviation among all revealed options, suggesting that internal exploration is focused on options that potentially lead to high gains but also contain great uncertainties. $Value_{External\ exploration}$ was defined as the average point of all hidden and revealed options, suggesting that exploring new options is more likely when the environment generally contains high gain options. The model arbitrates between the choices of accept, internal exploration and external exploration based on their corresponding value estimates, defined based on the best-fit model.

## Neuroimaging data acquisition and preprocessing

Neuroimaging data were acquired using a 3 Tesla Siemens Magnetom Prisma MR scanner at Center for Information and Neural Networks, National Institute of Information and Communications Technology (CiNet, NICT, Osaka, Japan) using a 32-channel phase array coil. Echo-planar imaging data was obtained from 72 slices [whole brain coverage,

repetition time (TR) = 2000 ms, echo time (TE) = 30 ms, field of view (FOV) = 200 ×200, flip angle = 75°. Field maps were acquired to correct for signal distortions using a dual echo 2D gradient echo sequence (TR = 75 ms, TE1 = 5.16 ms, TE2 = 7.62 ms, FOV = 200 × 200, flip angle = 90 deg, 2.5 × 2.5 × 2.5 mm3 resolution). For each participant, a high-resolution T1 image was acquired using a 1mm3 anatomical scan (208 slices, TR = 1900 ms, TE = 2.48 ms, FOV = 256 × 256, flip angle = 9°) for accurate co-registration of fMRI images to individual anatomy space and for reconstructing cortical surfaces.

Imaging data was analysed using FMRIB's Software Library (FSL)[56]. fMRI data were preprocessed using brain extraction (Brain Extraction Tool)[57], motion correction (FMRIB's Linear Image Registration Tool)[58], Gaussian spatial smoothing with full width at half maximum (FWHM) sizes of 5 mm, field-map correction for distorted signal[58], and high-pass temporal filtering (3 dB cut-off of 100 s). Functional images were aligned with each participant's anatomical scan and transformed to Montreal Neurological Institute (MNI) space[59].

## fMRI data analysis

Functional data were further analysed at two levels: using both a whole-brain analysis (univariate GLM) with FEAT[58,60] and a region-of-interest (ROI) time course analysis using MATLAB (The MathWorks).

**Whole-brain analysis.** To identify brain regions associated with internal exploration, external exploration and accept decisions, we performed whole-brain analyses using a univariate GLM approach (Fig. 3). We entered the internal exploration value, external exploration value, accept value, cumulative cost that were time-locked to stimulus onset at the beginning of each trial as regressors in the GLM, convolved with a canonical hemodynamic function (two-gamma model)[61] to provide idealised hemodynamic responses. 12 additional nuisance regressors were also included. Four parametric regressors related to the number of unveiled options, the accumulative gain across trial, the trial gain and the sum of the points of all unveiled options, time-locked to the stimulus onset, were included. Six box car regressors related to participants' motor movements, time-locked to their motor movements; interstimulus interval (isi), time-locked to stimulus onset, intertrial interval (iti), time-locked to trial onset; outcome phase, time-locked to outcome phase onset; trial durations used two types of event constants (duration = 1 and from trial onset to every decision), time-locked to participants' decisions, were included. Finally, two regressors related to the average BOLD signal in the cerebrospinal fluid (CSF) and white matter (WM) were included. At group level, FMRIB's local analysis of mixed effects was applied with outlier deweighing. All images were cluster-corrected results with voxel inclusion threshold of z = 3.1, and threshold of cluster significance at 0.05 (p-value).

**Region-of-interest (ROI) analysis.** To examine the dynamics of decision making (i.e., internal exploration, external exploration, and accept decisions), we extracted the time series of brain regions previously identified in the whole brain analysis. All ROIs were first defined based on literature-derived coordinates. Masks were then created to extract activity of ROIs. IPS, which has been implicated in uncertainty driven exploratory behaviour, was defined as a spherical ROI (3 mm rad) centered on MNI coordinates [50, -44, 43][62]. ACC, a region consistently associated with tracking search or foraging value, was defined similarly [0, 28, 30], coordinates from Kolling et al. [8]. The mPFC, wherein the value of the chosen option was reportedly encoded, was defined as [0, 51, 0], coordinates taken from Blair et al. [63]. The extracted ROI time courses were time-locked to stimulus onset. For each participant, the ROI activities at each time point were regressed via a GLM, such that time courses of beta weight for each regressor in the GLM were acquired. The beta weight time courses were then group averaged. Second, we extracted the peak of each signal time course to compute individual variability by

correlating decision value signals with decision type, and to measure signal distribution by the type of decision. The size of each peak was extracted for each participant, followed by running a one-sample t test to test whether each peak in the group time courses significantly differed from zero. Since traditional frequentists statistics is not ideal for confirming null effects, each analysis was also supplemented by a Bayesian t test. A leave-one-subject-out procedure was repeated 23 times, and ultimately averaged, to find the peak time (within a window of -5 to 13 s). Peak window selection was done by finding the full-width half-maximum of the peak in the group time course, which was determined as the period between two time-points at the peak half maximum value (i.e., regression weight).

**Multiple-comparisons correction.** To test whether each region encodes a given signal independently of the impending decision while controlling for multiple comparisons in the tests shown in Fig. 5, we first obtained the peak signal for the internal exploration, external exploration and accept decisions and performed three one-sample t tests against zero (one per decision type). We then controlled the false discovery rate across these three tests within each combination of region and signal using the Benjamini and Hochberg procedure (FDR, q = 0.05). In the Results, we refer to these adjusted values as FDR-adjusted p.

## Statistics and reproducibility

All statistical analyses were performed using MATLAB (The MathWorks) and FMRIB's Software Library (FSL)[56]. All data and code used are available and can be found at https://osf.io/zcfxy (https://doi.org/10.17605/OSF.IO/ZCFXY)[64] to ensure reproducibility.

## Data availability

All data can be found at https://osf.io/zcfxy (https://doi.org/10.17605/OSF.IO/ZCFXY)[64].

## Code availability

Custom code used can be found at https://osf.io/zcfxy (https://doi.org/10.17605/OSF.IO/ZCFXY)[64].

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

## Acknowledgements

The current work is supported by the Hong Kong Research Grants Council (15105522) and The Hong Kong Polytechnic University Project of Strategic Importance.

## Author contributions

T.F.W., V.K.S.C., C.K.L., B.K.H.C. designed the study. T.F.W., K.W., M.H., B.K.H.C. were involved in MR data acquisition. V.K.S.C., B.K.H.C. performed data analyses. V.K.S.C., N.H.L.W., T.F.W., K.W, M.H, C.K.L, B..K.H.C contributed to the preparation of this manuscript.

## Competing interests

The authors declare no competing interests.
