## [Transparent Peer Review file · Communications Biology]

Distinct contributions of prefrontal, parietal, and cingulate signals to exploratory decisions.

Corresponding Author: Mr Victor K.S. Chan

Version 0:

Reviewer comments:

Reviewer #1

(Remarks to the Author)

Summary

In this study, the authors aim to identify different characteristics of exploratory decisions, proposing a way to bridge two seemingly opposing theories: that people either seek more information or reduce uncertainty during exploratory decision-making. The authors break down exploratory decisions into two subtypes: internal exploration (to reduce uncertainty about the outcome of the current option) and external exploration (to gather information about alternative options). They design a well-structured study to distinguish each type of exploration and compare these decisions to the choice of accepting an option based on its current expected reward.

In their literature review, the authors highlight studies demonstrating that intraparietal sulcus (IPS) activity correlates with information gathering to reduce uncertainty, anterior cingulate cortex (ACC) activity correlates with tracking alternatives, and ventromedial prefrontal cortex (vmPFC) activity relates to general decision value in value-based decision-making. They hypothesize that each of these regions plays a complementary role in exploratory decisions and test whether the IPS supports internal exploration, the ACC external exploration, and the medial prefrontal cortex (mPFC) decision-making—whether this involves exploring internally, exploring externally, or exploiting the current option.

Method

The authors ingeniously designed a task that distinguishes exploratory decisions into external and internal exploration, enabling comparisons between these three types of decisions. They provide multiple metrics to quantify each value type (value of external exploration, value of internal exploration, and value of accepting the current choice) and use the Bayesian Information Criterion (BIC) to select the best-fitting model. Using this model, they study whole-brain fMRI activity, focusing specifically on their hypothesized regions of interest.

Key Findings

The findings reveal specific encoding roles for each region:

- The IPS encodes internal decision values,
- The ACC encodes external decision values,
- The mPFC encodes internal, external, or accept decision values depending on subsequent participant choices. More specifically, the mPFC encodes the value of the chosen option regardless of the decision and the value of the best unchosen alternative when participants accept the current option.

The authors suggest a collaborative role among these regions:

- The IPS tracks uncertainty,
- The ACC identifies alternative needs,
- The mPFC processes decision-relevant information, supporting the neural common currency hypothesis, suggesting that the mPFC enables a decision regardless of the value of the choice itself.

My General Review:

Overall, I appreciate the authors' approach to this topic and their task design. However, I believe the paper could benefit from (1) clarifying certain concepts or findings, (2) providing more analysis to strengthen their arguments and meet the standards of Communications Biology, and (3) connecting their work more fully to the literature for greater impact. Missing out on these points would prevent the paper from reaching its full potential. My suggestions are as follows:

- Provide further analysis detailing participant behavior,
- Decode choices from neural activity,
- Clarify the role of the ACC with more reference to the literature,
- Specify the ACC's anatomical definition in their analysis, introduction, and discussion,
- Specify the mPFC's anatomy in their analysis, introduction, and discussion,
- Add the vmPFC to the regions of interest (ROIs) or explain why it is excluded,
- Include the frontopolar cortex (FPI) in ROI analysis, or explain why it is excluded.

What I Think is Cool About This Paper and Why:

Elegance of the Task Design

The experimental task is thoughtfully designed to isolate different types of decision-related information.

Aim to Characterize a Network with Regional Functional Differentiation

I appreciate the authors' attempt to characterize each brain region as part of a complementary network, linking seemingly contradictory literature. Accurate functional and/or anatomical characterization is essential for achieving this.

Bayesian Model Selection

Using Bayesian model selection to identify the best-fitting decision-making model for each participant is an excellent approach. However, the authors should expand on this by explaining each metric's behavioral implications and characterizing behavior more thoroughly.

Relating Brain Activity to Behavior

The authors successfully demonstrate that brain activity explains specific choices. Further behavioral analysis and anatomically precise brain regions would strengthen this relationship.

Areas of Confusion

Role of the Lateral Frontal Pole (FPI)

The authors mention the FPI in the discussion and suggest that it interacts with the mPFC. They find a positive correlation between FPI activity and internal exploration in the whole-brain analysis and suggest that future research could investigate this further. However, it's unclear why they did not include the FPI in the network they study.

ACC's Role in Conflict/Control

The authors mention the ACC's involvement in tracking alternatives and future goals in the introduction and discussion, but they do not address its roles in conflict, control and difficulty (for example). Clarifying how their study fits into this literature would be beneficial.

mPFC or vmPFC

It is unclear why the mPFC was chosen over the vmPFC, given their introduction and discussion concerns the vmPFC mostly.

Interpretation of Temporal Deconstruction in fMRI

In Figure 4B, the onset of external exploration signals in the ACC appears to start before stimulus onset, raising questions about the interpretation.

Model Comparison (Figure S2)

The supplementary material could better clarify the metrics compared in the correlation and whether the third-best fit represents the average best fit for each subject. If so it would be informative to have a distribution of which models are the best fit for each subject.

Chosen/Unchosen Analysis

In Figure 6, what does it mean that the mPFC encodes the value for choosing internal exploration before stimulus onset?

Given that the ACC tracks external exploration and there interpretation about alternatives, why did they not look at unchosen options in the ACC, as they did in the mPFC?

My Suggestions for specific Corrections/clarification:

Sentences that I find problematic or that I think need clarification

- Main paper p.4 line 75 "[...] These findings were intriguing because it provides an alternative account of the widely reported role of IPS in signalling reward." It would be nice for them to make explicit which widely reported role and provide references about "the widely reported role of IPS in signaling reward" – does the IPS track the value of reward in 1D, does it respond in an acute way, does it track the cumulative reward if only on 1D, is it always the same dimension? Which widely reported role are they talking about, and which studies say so (doesn't have to be an exhaustive list).

- Main paper p.19 line 371 to 374: "Taken together, these results support the influential neural common currency hypothesis suggesting that a single brain region, often considered to be the mPFC or the orbitofrontal cortex, can signal general utility of any kinds of item." I don't understand what they mean by "often considered to be OFC or mPFC"; they do not look at the OFC nor mention it; they do not reference this sentence.

- In Sup mat, p.3: "The first type of the external exploration value was formulated as the average point of all hidden and revealed options ($Value_{External\ Exploration}(1)$). The second type of the external exploration value was formulated as the

average point amongst all revealed options ($ValueExternal\ Exploration(2)$)." In the formula, ($ValueExternal\ Exploration(1)$) = Points(all options) and ($ValueExternal\ Exploration(2)$) = Points(all existing options). It sounds like "All existing options" means all options including the hidden ones. If that is correct, there is a mistake in the indexing, where ($ValueExternal\ Exploration(1)$) is the average point of all hidden and revealed options and therefore = Points(all existing options). If my understanding is incorrect, this should be clarified. It also asks the question of how participants can estimate the value of hidden possibilities?

- Figure 1B: The proportion of accept, internal and external: explicitly say if this is the proportion of those decisions as a first decision on every trial. I understand Figure 1B second and third rows to be respectively the proportion of internal and external decisions for each number of decisions made before an accept decision. If so, another way to plot this in a more informative way could be to stack on top of each other the proportion of internal, external and accept choices as a function of number of decisions before reaching accept and finishing the trial.

- Figure 2B Clarity: Could do with the equation on the figure for each measure of internal, external and accept value. Also, it would be nice to clarify how the authors binned the trial to analyse 3 decision option into a binary way what is being compared (choice proportion done in a binary way).

- Colour coding in the paper (Figure 4 and 5) throughout the paper, colour is confusing as red means IPS signal but also means internal exploration, blue ACC but also external, and green means mPFC signal but also accept. Although the reasoning for making that association makes sense, it limits / hinders the clarity of what is tested on each graph (is what is plotted the behaviour or the brain activity?). If the results are compelling enough, the colour coding doesn't need to force interpretation but guide it.

- stimulus onset timing and decision distribution within time window: There's ambiguity regarding which stimulus onset they focus on—whether it's the initial stimulus onset (i.e. the beginning of the trial) or the latest stimulus change in each response. If it's the second answer, then doesn't this mean that there are different number of trials and/or timing of the trial for each decision type? This raises questions about the nature of the signal that the authors are comparing – it would be helpful to know their perspective on this rather than have to infer it. They mention that for some analysis they removed the first trial/decision, does that mean that they included all the other decisions regardless of the distributions of decisions that were made at each decision number?

- figure 4C : i like this analysis and understand it as a demonstration of how neural activity reflects the behaviour of each participant. However, the individual variability title through me off because it sounds like they will now look at individual differences in this study – do they from then on use the best fit for each participant instead of the best fit model? What do they want to demonstrate? if they want to investigate the neural correlates that support the decisions of participant then I feel they should clarify in text/legend and also either by changing the title or removing it all together for consistency given that other figure do not have one. If that is not it, I feel it would be helpful to make to goal clear and add why that is important/relevant to the paper.

- Figure 5 Clarity: The hollow bars' meaning is unclear to me—it says that they indicate non-significance or negativity. However, some bars appear negative, and some include significance stars.

My Suggestions for Further Analysis

1. Behavioural Analysis:

To better understand the figures and explore what this task can tell us about exploration, it would be beneficial to conduct further analysis on participant behaviour.

This could include (but these are just suggestions) examining:

- The stacked bar analysis suggested previously here would give an understanding of the distribution of choices across trials and author could use this to illustrate which decisions they used to compare the different type of signals (first or second decision)

- Average points participants obtained on a trial

- The minimum expected value at which participants make an "accept" choice. If Figure 2B correctly indicates this, it would be helpful to have the authors explicitly describe their interpretation in relation to this figure.

- Patterns of alternation between internal and external exploration before accepting (e.g., the proportions of "In-Ex-Ac," "Ex-In-Ac," "In-In-Ex-Ex-Ac," where "In" represents Internal Exploration, "Ex" represents External Exploration, and "Ac" represents Accept decisions).

- The average expected value at which participants switch to accepting after beginning external exploration, and possibly after starting internal exploration if there is no constraint on the order of decisions. This could reveal whether participants' exploration is affected by the known range or distribution of outcomes. If participants are unaware of this range and mean, exploring may make them more likely to revert to previous options; however, if they know the range and mean, their

decisions could reflect an ability to compare current options with unknown alternatives.

2. Inclusion of the frontopolar cortex:

The frontopolar cortex (FPI) seems relevant to internal exploration (as indicated by their analysis) and to tracking alternatives. Studies by Badre et al. (2012), Boorman et al. (2011), Donoso et al. (2014), Tomov et al. (2020), Yoshida & Ishii (2006), and Zajkowski et al. (2017) emphasize the FPI's involvement in these processes. Additionally, several studies have demonstrated the role of the FPI in exploration in non-human primates (Boschin et al., 2015; Mansouri et al., 2015; Tsujimoto et al., 2010, 2012). It would enhance the paper to either include FPI in the ROI analysis or justify its exclusion with a scientific rationale.

3. Further characterization of the ACC

- Conflict/Control Framework and More

There are many different perspectives on the function of the ACC, particularly regarding its subdivisions (Clairs & Lopez-Persem, 2023; Vassena et al., 2017). It would be valuable to see how the authors' results align with existing literature on control and conflict (Botvinick et al., 2001, 2004, Shenhav et al. 2016) and Kolling et al. 2016). For example, if the values of internal, external, and accept choices are close, this could lead to higher conflict or difficulty in making a choice. Examining ACC activity on these trials would acknowledge the ACC's established role in conflict or difficulty and its relation to task-switching.

- Functional Anatomy of the ACC

Clarifying which part of the ACC is under examination and the rationale behind it would strengthen the paper, especially given the anatomical and functional subdivisions highlighted in the introduction. If the authors have chosen a spherical ROI, discussing the implications of this choice—such as the potential averaging effect across opposing activities in different subregions—would provide insight into the results. Are there specific subregions they aim to measure, and how might this choice affect the findings?

4. Predictive ROI Analysis to Strengthen Their Demonstration:

It would be interesting to see if ACC, IPS, and mPFC activity can independently predict a participant's choice and if the ACC and IPS could do so above and beyond any other information than the external exploration and internal exploration respectively?

For example: If their hypothesis is correct, IPS might predict internal exploration choices but no others decisions, ACC might best predict external exploration but not others, and mPFC might predict all choices.

5. Interaction Between Frontal Pole & mPFC ; and between Frontal pole & ACC:

Testing their suggested interaction between the frontal pole and mPFC, and between the ACC and frontal pole, might reveal distinct functional roles. Given their describe role of the ACC in planning future choices and selecting the best alternative for the future, I would also discuss the frontal pole in predicting future decisions and tracking alternatives and verify their respective roles here

REFERENCES

- Badre, D., Doll, B. B., Long, N. M., & Frank, M. J. (2012). Rostrolateral prefrontal cortex and individual differences in uncertainty-driven exploration. *Neuron*, 73(3), 595–607. <https://doi.org/10.1016/j.neuron.2011.12.025>
- Boorman, E. D., Behrens, T. E., & Rushworth, M. F. (2011). Counterfactual Choice and Learning in a Neural Network Centered on Human Lateral Frontopolar Cortex. *PLOS Biology*, 9(6), e1001093. <https://doi.org/10.1371/journal.pbio.1001093>
- Boschin, E. A., Piekema, C., & Buckley, M. J. (2015). Essential functions of primate frontopolar cortex in cognition. *Proceedings of the National Academy of Sciences*, 112(9), E1020–E1027. <https://doi.org/10.1073/pnas.1419649112>
- Botvinick, M. M., Braver, T. S., Barch, D. M., Carter, C. S., & Cohen, J. D. (2001). Conflict monitoring and cognitive control. *Psychological review*, 108(3), 624.
- Botvinick, M. M., Cohen, J. D., & Carter, C. S. (2004). Conflict monitoring and anterior cingulate cortex: an update. *Trends in cognitive sciences*, 8(12), 539-546.
- Clairis, N., & Lopez-Persem, A., (2023). Debates on the dorsomedial prefrontal/dorsal anterior cingulate cortex: insights for future research, *Brain*, 146 (12), 4826–4844, <https://doi.org/10.1093/brain/awad263>
- Donoso, M., Collins, A. G. E., & Koechlin, E. (2014). Foundations of human reasoning in the prefrontal cortex. *Science*, 344(6191), 1481–1486. <https://doi.org/10.1126/science.1252254>
- Kolling, N., Behrens, T. E. J., Wittmann, M. K., & Rushworth, M. F. S. (2016). Multiple signals in anterior cingulate cortex. *Current opinion in neurobiology*, 37, 36-43.
- Mansouri, F. A., Buckley, M. J., Mahboubi, M., & Tanaka, K. (2015). Behavioral consequences of selective damage to frontal pole and posterior cingulate cortices. *Proceedings of the National Academy of Sciences*, 112(29). <https://doi.org/10.1073/pnas.1422629112>
- Shenhav, A., Cohen, J. D., & Botvinick, M. M. (2016). Dorsal anterior cingulate cortex and the value of control. *Nature neuroscience*, 19(10), 1286-1291.
- Tomov, M. S., Truong, V. Q., Hundia, R. A., & Gershman, S. J. (2020). Dissociable neural correlates of uncertainty underlie different exploration strategies. *Nature Communications*, 11(1). <https://doi.org/10.1038/s41467-020-15766-z>
- Tsujimoto, S., Genovesio, A., & Wise, S. P. (2010). Evaluating self-generated decisions in frontal pole cortex of monkeys. *Nature Neuroscience*, 13(1), Article 1. <https://doi.org/10.1038/nn.2453>
- Tsujimoto, S., Genovesio, A., & Wise, S. P. (2012). Neuronal Activity during a Cued Strategy Task: Comparison of Dorsolateral, Orbital, and Polar Prefrontal Cortex. *Journal of Neuroscience*, 32(32), 11017–11031. <https://doi.org/10.1523/JNEUROSCI.1230-12.2012>

- Vassena, E., Holroyd, C. B., & Alexander, W. H. (2017). Computational models of anterior cingulate cortex: At the crossroads between prediction and effort. *Frontiers in neuroscience*, 11, 316.
- Yoshida, W., & Ishii, S. (2006). Resolution of Uncertainty in Prefrontal Cortex. *Neuron*, 50(5), 781–789. <https://doi.org/10.1016/j.neuron.2006.05.006>
- Zajkowski, W. K., Kossut, M., & Wilson, R. C. (2017). A causal role for right frontopolar cortex in directed, but not random, exploration. *eLife*, 6, e27430. <https://doi.org/10.7554/eLife.27430>

Reviewer #2

(Remarks to the Author)

The authors present a new behavioral paradigm in which participants can explore the environment (adds a new option with four gambles), explore an existing option (decreases the number of gambles within one option), or accept an existing option (one gamble within the selected option is played). Participants perform the task during fMRI scanning, and model-based fMRI is used for the analysis.

The main findings claim that the computations contributing to the three types of decisions (internal- or external- exploration, or accept) are represented in an '... appealing one-to-one matching between brain regions and value signal types.' As such, the IPS, ACC, and the mPFC respectively encode internal, external, and accept values independently of decision-type. By contrast, the authors also report that the mPFC encodes all value-types based on the impending decision.

I found the Abstract promising and intriguing, and the task novel and interesting, but I'm not convinced by the results. In particular, the modeling section needs to be much improved and the fMRI analysis suffers from multiple comparison issues. My suggestions are presented below.

Main concerns:

• Modeling

The authors compare 90 different models, each with a different combination of computations believed to underpin the different decisions. Getting the model right is crucial for the interpretation of both behavior and fMRI data.

- If I understand correctly, all tested models have 10 parameters (lines 516-517). However, simpler models may provide more parsimonious fits to behavior. For example, in theory a model with only an Accept value could provide the most parsimonious fit to behavior (exploration rate would then be determined by the inverse temperature parameter). Accordingly, please also include all nested models in your model comparisons (i.e. models with two and one parameter(s) only).

- Using a fixed-effect analysis for model comparison (e.g. summed BIC) is now quite frowned upon, and I would strongly recommend using a different approach, such as calculating exceedance probabilities (Rigoux et al., 2014, Bayesian model selection for group studies - revisited).

- I sincerely appreciate the robustness analysis replicating findings using second-best models et cetera, but please add a table with log-likelihoods and BICs to make the comparison between models easier.

- The BICs provide a measure of the relative fit between models, but doesn't tell us if a model is doing a good job or not (there is always a winner even when comparing relative fits between only bad models). Therefore, please demonstrate how well your winning model replicates actual choice behavior by showing, for example, the model-predicted probability of making the same choice as participants.

- It seems to me that some models are equivalent. For example, the best model (IE:5, EE:1, AV:1) should perfectly overlap with the (IE:5, EE:5, AV:1) model. Providing tables and conducting the model-recovery procedures will avoid such potential concerns. Accordingly, please perform a model-recovery analysis for all tested models, as well as a parameter-recovery analysis for your winning model. The validity of these approaches have been nicely described (Wilson & Collins, 2019, Ten simple rules for the computational modeling of behavioral data).

- Please describe in detail how your models were fitted. Your explanation (lines 513-521) is short and lack details, making the procedure difficult to understand for someone unfamiliar with multinomial logistic regression.

- The mean-variance tradeoff is known to drive decision making. Please include another computation for ValueAccept as: Point (best option) – Variance(best option). Please observe that this model will have two betas, one for Point and one for Variance.

• fMRI analysis

- Could you please describe the timings used for the paradigm in more detail? For example, how long was the time between making a decision and the feedback presentation? Was there any kind of jitter? This knowledge is important to remove any doubts of 'smeared' brain activation related to decision computations (e.g. Accept value) and feedback (e.g. actual outcome).

- I'm puzzled by the contrasts shown in Fig. 3. In the GLM, the three decision values are entered as separate regressors. Accordingly, to assess the neural correlates of each decision value, the average betas should be compared to 0. However, in Fig. 3 it is stated that the average betas are contrasted with the average effect of the remaining two types of value. Why is this necessary and how is this mean calculated? What are the results if the average betas are compared to 0?

- How were multiple comparisons corrected for? For example, Fig. 5 contains 27 different comparisons (yielding a Bonferroni-corrected alpha of 0.0019). Personally, I would conduct ANOVA-type analysis with appropriate post-hoc comparisons and corrections applied. This would provide a clearer oversight of the results, such as potential interactions between brain regions and decision values. In Fig. 5, please display individual data-points (e.g. violin-plot) rather than just mean+errorbar.

Minor concerns:

• BIC is not needed when comparing models with the same number of factors.

• It is problematic to assign specific computations to specific decisions. For example, intuitively, Point(best)-Point(average) is for me what drives an Accept decision. However, the authors argue that the negative (i.e. Point(average)-Point(best)) drives

external exploration. Without asking participants directly, it seems problematic to map specific computations to specific types of decision.

- The authors discuss the impact of uncertainty on exploratory decision making, but surprisingly little uncertainty is included in the models (with actually none for external exploration). Could the authors include some measure of uncertainty reduction / information gain in the models? The number of unknown options seems like a plausible heuristic for external exploration. For internal exploration, the best option is the value multiplied with the variance, which for me indicates a riskier choice rather than a large information gain. Again, what is actually gained in terms of such a decision? Risk? It seems like including measures of information gain that may drive internal and external exploration is highly warranted.

- Why is the accumulated cost function included in the decision model, rather than a 'constant' loss function? The accumulated cost is based on previous decisions, while the actual loss-to-be-incurred is what affects the current decision.

Version 1:

Reviewer comments:

Reviewer #1

(Remarks to the Author)

1. Summary of Revisions

The authors have made a real effort to respond to the comments, and the manuscript is improved in several places. The task itself is thoughtful and well-designed, and some of the new analyses — particularly the expanded behavioral modeling and clearer figure layouts (e.g., Figure 4c) — do help clarify parts of the paper.

That said, I still find the overall manuscript difficult to interpret. The analyses are complex, which is justified by the richness of the design, but the explanations often simplify things in a way that makes the results feel less meaningful or harder to trust. There's a lot of interesting data here, but the narrative doesn't fully do it justice. A more careful, precise framing would help make the findings clearer and more informative.

2. Major Concerns

ROI Definition

The manuscript still doesn't clearly explain how ROIs were defined — whether they were based on peak activation, anatomical labels, or previous studies. This is important for interpreting the regression and time-course analyses. Please explicitly state how each ROI was defined, and include justification where relevant.

Temporal Alignment and Reporting Consistency

While the authors did clarify some timing definitions in the rebuttal, I found the resulting analysis choices hard to follow — not because they're wrong, but because the framing doesn't help make sense of them. The inclusion of many different trial types and decision points leads to results that are difficult to interpret, especially when significance is reported inconsistently or when some effects are emphasized while others are downplayed. This makes it difficult to assess the implications of the time-course analyses. Clarifying these temporal definitions by, for example, making separate figures for whole-brain and ROIs analysis separate when possible for internal exploration, external exploration and accept decisions could help strongly with understanding and feels essential for evaluating the reported effects. It would also be helpful to include the average response time directly on the time-course figures, or to indicate the mean response time per decision type if they differ significantly. This would provide important context for interpreting neural dynamics and help clarify whether observed time-locked activity patterns align meaningfully with decision execution.

To improve clarity:

- Make sure all significant effects are marked consistently and clearly.
- Avoid downplaying or omitting non-significant or contradictory effects without explanation.
- add reaction time for each different decision type and maybe include a marker for the average RT those figures of time course analysis?

Figure S8 does a good job of laying out decision timing and would help with these issues — see below.

ACC and Unchosen Value

I greatly appreciate their answers about the ACC and conflict, and agree to their take on those results. However, I still believe it could potentially strengthen their result to see (and possibly demonstrate) that the ACC tracks when the second best options requires external exploration for example. This is just an idea but if the authors are defining the unchosen option as the second-best alternative, they could time-lock to the stimulus onset following a switch between chosen and unchosen, and look at ACC activity around that point. Breaking down these trials by what happens next (e.g., repeat explore, accept, or switch type) and showing the distribution across participants would make the analysis more informative and better suited to the task.

3. Figures and Supplementary Material

I appreciate the figures the authors did in answer to comment 18 (part 2 3 and 4); I don't think it would strengthen the manuscript to include it in the paper; however this did make me feel like internal exploration, external exploration and accept decision are maybe not as independent as the time series analysis makes it seem by comparing those trial types in the

same figure.

Figure S8 is one a very figure in the paper and should probably be moved to the main text. It gives a helpful overview of decision timing, but some aspects are still confusing:

- The inset panel isn't explained and doesn't include stats. If it highlights something specific, that should be stated clearly, with the appropriate statistical test.
- Significance stars need to be used consistently. A p-value of .134 isn't a trend. If trends are to be reported, please define what threshold you're using (e.g., $p < .06$) and use different symbols from actual significance markers; maybe also report the effect size to explain why although it is not significant you believe it is still important to report.

More broadly, the timing information in this figure could be used more in the main narrative to clarify when decisions tend to happen and how different decision types unfold.

4. Minor Clarifications

- In the behavioral analysis (Figure 2b), the sliding window method is now clearer, but it would help to state why this transformation was used and what it's meant to reveal.
- Please clarify what "trial 0" refers to.
- Specify which trial durations were included in whole-brain analyses (e.g., Figure 3).
- Avoid using "trend" for p-values $> .10$.
- Check that all figures referenced in the text (e.g., "middle panel of Figure 1b") are present and labeled.

5. Overall Assessment

This is a promising and ambitious study built around a rich experimental task. The data are interesting, and the authors clearly put in significant effort during revision. But I still find the current version difficult to follow, and I think the manuscript doesn't quite live up to the potential of the design. The presentation simplifies complex results and misses opportunities to clarify what different regions are actually doing during decision-making.

The findings aren't wrong — just not as compelling or interpretable as they could be. With clearer framing, more consistent reporting, and a few targeted improvements (like elevating S8 and rethinking some interpretations), this could become a solid and informative contribution.

I recommend major revisions, focused on clarity, interpretive depth, and doing justice to the complexity of the design.

Reviewer #2

(Remarks to the Author)

I sincerely thank the authors for their extensive work in replying to my concerns. I am satisfied with most responses, but still have some issues with the following replies:

Concerning major comment 1:

- The models used are very difficult to understand in terms of how many parameters are fitted in each model. Can the authors please provide a Table with all the models listed together with the number of parameters?
- Why was the cost only included in the full model? This still leaves the question whether a simpler model with the cost function (e.g. $Ac + cost$) provides a more parsimonious fit.

Concerning major comment 2:

- There are two identical models in the VBA analyses (EEV1, IEV1 and EEV5, IEV1), yet only one of them is considered the best. How is this possible?

Concerning major comment 6:

- The model descriptions are still not good enough. For example, in Equation 1 there are 10 beta values listed and I understand that each model uses a subset of them? Still, it would be really valuable to have a Table with each mode and its parameters (i.e. betas).

- Your multinomial approach to modeling behavior is highly interesting and other researchers may profit from understanding it better. Currently, your description of the modeling presumes that the reader is familiar with the approach, which most are not (I asked around). I would insist that you elaborate on the whole process, and describes it more clearly. For example, why does one decision need to be used as a 'reference'?

Concerning major comment 9:

- Using a contrast to identify brain regions has a big disadvantage in that a significant contrast $A > B$ may be due to either A being more activated than B, or because B is more de-activated than A. Because you look at correlations, it means that significant $A > B$ contrast may wrongly be interpreted as A showing a positive correlation with parameter X, while in reality it may be that it shows no correlation, and B instead shows a strong negative correlation with X. It is therefore critical to show that the correlation between A and X is also larger than zero.

Concerning major 10:

- The ANOVA is ok, but only shows that there are differences somewhere in your data. To find out where these differences are, you need to run post-hoc tests, which need to be corrected for multiple comparisons. This is a fundamental statistical approach which is highly relevant for your data, given all the uncontrolled comparisons made in Figure 5. Without the proper corrections, these results are not convincing at all.

Version 2:

Reviewer comments:

Reviewer #1

(Remarks to the Author)

I find the paper much better and clearer ! Thank you for thoroughly answering all of the comments with so much attention.

I have 3 final minor suggestion to help with clarity but do not require any further analysis or explanation:

1. you say the whole brain analysis was time locked at stimulus onset, given the complexity of task please specify which stimulus onset with more precision line 596: 599:
 - a. "time-locked to each decision phase stimulus onset (or offset following a internal exploration decision) " or "time-locked to stimulus onset at the beginning of each trial "
2. I still find figure 5 and its description disorganized/lacking in consistency about the results that you report (i.e. some significant results are not discussed).
 - a. Explaining the reasoning behind the trials selection in the ACC/exploratory signal analysis at the beginning of that paragraph would help with clarity. (Unless it was only done for the exploratory signal analysis in which case this should be made a lot more clearer and the negative internal exploration signal during an internal exploration decision should not be reported at the same time as the positive external exploration signal during an internal exploration decision as they are not done on the same trials) also being consistent in order in which you report really helps (internal, external accept. And consistently specify signal or decision (and not switching terms half way to accept value for example)
 - b. similarly, when reporting the mPFC effect specify the direction of the effect line 359 "and a positive accept signal during internal exploration decisions" and you need to discuss or explain why don't you mention the negative significant effect of internal exploration signal in the mPFC for external decision and accept decision ?
 - c. finally, stay consistent with the vocabulary that you use rather than use a new term (use accept signal or accept decision rather than accept "value"). "mPFC signalled external exploration value signal [...]" (Fig. 5c, middle panel; $\beta = 0.071$, $t_{23} = 2.872$, $p = 0.009$), but not accept value"
4. Supp Figure S9: please make sure that the * are not overlapping each other for visibility. If their size difference is not meaningful, please make them all the same size. if it is, please explain the difference.

Reviewer #2

(Remarks to the Author)

I sincerely thank the authors for their work on my behalf. Many concerns have been removed, except for a minor one and a (potentially) major one.

Minor 1:

Thank you for clarifying the modeling. However, now that I understand the procedure better, I realize that you are forcing the models to contain all your terms (i.e. each model contains IEV + EEV + ACCV terms), while less complex models (e.g. IEV only, EEV + ACCV, IEV + EEV, ...) are left out of the analyses and model comparisons.

While I would like to see an analysis of the full model space to make sure simpler models are not more parsimonious (which is usually conducted), this limitation should at the very least be mentioned in the manuscript.

Major 1:

Concerning the multiple comparisons issue, thank you for adding Supp. Fig 9.

However, I was mainly concerned about the 27 uncorrected tests reported in Fig. 5. I see only 7 tests with p-values < .01, which makes me suspicious of the correction method used. In my experience, using FDR-correction methods for so many tests (with low uncorrected significance) is unlikely to yield so many reported significant results.

Until FDR-correction has been applied to the tests reported in Fig. 5, those results are hard to believe.

Version 3:

Reviewer comments:

Reviewer #2

(Remarks to the Author)

i'm now satisfied. I recommend publication.

Responses to the reviews

Distinct contributions of prefrontal, parietal, and cingulate signals to exploratory decisions

Victor K. S. Chan, Nicole H. L. Wong, Tsz-Fung Woo, Kei Watanabe, Masahiko Haruno, Chun-Kit Law, Bolton K. H. Chau

Reviewer #1

In this study, the authors aim to identify different characteristics of exploratory decisions, proposing a way to bridge two seemingly opposing theories: that people either seek more information or reduce uncertainty during exploratory decision-making. The authors break down exploratory decisions into two subtypes: internal exploration (to reduce uncertainty about the outcome of the current option) and external exploration (to gather information about alternative options). They design a well-structured study to distinguish each type of exploration and compare these decisions to the choice of accepting an option based on its current expected reward.

In their literature review, the authors highlight studies demonstrating that intraparietal sulcus (IPS) activity correlates with information gathering to reduce uncertainty, anterior cingulate cortex (ACC) activity correlates with tracking alternatives, and ventromedial prefrontal cortex (vmPFC) activity relates to general decision value in value-based decision-making. They hypothesize that each of these regions plays a complementary role in exploratory decisions and test whether the IPS supports internal exploration, the ACC external exploration, and the medial prefrontal cortex (mPFC) decision-making—whether this involves exploring internally, exploring externally, or exploiting the current option.

Method

The authors ingeniously designed a task that distinguishes exploratory decisions into external and internal exploration, enabling comparisons between these three types of decisions. They provide multiple metrics to quantify each value type (value of external exploration, value of internal exploration, and value of accepting the current choice) and use the Bayesian Information Criterion (BIC) to select the best-fitting model. Using this model, they study whole-brain fMRI activity, focusing specifically on their hypothesized regions of interest.

Key Findings

The findings reveal specific encoding roles for each region:

- The IPS encodes internal decision values,
- The ACC encodes external decision values,

- The mPFC encodes internal, external, or accept decision values depending on subsequent participant choices. More specifically, the mPFC encodes the value of the chosen option regardless of the decision and the value of the best unchosen alternative when participants accept the current option.

The authors suggest a collaborative role among these regions:

- The IPS tracks uncertainty,
- The ACC identifies alternative needs,
- The mPFC processes decision-relevant information, supporting the neural common currency hypothesis, suggesting that the mPFC enables a decision regardless of the value of the choice itself.

My General Review:

Overall, I appreciate the authors' approach to this topic and their task design. However, I believe the paper could benefit from (1) clarifying certain concepts or findings, (2) providing more analysis to strengthen their arguments and meet the standards of Communications Biology, and (3) connecting their work more fully to the literature for greater impact. Missing out on these points would prevent the paper from reaching its full potential. My suggestions are as follows:

- Provide further analysis detailing participant behavior,
- Decode choices from neural activity,
- Clarify the role of the ACC with more reference to the literature,
- Specify the ACC's anatomical definition in their analysis, introduction, and discussion,
- Specify the mPFC's anatomy in their analysis, introduction, and discussion,
- Add the vmPFC to the regions of interest (ROIs) or explain why it is excluded,
- Include the frontopolar cortex (FPI) in ROI analysis, or explain why it is excluded.

What I Think is Cool About This Paper and Why:

Elegance of the Task Design

The experimental task is thoughtfully designed to isolate different types of decision-related information.

Aim to Characterize a Network with Regional Functional Differentiation

I appreciate the authors' attempt to characterize each brain region as part of a complementary network, linking seemingly contradictory literature. Accurate functional and/or anatomical characterization is essential for achieving this.

Response

We are deeply grateful for the constructive evaluation by the reviewer that recognized the novelty of our manuscript. In response to the reviewer's valuable comments for improving and clarifying our manuscript, we have prepared a detailed point-by-point response to each comment. In brief, first, we have now included comprehensive behavioral analyses that better characterize participants' decision patterns across different exploration types. Second, we performed additional analyses demonstrating that FPI did not play a critical role in our specific task context. Third, we demonstrated that the ACC specifically encoded external exploration values rather than general task difficulty. We hope these could adequately address the reviewer's concerns and strengthen the paper's scientific rigor.

Comment 1

Bayesian Model Selection

Using Bayesian model selection to identify the best-fitting decision-making model for each participant is an excellent approach. However, the authors should expand on this by explaining each metric's behavioral implications and characterizing behavior more thoroughly.

Response

Thank you for the comment. We have now included a detailed description about each metric's behavioral implications in our manuscript (Lines 555-560; For ease of reading, original text is shown in black font color, while revisions are shown in red font color):

where $Value_{Accept}$ was defined as the average point of the best option. $Value_{Internal\ exploration}$ was defined as the greatest product of the average point and the standard deviation among all revealed options, suggesting that internal exploration is focused on options that potentially lead to high gains but also contain great uncertainties. $Value_{External\ exploration}$ was defined as the average point of all hidden and revealed options, suggesting that exploring new options is more likely when the environment generally contains high gain options. The model arbitrates between the choices of accept, internal exploration and external exploration based on their corresponding value estimates, defined based on the best-fit model.

Comment 2

Relating Brain Activity to Behavior

The authors successfully demonstrate that brain activity explains specific choices. Further behavioral analysis and anatomically precise brain regions would strengthen this relationship.

Response

Thank you for the comment. To strengthen the relationship between brain activity and behavior, we conducted a multinomial logistic regression analysis using participants' IPS, ACC and mPFC BOLD signals at decision points as predictors to examine whether the neural activity could predict choices. Our analysis revealed distinct patterns of neural activity that selectively predicted specific decision types. The IPS activity showed a strong positive association with the internal exploration/accept choice ratio ($\beta = 0.666$, $t_{23} = 4.780$, $p < 0.001$), supporting its specialized role in internal exploration. The same analysis in ACC activity also demonstrated a robust positive correlation with the external exploration/accept choice ratio ($\beta = 0.666$, $t_{23} = 4.78$, $p < 0.001$), consistent with its hypothesized role in external exploration. The mPFC exhibited a broader influence, showing significant negative associations with both internal exploration/accept ($\beta = -0.814$, $t_{23} = -7.065$, $p < 0.001$) and external exploration/accept choice ratios ($\beta = -0.957$, $t_{23} = -6.918$, $p < 0.001$). These results provide strong evidence that neural activity in these regions can indeed predict participants' choices, with each region showing distinct patterns of predictive power. The findings largely support our hypotheses about the specialized roles of IPS, ACC and mPFC in internal exploration, external exploration and accept, respectively. Surprisingly, we observed an unexpected relationship, the internal exploration/accept choice ratio was positively associated with ACC activity ($\beta = 0.397$, $t_{23} = 3.394$, $p = 0.002$). This unexpected ACC-internal exploration relationship warrants further investigation.

A multinomial logistic regression showing the effects of IPS, ACC and mPFC activities on the ratios of internal exploration / accept decision (left panel) and external exploration / accept decision (right panel). ** denotes $p < 0.01$; *** denotes $p < 0.001$. Error bars represent \pm standard error of the mean (SEM).

Comment 3

Areas of Confusion

Role of the Lateral Frontal Pole (FPI)

The authors mention the FPI in the discussion and suggest that it interacts with the mPFC. They find a positive correlation between FPI activity and internal exploration in the whole-brain analysis and suggest that future research could investigate this further. However, it's unclear why they did not include the FPI in the network they study.

Response

Thank you for the comment. To test the role of FPI during explorations, we conducted additional analyses on the FPI using a 3mm radius sphere centered at coordinates derived from Law et al. (2023). Our time-course analysis revealed significant positive external exploration signals in the FPI at multiple time points (middle panel; peaks observed at -3.06s: $t_{23} = 2.483$, $p = 0.021$; 1.53s: $t_{23} = 2.703$, $p = 0.013$; and 7.14s: $t_{23} = 2.894$, $p = 0.008$). We then extracted peak signals for each participant and examined their correlations with the proportions of internal exploration, external exploration, and accept decisions made during the task. However, none of these correlations were significant (right panel; internal exploration: $r = -0.255$, $p = 0.266$; external exploration: $r = 0.221$, $p = 0.335$; accept decisions: $r = 0.258$, $p = 0.259$). Given the lack of strong evidence for a clear functional role of the FPI in our task, we refrained from focusing too much on this region. We acknowledge this region could be an interesting target for future research and we have included these results in Supplementary Fig S10. We also revised our manuscript (Lines 464-466):

This is particularly interesting because exploratory behaviours in humans are not only related to the ACC, but additionally involves the lateral frontopolar cortex (FPI)(Boorman et al., 2009; Hogeveen et al., 2022). Neuroanatomical studies showed that the FPI is greatly expanded in the human brain and a homolog is arguably absent in the macaque brain(Neubert et al., 2014; Tsujimoto et al., 2011), however, the function of this additional activity in the FPI is broadly unclear. Recently, Law and colleagues suggested that the FPI is involved in decomposing complex decision information and such ability may be particularly useful during exploration (Law et al., 2023). **Our preliminary analysis on FPI also showed that it encoded the external exploration value, providing additional insights into its functional role during exploration decisions (see Supplementary Fig. S10).** Future work may also address the unique functions of ACC and FPI for guiding exploratory decisions in humans.

Supplementary Figure. S10. Signals in the FPI. The coordinates were taken from Law et al. (2023) (left panel). A time course analysis showing a significant signal in the FPI related to external exploration, but it was unrelated to internal exploration and accept values (middle panel). There was no significant correlation between the peak signal of each participant and their proportions of internal exploration, external exploration, and accept decision made (right panel). * denotes $p < 0.05$; ** denotes $p < 0.01$. Shaded areas represent \pm SEM.

Comment 4

ACC's Role in Conflict/Control

The authors mention the ACC's involvement in tracking alternatives and future goals in the introduction and discussion, but they do not address its roles in conflict, control and difficulty (for example). Clarifying how their study fits into this literature would be beneficial.

Response

Thank you for the comment. We conducted a time-course analysis to test the relationship between ACC activity and task difficulty. Here, we defined task difficulty as the value of the second-best choice minus the value of the best choice. However, the result did not reveal any significant signal related to these processes ($t_{23} < -0.98$, $ps > 0.115$). Given the lack of clear evidence that the ACC's activity was related to task difficulty, we chose to maintain a focused narrative on our initial findings.

Comment 5

mPFC or vmPFC

It is unclear why the mPFC was chosen over the vmPFC, given their introduction and discussion concerns the vmPFC mostly.

Response

Thank you for the comment. Our intention was to examine the ventromedial prefrontal cortex (vmPFC) as it is widely studied in the context of decision making. However, the results of our whole-brain analysis (Fig. 3) showed a cluster with the ventral side overlapping with the traditionally known vmPFC, but extends dorsally to the pregenual cingulate area. Hence, we refrained from using the more specific vmPFC label and used a broader label, which is mPFC.

Comment 6

Interpretation of Temporal Deconstruction in fMRI

In Figure 4B, the onset of external exploration signals in the ACC appears to start before stimulus onset, raising questions about the interpretation.

Response

Thank you for the comment. The analysis in Fig. 4b focused on the fMRI data time-locked to stimulus onset. Each trial could have multiple scenarios of stimulus onset – at the start of each trial or after each exploration when the set of options were presented again. Since the value of external exploration remained unchanged throughout each trial, this information about external exploration would be known to participants during the interstimulus intervals prior to the stimulus onset. Hence, this signal in the ACC was possible to appear before the start of stimulus onset.

To avoid confusion and clarify the definition of stimulus onset, we have now revised the manuscript (Lines 280-281):

... locked to the onset of the decision phase (solid black line), **which could occur at the start of each trial or after each internal/external exploration.**

Comment 7

Model Comparison (Figure S2)

The supplementary material could better clarify the metrics compared in the correlation and whether the third-best fit represents the average best fit for each subject. If so it would be informative to have a distribution of which models are the best fit for each subject.

Response

Thank you for the comment. We have added the equations in Fig. S2a, and hope it helps clarify how the value is defined.

Revised Fig. S2:

We also conducted additional analysis showing the distribution of the best-fit model for each participant. The best-fit model at the group level was also the most frequent best-fit model at the participant level. We included a description in our manuscript (Lines 146-147):

...The robustness of our model selection was further validated through Bayesian model selection analysis, which showed that our best-fit model had the highest exceedance probability among all candidate models (Supplementary Fig. S2). **Additionally, at the**

individual level, this model best described the greatest proportion of participants' behaviors (20.83% of participants; Supplementary Fig. S3). To ensure the reliability of our model fitting procedure, we conducted parameter recovery analyses, which demonstrated that all ten parameters could be successfully recovered with high precision (all correlations: $r_s > 0.801$, $p_s < 0.001$; Supplementary Fig. S4).

Supplementary Figure. S3. The distribution of best-fit model at participant level. In this analysis, we counted the percentage of participants that was best described by each model. The results showed that the best-fit model at the group level (labelled by a green box) was also the model that best describes the greatest proportion of participants' behaviour.

Comment 8

Chosen/Unchosen Analysis

In Figure 6, what does it mean that the mPFC encodes the value for choosing internal exploration before stimulus onset? Given that the ACC tracks external exploration and there interpretation about alternatives, why did they not look at unchosen options in the ACC, as they did in the mPFC?

Response

Thank you for the comment. We apologize that the labelling in Fig. 6 was misleading. We mistakenly left out labelling the second peak of the mPFC signal during internal exploration, that is more relevant. We have now revised Fig. 6 in the manuscript:

Our decision not to pursue this analysis in the ACC was guided by our findings regarding its functional characteristics. Specifically, as shown in Fig. 5, the ACC signals were not flexible to the context, as they consistently encoded external exploration value regardless of the subsequent choice type. The analysis in Fig. 6 was specifically designed to test the mPFC signal as it showed flexibility in its coding according to the decision context.

Comment 9

Main paper p.4 line 75 “[...] These findings were intriguing because it provides an alternative account of the widely reported role of IPS in signalling reward.” It would be nice for them to make explicit which widely reported role and provide references about “the widely reported role of IPS in signaling reward” – does the IPS track the value of reward in 1D, does it respond in an acute way, does it track the cumulative reward if only on 1D, is it always the same dimension? Which widely reported role are they talking about, and which studies say so (doesn’t have to be an exhaustive list).

Response

Thank you for the comment. Here, we refrain from commenting specifically the aspects of reward signalling that are involved in the IPS, because this could be another long debate and is beyond the scope of the current study. However, there has been a great volume of studies suggesting the IPS is involved in reward processing and evidence accumulation during decision making. We have made the revision and references are now included in the manuscript. (Lines 67-69):

...These findings were intriguing because they provide an alternative account of the widely reported role of IPS in signalling reward, such as in the process of evidence accumulation

(Kahnt et al., 2014; Louie & Glimcher, 2010; Platt & Glimcher, 1999; Shadlen & Newsome, 1996; Sugrue et al., 2004).

References:

Kahnt, T., Park, S. Q., Haynes, J. D. & Tobler, P. N. Disentangling neural representations of value and salience in the human brain. *Proc Natl Acad Sci U S A* 111, 5000–5005 (2014).

Louie, K. & Glimcher, P. W. Separating value from choice: Delay discounting activity in the lateral intraparietal area. *Journal of Neuroscience* 30, 5498–5507 (2010).

Platt, M. L. & Glimcher, P. W. Neural correlates of decision variables in parietal cortex. *Nature* 400, 233–238 (1999).

Shadlen, M. N. & Newsome, W. T. Motion Perception: Seeing and Deciding (Motion Perceptionpsychophysicsdecision Makingparietal Cortex). vol. 93 (1996).

Sugrue, L. P., Corrado, G. S. & Newsome, W. T. Matching Behavior and the Representation of Value in the Parietal Cortex. *Science* (1979) 304, 1782 (2004).

Comment 10

Main paper p.19 line 371 to 374: “Taken together, these results support the influential neural common currency hypothesis suggesting that a single brain region, often considered to be the mPFC or the orbitofrontal cortex, can signal general utility of any kinds of item.” I don’t understand what they mean by “often considered to be OFC or mPFC”; they do not look at the OFC nor mention it; they do not reference this sentence.

Response

Thank you for the comment. The neural common currency hypothesis has been extensively discussed in the literature, where researchers have historically used the terms ventromedial prefrontal cortex (vmPFC) and orbitofrontal cortex (OFC) somewhat interchangeably due to their anatomical proximity and functional overlap. We have now also referenced this sentence in our manuscript (Line 396):

... these results support the influential neural common currency hypothesis suggesting that a single brain region, often considered to be the mPFC or the orbitofrontal cortex (Bartra et al., 2013; Levy & Glimcher, 2012).

References:

Bartra, O., McGuire, J. T. & Kable, J. W. The valuation system: A coordinate-based meta-analysis of BOLD fMRI experiments examining neural correlates of subjective value. *Neuroimage* 76, 412–427 (2013).

Comment 11

In Sup mat, p.3: “The first type of the external exploration value was formulated as the average point of all hidden and revealed options (*ValueExternal Exploration(1)*). The second type of the external exploration value was formulated as the average point amongst all revealed options (*ValueExternal Exploration(2)*).” In the formula, (*ValueExternal Exploration(1)*) = Points(all options) and (*ValueExternal Exploration(2)*) = Points(all existing options). It sounds like "All existing options" means all options including the hidden ones. If that is correct, there is a mistake in the indexing, where (*ValueExternal Exploration(1)*) is the average point of all hidden and revealed options and therefore = Points(all existing options). If my understanding is incorrect, this should be clarified. It also asks the question of how participants can estimate the value of hidden possibilities?

Response

Thank you for the comment. The reviewer’s interpretation is correct – “all existing option” is better described as “all revealed options” to avoid confusion. We have now revised the supplementary information and hope this could improve the clarity. (Lines 45; 47; 51; 58):

$$Value_{External\ exploration(2)} = \overline{Point}_{(all\ revealed\ options)}$$

$$Value_{External\ exploration(4)} = \overline{Point}_{(all\ options)} - \overline{Point}_{(all\ revealed\ options)}$$

... value was formulated as the average point amongst all **revealed** options.

... the point difference between **revealed** options and the environment might lead to an external exploration decision.

Comment 12

Figure 1B: The proportion of accept, internal and external: explicitly say if this is the proportion of those decisions as a first decision on every trial. I understand Figure 1B second and third rows to be respectively the proportion of internal and external decisions for each number of decisions made before an accept decision. If so, another way to plot this in a more informative way could be to stack on top of each other the proportion of internal, external and accept choices as a function of number of decisions before reaching accept and finishing the trial.

Response

Thank you for the comments and suggestions. In our experiment, each trial always involved one accept choice, because each trial was terminated whenever an accept choice

was made. The purpose of Fig. 1b was to show how many internal exploration and external explorations were made before a trial was terminated (accept choice made). Perhaps one way of better showing that is to combine the second and third rows, as the reviewer implied, and plot them as clustered bars. We have now revised Fig. 1b in the manuscript and hope this could help improving the clarity.

Revised Fig. 1b. Task performances. The proportion of accept, internal exploration and external exploration made by participants throughout the task (top panel). **The proportion of trials with different total number of internal exploration and external explorations before an accept choice was made (bottom panel).**

Comment 13

Figure 2B Clarity: Could do with the equation on the figure for each measure of internal, external and accept value. Also, it would be nice to clarify how the authors binned the trial to analyse 3 decision option into a binary way what is being compared (choice proportion done in a binary way).

Response

Thank you for the comment. We have revised Fig. 2b by adding the equation for each measure of internal, external and accept values.

We have also included a description in our manuscript to clarify how we binned the trial to analyse 3 decision option into a binary way (Lines 186-190):

“A participant-specific analysis was conducted by ranking internal exploration, external exploration, and accept values into 30 probability classes, which were then consolidated into 20 levels using an 11-rank sliding window approach. The choice ratios between decision pairs were analyzed across these levels, excluding trials involving the third decision type. The final results represented the averaged behavioral patterns across all participants.”

Comment 14

Colour coding in the paper (Figure 4 and 5) throughout the paper, colour is confusing as red means IPS signal but also means internal exploration, blue ACC but also external, and green means mPFC signal but also accept. Although the reasoning for making that association makes sense, it limits / hinders the clarity of what is tested on each graph (is what is plotted the behaviour or the brain activity?). If the results are compelling enough, the colour coding doesn't need to force interpretation but guide it.

Response

Thank you for the comment. We have now revised the color of our ROIs to yellow in Fig. 4. We hope this could clear up the confusion. Revised Fig.4:

Comment 15

stimulus onset timing and decision distribution within time window: There's ambiguity regarding which stimulus onset they focus on—whether it's the initial stimulus onset (i.e. the beginning of the trial) or the latest stimulus change in each response. If it's the second answer, then doesn't this mean that there are different number of trials and/or timing of the trial for each decision type? This raises questions about the nature of the signal that the authors are comparing – it would be helpful to know their perspective on this rather than have to infer it. They mention that for some analysis they removed the first trial/decision, does that mean that they included all the other decisions regardless of the distributions of decisions that were made at each decision number?

Response

Thank you for raising this important point about stimulus onset timing and decision distributions. For the main analyses (all figures except Fig. S5 & S6), we deliberately included stimulus onsets both at trial initiation and after each exploration when options were re-presented. This enable us to analyse signals that were elicited at the time when any choice was made. We have now explicitly stated this in all relevant figure legends to prevent any ambiguity. The supplementary analyses (Fig. S5 & S6) had specific analytical goals that required different approaches. For Fig. S5, we focused on internal exploration decisions, excluding the first instance when participants explored an option internally. This decision was driven by our observation that participants often made repeated internal exploration decisions for the same option, and we hypothesized that ACC signals would be more pronounced in subsequent explorations. For Fig. S6, as we specifically investigated information gain encoding in the IPS following internal exploration, we analyzed only the periods after internal exploration events, with stimulus onset defined as the moment when updated option information was presented. These methodological choices were guided by our specific hypotheses about the roles of different brain regions in the decision-making process.

Comment 16

figure 4C : i like this analysis and understand it as a demonstration of how neural activity reflects the behaviour of each participant. However, the individual variability title through me off because it sounds like they will now look at individual differences in this study – do they from then on use the best fit for each participant instead of the best fit model? What do they want to demonstrate? if they want to investigate the neural correlates that support the decisions of participant then I feel the should clarify in text/legend and also either by changing the title or removing it all together for consistency given that other figure do not have one. If that is not it, I feel it would be helpful to make to goal clear and add why that is important/relevant to the paper.

Response

Thank you for the comment. We agree that the term “individual variability” could be ambiguous when it was not explained clearly. Here, we aimed at showing that (1) individuals were variable in terms of their tendency of making each type of decision and (2) that such variability was related to the strength of the corresponding neural signal. We have now revised the manuscript and the figure to avoid confusion (Lines 234-236):

...Next, **we further validated whether the observed brain activity was functionally relevant to the decisions made.** We tested the relationship between individual variabilities **in the proportion** of the internal exploration, external exploration and accept signals and the proportion of their corresponding choice in a between-participant analysis.

Revised Fig. 4c:

Comment 17

Figure 5 Clarity: The hollow bars' meaning is unclear to me—it says that they indicate non-significance or negativity. However, some bars appear negative, and some include significance stars.

Response

Thank you for your comments. In this analysis, our objective was to examine the roles of the IPS, ACC, and mPFC in internal exploration, external exploration, and accept decisions, where the observed signals were expected to demonstrate both positivity and significance. However, in some cases the signals were both negative and significant. We opt to keep the star in order to be transparent in reporting. However, in view of potential confusions, we have revised the legend (Line XXX):

Signals that were not significant or **significantly** negative are presented as hollow bars.

Comment 18 (part 1)

Behavioural Analysis:

To better understand the figures and explore what this task can tell us about exploration, it would be beneficial to conduct further analysis on participant behaviour. This could include (but these are just suggestions) examining:

The staked bar analysis suggested previously here would give an understanding of the distribution of choices across trials and author could use this to illustrate which decisions they used to compare the different type of signals (first or second decision)

Response

Thank you for your comments. Perhaps one way of better showing that is to combine the second and third rows, as the reviewer implied, and plot them as clustered bars. We have now revised Fig. 1b in the manuscript and hope this could help improving the clarity (Lines 136-137).

Revised Fig. 1b. Task performances. The proportion of accept, internal exploration and external exploration made by participants throughout the task (top panel). **The proportion of trials with different total number of internal exploration and external exploration (bottom panel).**

Comment 18 (part 2)

Average points participants obtained on a trial,

The minimum expected value at which participants make an “accept” choice. If Figure 2B correctly indicates this, it would be helpful to have the authors explicitly describe their interpretation in relation to this figure.

Response

Thank you for the comment. We conducted additional analyses on the average points participants obtained on a trial ($M = 13.700$, $SD = .075$), and the minimum expected value at which participants make an “accept” choice ($M = 8.219$, $SD = .041$). We could include this in the supplementary information if the reviewer recommends this would strengthen the manuscript.

Comment 18 (part 3)

Patterns of alternation between internal and external exploration before accepting (e.g., the proportions of “In-Ex-Ac,” “Ex-In-Ac,” “In-In-Ex-Ex-Ac,” where “In” represents Internal Exploration, “Ex” represents External Exploration, and “Ac” represents Accept decisions).

Response

Thank you for your comments. Our analysis revealed 218 distinct patterns of alternation between internal and external exploration phases before accepting. While the majority of these patterns only occurred once, we present here the 20 most frequently observed sequences. We could include this in the supplementary information if the reviewer recommends this would strengthen the manuscript.

The proportions of the patterns of alternation between internal and external exploration before accept decisions were made. The 20 most frequently observed sequences are shown.

Comment 18 (part 4)

The average expected value at which participants switch to accepting after beginning external exploration, and possibly after starting internal exploration if there is no constraint on the order of decisions. This could reveal whether participants' exploration is affected by the known range or distribution of outcomes. If participants are unaware of this range and mean, exploring may make them more likely to revert to previous options; however, if they know the range and mean, their decisions could reflect an ability to compare current options with unknown alternatives.

Response

Thank you for the comment. To test whether participants were aware of the known range or distribution of outcomes, and capable of making comparison between current options with unknown alternatives before accepting an option, we conducted additional analyses that considered the critical decision points (expected value thresholds) where participants switched between accepting an option and explorations compared to the current external exploration value. Our findings reveal that participants switched from external exploration to accepting decisions at an average value of 14.570 (SD = .474), which was significantly higher than the known distribution of external exploration value (M = 13.615, SD = .139; $t_{23} = 2.584, p = .017$).

Similarly, in our second analysis, the average switching point from internal exploration to accepting decisions ($M = 16.182$, $SD = .527$) was significantly higher than the known distribution of external exploration values ($M = 13.347$, $SD = .213$; $t_{23} = 5.663$, $p < .001$). These results suggest that participants were capable of making accept decisions at values higher than the known distribution of outcomes consistently, rather than making decisions in isolation. We would like to include this in the supplementary information if the reviewer recommends this would strengthen the manuscript.

Comment 19

Inclusion of the frontopolar cortex:

The frontopolar cortex (FPI) seems relevant to internal exploration (as indicated by their analysis) and to tracking alternatives. Studies by Badre et al. (2012), Boorman et al. (2011), Donoso et al. (2014), Tomov et al. (2020), Yoshida & Ishii (2006), and Zajkowski et al. (2017) emphasize the FPI's involvement in these processes. Additionally, several studies have demonstrated the role of the FPI in exploration in non-human primates (Boschin et al., 2015; Mansouri et al., 2015; Tsujimoto et al., 2010, 2012). It would enhance the paper to either include FPI in the ROI analysis or justify its exclusion with a scientific rationale.

Response

Thank you for the comment. We conducted analyses on the FPI (3mm radius sphere, coordinates from Law et al., 2023) and found significant positive external exploration signals at multiple timepoints (middle panel; peaks observed at -3.06s: $t_{23} = 2.483$, $p = 0.021$; 1.53s: $t_{23} = 2.703$, $p = 0.013$; and 7.14s: $t_{23} = 2.894$, $p = 0.008$). However, subsequent correlation analyses between peak signals and behavioral measures (internal exploration, external exploration, and accept decisions) yielded no significant relationships (all $ps > 0.25$). Given the lack of strong evidence for a clear functional role of the FPI in our task, we opted to include these analyses in the supplementary materials rather than in our primary analyses. We acknowledge these findings suggest potential investigations into the FPI's role in the future.

Supplementary Figure. S10. Signals in the FPI. The coordinates were taken from Law et al. (2023) (left panel). A time course analysis showing a significant signal in the FPI related to external exploration, but it was unrelated to internal exploration and accept values (middle panel). There was no significant correlation between the peak signal of each participant and their proportions of internal exploration, external exploration, and accept decision made (right panel). * denotes $p < 0.05$; ** denotes $p < 0.01$. Shaded areas represent \pm SEM.

Comment 20 (part 1)

Further characterization of the ACC

- Conflict/Control Framework and More

There are many different perspectives on the function of the ACC, particularly regarding its subdivisions (Clairs & Lopez-Persem, 2023; Vassena et al., 2017). It would be valuable to see how the authors' results align with existing literature on control and conflict (Botvinick et al., 2001, 2004, Shenhav et al. 2016) and Kolling et al. 2016). For example, if the values of internal, external, and accept choices are close, this could lead to higher conflict or difficulty in making a choice. Examining ACC activity on these trials would acknowledge the ACC's established role in conflict or difficulty and its relation to task-switching.

Response

Thank you for the comment. We specifically examine the relationship between ACC activity and task difficulty through time-course analysis. We defined task difficulty as the value of the second-best choice minus the value of the best choice. However, our analysis did not reveal significant task difficulty related signal ($t_{23} < -0.98$, $ps > 0.115$). While we focused our manuscript on our primary findings, we would like to expand our discussion to contextualize our results including task difficulty if the reviewer recommends this would strengthen the manuscript.

Comment 20 (part 2)

Functional Anatomy of the ACC

Clarifying which part of the ACC is under examination and the rationale behind it would strengthen the paper, especially given the anatomical and functional subdivisions highlighted in the introduction. If the authors have chosen a spherical ROI, discussing the implications of this choice—such as the potential averaging effect across opposing

activities in different subregions—would provide insight into the results. Are there specific subregions they aim to measure, and how might this choice affect the findings?

Response

Thank you for the comment. We have now revised our manuscript to provide more descriptions of the choice of the ACC ROI (Lines 203-206):

Second, we identified the sulcus of the ACC encoded an external exploration signal to track the overall value in the environment (Fig. 3, middle panel; cluster-based threshold $z > 3.10$, $p < 0.05$), which is similar to previous reports suggesting the ACC sulcus reflects the recent reward history or the environment's overall reward value (Behrens et al., 2008; Kolling et al., 2012, 2018). We notice that the cluster had an elongated shape that lies along the cingulate sulcus. In subsequent analyses, we focused on the anterior part of the cluster, which is most related to reward processing in the cingulate cortex (Cluster 3 of the cingulate parcellation map by Beckmann et al. (Beckmann et al., 2009)).

Comment 21

Predictive ROI Analysis to Strengthen Their Demonstration:

It would be interesting to see if ACC, IPS, and mPFC activity can independently predict a participant's choice and if the ACC and IPS could do so above and beyond any other information than the external exploration and internal exploration respectively?

For example: If their hypothesis is correct, IPS might predict internal exploration choices but no others decisions, ACC might best predict external exploration but not others, and mPFC might predict all choices.

Response

Thank you for the comment. Our multinomial logistic regression analysis investigated how BOLD signals from three brain regions (IPS, ACC, and mPFC) at decision points predicted participants' choice behavior. The results revealed region-specific predictive patterns. The IPS activity showed a strong positive association with the internal exploration/accept choice ratio ($\beta = 0.666$, $t_{23} = 4.780$, $p < 0.001$). The ACC activity also showed a positive correlation with the external exploration/accept choice ratio ($\beta = 0.666$, $t_{23} = 4.78$, $p < 0.001$). While the mPFC showed significant negative associations with both internal exploration/accept ($\beta = -0.814$, $t_{23} = -7.065$, $p < 0.001$) and external exploration/accept choice ratios ($\beta = -0.957$, $t_{23} = -6.918$, $p < 0.001$). These findings provide compelling evidence for neural predictors of choice behavior, with distinct regional specificity. The results largely confirm our hypotheses regarding IPS, ACC and mPFC's specialized roles in internal exploration, external exploration and accept, respectively. Surprisingly, there was an unexpected relationship between ACC activity and the internal exploration/accept choice ratio ($\beta = 0.397$, $t_{23} = 3.394$, $p = 0.002$). This

unexpected positive association merits further investigation, potentially indicating more complex neural interactions during decision-making.

A multinomial logistic regression showing the effects of IPS, ACC and mPFC activities on the ratios of internal exploration / accept decision (left panel) and external exploration / accept decision (right panel). ** denotes $p < 0.01$; *** denotes $p < 0.001$. Error bars represent \pm standard error of the mean (SEM).

Comment 22

Interaction Between Frontal Pole & mPFC ; and between Frontal pole & ACC:

Testing their suggested interaction between the frontal pole and mPFC, and between the ACC and frontal pole, might reveal distinct functional roles. Given their describe role of the ACC in planning future choices and selecting the best alternative for the future, I would also discuss the frontal pole in predicting future decisions and tracking alternatives and verify their respective roles here

Response

Thank you for the comment. We conducted psychophysiological interaction analyses (PPI) to test the functional interactions between FPI and mPFC, as well as between FPI and ACC. Our findings revealed a selective pattern of interaction, particularly between FPI and mPFC. Specifically, we observed significant modulatory effects of FPI on the internal exploration signal (left panel: peak at 9.28s: $t_{23} = -2.084$, $p = 0.049$) during internal exploration decision. However, this modulatory relationship did not extend to external exploration signal during external exploration decision (middle panel: $t_{23} =$

1.654, $p = 0.112$) or accept signal during accept decisions (right panel: $t_{23} = -2.038$, $p = 0.053$), suggesting a context-specific functional coupling between FPI and mPFC that is primarily engaged during internal exploration decision.

On the other hand, we did not observe a significant modulatory effect of FPI on the external exploration signal in the ACC across the entire task ($t_{23} = 1.641$, $p = 0.114$). The hypothesized interaction between FPI and ACC may require further investigation to fully understand their relationship during exploration.

Reviewer #2

The authors present a new behavioral paradigm in which participants can explore the environment (adds a new option with four gambles), explore an existing option (decreases the number of gambles within one option), or accept an existing option (one gamble within the selected option is played). Participants perform the task during fMRI scanning, and model-based fMRI is used for the analysis.

The main findings claim that the computations contributing to the three types of decisions (internal- or external- exploration, or accept) are represented in an ‘... appealing one-to-one matching between brain regions and value signal types.’ As such, the IPS, ACC, and the mPFC respectively encode internal, external, and accept values independently of decision-type. By contrast, the authors also report that the mPFC encodes all value-types based on the impending decision.

I found the Abstract promising and intriguing, and the task novel and interesting, but I’m not convinced by the results. In particular, the modeling section needs to be much improved and the fMRI analysis suffers from multiple comparison issues. My suggestions are presented below.

Response

We are deeply grateful for the constructive evaluation by the reviewer that recognized the novelty of our manuscript. In response to the reviewer's valuable comments for improving and clarifying our manuscript, we have prepared a detailed point-by-point response to each comment. In brief, first, we have included additional model comparisons demonstrating the robustness of our winning model compared to more parsimonious alternatives. Second, we have performed parameter recovery analyses validating the reliability and identifiability of our model parameters. Third, we have conducted a three-way ANOVA, revealing a significant interaction between brain region, decision type, and signal type, which substantiates our claims about the unique role of brain regions in specific exploratory decision processes. We hope these additional analyses adequately support our model selection and theoretical framework, and help address the reviewer's concerns.

Major comment 1

The authors compare 90 different models, each with a different combination of computations believed to underpin the different decisions. Getting the model right is crucial for the interpretation of both behavior and fMRI data.

- If I understand correctly, all tested models have 10 parameters (lines 516-517). However, simpler models may provide more parsimonious fits to behavior. For example, in theory a model with only an Accept value could provide the most parsimonious fit to behavior (exploration rate would then be determined by the inverse temperature parameter).

Accordingly, please also include all nested models in your model comparisons (i.e. models with two and one parameter(s) only).

Response

Thank you for the comment. We have now considered more parsimonious versions with fewer parameters. Our results showed that there were significant differences among models' BIC scores ($F(7,184) = 12.812, p < .001$) and prediction accuracy ($F(7,184) = 44.337, p < .001$). Importantly, our winning model (the red bars) provides better descriptions of participants' choices when compared to those alternative models in terms of BIC score (all $t_{s23} < -4.029, p_s < .001$) and prediction accuracy (all $t_{s23} > 4.053, p_s < .001$). These results support our model selection, indicating that despite its higher complexity, our winning model provides the best account of the behavioral data while justifying the inclusion of additional parameters. For parameter labels, where "In" represents Internal exploration value, "Ex" represents External exploration value, and "Ac" represents Accept value.

Major comment 2

Using a fixed-effect analysis for model comparison (e.g. summed BIC) is now quite frowned upon, and I would strongly recommend using a different approach, such as calculating exceedance probabilities (Rigoux et al., 2014, Bayesian model selection for group studies - revisited).

Response

Thank you for the comment. We performed Bayesian model selection using Variational Bayesian Analysis (VBA). This analysis corroborated our initial findings, with the same

model emerging as the best-fit, thus strengthening our conclusions through converging evidence from multiple analytical approaches. We have now included this in supplementary Fig S2 and added a sentence in our manuscript.

Supplementary Figure S2. Bayesian model selection results. (a) General Linear Model. (b) Variational Bayesian Analysis (VBA). Exceedance probabilities for each candidate model, representing the probability that a given model is more frequent than all other models in the population. The same model showed the highest exceedance probability (the Best-fit model). This analysis corroborated our initial findings, with the same model emerging as the best-fit, thus strengthening our conclusions through converging evidence from multiple analytical approaches.

Major comment 3

I sincerely appreciate the robustness analysis replicating findings using second-best models et cetera, but please add a table with log-likelihoods and BICs to make the comparison between models easier.

Response

Thank you for the comment. We have now included a table showing the log-likelihoods and BICs of the models (Supplementary Table S1):

Supplementary Table S1 – Model fit in BIC and Log-likelihood								
Internal exploration value (type 1)								
External exploration value (type)	Accept value (type)							
	BIC – Mean (SD)				Log-likelihood – Mean (SD)			
	1	2	3	4	1	2	3	4
1	459.662 (166.695)	618.582 (211.710)	605.671 (212.056)	569.832 (172.080)	-216.447 (83.158)	-295.907 (105.733)	-289.452 (105.817)	-271.532 (85.905)
2	516.638 (188.894)	490.787 (185.500)	478.409 (176.202)	453.255 (153.666)	-244.935 (94.212)	-232.010 (92.685)	-225.821 (88.001)	-213.244 (76.702)
3	498.729 (174.845)	451.874 (154.413)	442.471 (145.270)	418.800 (126.043)	-235.981 (87.228)	-212.553 (77.171)	-207.852 (72.568)	-222.279 (87.177)
4	471.325 (174.693)	498.220 (181.218)	490.181 (180.177)	455.644 (144.362)	-222.279 (87.177)	-235.726 (90.512)	-231.707 (89.903)	-214.438 (72.049)
5	459.662 (166.695)	438.442 (141.991)	434.657 (137.139)	397.573 (105.902)	-216.447 (13.158)	-205.837 (70.977)	-203.945 (68.447)	-185.403 (52.927)
Internal exploration value (type 2)								
External exploration value (type)	Accept value (type)							

	BIC – Mean (SD)				Log-likelihood – Mean (SD)			
	1	2	3	4	1	2	3	4
1	448.731 (165.276)	647.370 (243.931)	663.797 (253.903)	566.707 (172.783)	-210.982 (82.473)	-310.301 (121.759)	-318.515 (126.668)	-269.970 (94.709)
2	510.222 (189.860)	517.826 (205.147)	525.274 (206.489)	454.319 (153.095)	-241.728 (94.708)	-245.529 (102.411)	-249.254 (103.027)	-213.776 (76.422)
3	494.637 (175.907)	474.250 (165.097)	481.174 (163.071)	421.210 (125.839)	-233.935 (87.762)	-223.741 (82.414)	-227.203 (81.333)	-197.221 (62.816)
4	461.594 (174.728)	519.604 (201.600)	529.567 (205.753)	453.124 (143.826)	-217.414 (87.205)	-246.418 (100.612)	-251.400 (102.628)	-213.178 (71.806)
5	448.731 (165.276)	454.211 (150.244)	463.788 (148.809)	397.212 (106.329)	-210.982 (82.473)	-213.722 (75.018)	-218.511 (74.210)	-185.223 (53.169)

Internal exploration value (type 3)

**External
exploration
value (type)**

Accept value (type)

	BIC – Mean (SD)				Log-likelihood – Mean (SD)			
	1	2	3	4	1	2	3	4
1	385.347 (123.224)	586.623 (184.131)	572.421 (185.298)	560.925 (167.231)	-179.290 (61.616)	-279.928 (92.008)	-272.827 (92.555)	-267.079 (83.530)
2	439.178 (146.272)	463.989 (167.175)	463.980 (166.018)	445.129 (150.499)	-206.206 (73.065)	-218.611 (83.534)	-218.606 (82.939)	-209.181 (75.160)
3	427.210 (136.486)	424.416 (136.249)	425.476 (135.469)	410.223 (123.933)	-200.221 (68.209)	-198.824 (68.109)	-199.354 (67.704)	-191.728 (61.911)
4	397.560 (134.170)	466.384 (151.914)	457.410 (150.355)	444.467 (136.413)	-185.396 (67.076)	-219.809 (75.936)	-215.322 (65.139)	-208.850 (68.140)
5	385.347 (123.224)	404.158 (114.831)	401.566 (113.545)	388.064 (101.673)	-179.290 (61.616)	-188.696 (57.507)	-187.399 (56.824)	-180.648 (50.876)

Internal exploration value (type 4)

**External
exploration
value (type)****Accept value (type)**

	BIC – Mean (SD)				Log-likelihood – Mean (SD)			
	1	2	3	4	1	2	3	4
1	369.365 (110.240)	563.306 (165.495)	541.801 (167.449)	497.177 (155.561)	-171.299 (55.154)	-268.270 (82.741)	-257.517 (86.680)	-235.205 (77.773)
2	427.423 (137.051)	454.912 (156.952)	454.813 (156.219)	447.461 (148.116)	-200.328 (68.436)	-214.073 (78.387)	-214.023 (78.013)	-210.347 (73.968)
3	416.920 (129.938)	416.975 (130.487)	418.416 (130.733)	414.654 (123.539)	-195.076 (64.916)	-195.104 (65.189)	-195.824 (65.305)	-193.944 (61.692)
4	383.047 (124.311)	454.567 (140.884)	441.716 (139.199)	418.302 (130.572)	-178.140 (62.148)	-213.900 (70.469)	-207.474 (69.610)	-195.767 (65.264)
5	369.365 (110.240)	393.318 (106.938)	388.396 (106.112)	377.919 (102.326)	-171.299 (55.154)	-183.275 (53.611)	-180.814 (53.167)	-175.576 (51.201)

Internal exploration value (type 5)

**External
exploration
value (type)****Accept value (type)**

	BIC – Mean (SD)				Log-likelihood – Mean (SD)			
	1	2	3	4	1	2	3	4
1	361.257 (105.013)	548.604 (156.107)	522.921 (159.884)	477.867 (148.357)	-167.245 (52.537)	-260.919 (78.060)	-248.077 (79.884)	-225.550 (74.076)
2	417.279 (129.080)	457.209 (151.816)	455.005 (150.254)	444.104 (145.152)	-195.256 (64.450)	-215.221 (75.806)	-214.119 (75.021)	-208.668 (72.449)
3	406.971	414.234	414.596	409.015	-190.102	-193.734	-193.914	-191.124

	(122.403)	(127.378)	(126.994)	(119.598)	(61.148)	(63.612)	(63.418)	(59.696)
4	374.206	454.897	438.342	409.021	-173.719	-214.065	-205.787	-191.127
	(117.786)	(141.983)	(138.963)	(128.375)	(58.884)	(71.011)	(69.474)	(64.105)
5	361.257	397.480	388.639	375.377	-167.245	-185.357	-180.936	-174.305
	(105.013)	(110.894)	(108.673)	(103.269)	(52.537)	(55.551)	(54.419)	(151.62)

Internal exploration value (type 6)

**External
exploration
value (type)**

Accept value (type)

	BIC – Mean (SD)				Log-likelihood – Mean (SD)			
	1	2	3	4	1	2	3	4
1	370.055	556.119	530.452	493.283	-171.644	-264.676	-251.842	-233.258
	(112.401)	(162.204)	(165.195)	(154.203)	(56.238)	(81.116)	(82.546)	(77.013)
2	421.947	463.749	460.890	444.401	-197.590	-218.491	-217.061	-208.817
	(133.210)	(156.479)	(154.319)	(145.822)	(66.537)	(78.149)	(77.061)	(72.793)
3	411.349	419.220	419.042	407.035	-192.291	-196.226	-196.137	-190.134
	(125.098)	(132.693)	(130.918)	(118.731)	(62.527)	(66.300)	(65.409)	(59.298)
4	381.317	461.956	445.253	416.392	-177.275	-217.594	-209.243	-194.812
	(123.631)	(148.307)	(144.228)	(130.235)	(61.822)	(74.169)	(72.102)	(65.035)
5	370.055	405.821	395.908	378.552	-171.644	-189.527	-184.570	-175.893
	(112.401)	(118.412)	(114.466)	(103.144)	(56.238)	(59.306)	(57.316)	(51.579)

Major comment 4

The BICs provide a measure of the relative fit between models, but doesn't tell us if a model is doing a good job or not (there is always a winner even when comparing relative fits between only bad models). Therefore, please demonstrate how well your winning model replicates actual choice behavior by showing, for example, the model-predicted probability of making the same choice as participants.

Response

Thank you for the comment. In response, we have added it to supplementary Fig. S1, which presents our model comparison results based on the accuracy of each model predicting actual participants' choices. Our results reveal that the winning model demonstrates strong predictive accuracy, successfully reproducing participants' choices with an accuracy rate of 81.137% (SD = 5.408%).

Supplementary Figure. S1b. The accuracy of each model's prediction of participants' choices.

Major comment 5

It seems to me that some models are equivalent. For example, the best model (IE:5, EE:1, AV:1) should perfectly overlap with the (IE:5, EE:5, AV:1) model. Providing tables and conducting the model-recovery procedures will avoid such potential concerns. Accordingly, please perform a model-recovery analysis for all tested models, as well as a parameter-recovery analysis for your winning model. The validity of these approaches have been nicely described (Wilson & Collins, 2019, Ten simple rules for the computational modeling of behavioral data).

Response

Thank you for the comment. We acknowledged the suggestion regarding model and parameter recovery analyses. While our modeling serves two specific purposes: (1) to identify a model that best describes the observed behavioral patterns, and (2) to generate utility estimates for performing subsequent fMRI analyses. Hence, we prioritized parameter recovery that directly evaluates the reliability of our model parameters, rather than investigating the theoretical distinguishability between models through model through model recovery analyses. The results showed that our winning model could successfully recover all ten parameters (comprised four regressors in the three classes multinomial logistic regression framework) with high precision (all correlations: $r_s > 0.801$, $p_s < 0.001$). These strong parameter recovery results provided confidence in the reliability of our model's parameter estimates and their utility for the fMRI analyses. We have now included this in Supplementary Fig. S4 and a description in the manuscript (Lines 147-150).

To ensure the reliability of our model fitting procedure, we conducted parameter recovery analyses. They results showed that all ten parameters could be successfully recovered with high precision (all correlations: $r_s > 0.801$, $p_s < 0.001$; Supplementary Fig. S4).

Supplementary Figure. S4. Parameter recovery analysis of the best-fit model. The fitted parameters of each participant were first added with Gaussian noise ($\sigma = 0.01$ of the group SD), employed to generate simulated choice probability data, which was then randomly converted to simulated, categorical choice data. The simulation of each participants data was iterated for 10 times, resulting in choice data from 260 simulated participants. These simulated data were then fitted using the same model to test the correlation between the true, simulated parameters and the fitted parameters. The results showed that the fitting procedures could successfully recover all ten parameters with high precision (all correlations: $r_s > 0.801$). Black dashed lines indicate perfect recovery, and filled circles represents individual simulations.

Major comment 6

Please describe in detail how your models were fitted. Your explanation (lines 513-521) is short and lack details, making the procedure difficult to understand for someone unfamiliar with multinomial logistic regression.

Response

Thank you for the comment. We followed the reviewer's suggestion and have now revised the manuscript (Lines 537-539; 542-547 For ease of reading, original text is shown in black font color, while revisions are shown in red font color):

... The GLM involved a multinomial logistic regression analysis to estimate the probability of each decision type that follows two logit equations:

where y is the decision, and $P(y = Accept)$, $P(y = Internal\ exploration)$, and $P(y = External\ exploration)$ are the choice probability of accept, internal exploration, and external exploration decisions, respectively. The accept decision served as the reference category. This approach allowed us to simultaneously estimate the relative probabilities of making internal exploration and external exploration decision compared to accept decision. The model incorporated $Value_{Internal\ exploration}$, $Value_{External\ exploration}$, $Value_{Accept}$ and $Cost$ as predictors, where the three values are values of their corresponding decisions, and the cost is cumulated by each internal or external exploration within a trial.

Major comment 7

The mean-variance tradeoff is known to drive decision making. Please include another computation for ValueAccept as: Point (best option) – Variance(best option). Please observe that this model will have two betas, one for Point and one for Variance.

Response

Thank you for the comment. Following your recommendation, we implemented an alternative formulation of $Value_{Accept}$ using the equation:

$$Value_{Accept} = Point_{(the\ best\ option)} - Variance_{(the\ best\ option)}$$

We now have 120 models and we compared the BIC scores. However, our results indicated that this alternative formulation did not improve model performance. Specifically, the best-fit model and the second-best fit model remain the same. We have now included the results in Fig S1:

Major comment 8

Could you please describe the timings used for the paradigm in more detail? For example, how long was the time between making a decision and the feedback presentation? Was there any kind of jitter? This knowledge is important to remove any doubts of ‘smeared’ brain activation related to decision computations (e.g. Accept value) and feedback (e.g. actual outcome).

Response

Thank you for the comment. We followed the reviewer's suggestion that we have now included the feedback presentation, intertrial interval (ITI) and interstimulus interval (ISI) in our manuscript (Lines 522-523; 529-530):

... *Accept*, to end the trial by selecting an option and earning its actual points, followed by an outcome phrase that varied randomly between 1,000 and 3,000 ms, and an intertrial interval (ITI) that varied randomly between 3,000 and 6,000 ms.

... Each internal or external exploration costed one point and the cumulated cost on a trial was displayed at the top left-hand corner. A variable interstimulus interval (ISI) ranging from 1,000 to 3,000 ms was implemented after each internal or external exploration.

Major comment 9

Im puzzled by the contrasts shown in Fig. 3. In the GLM, the three decision values are entered as separate regressors. Accordingly, to assess the neural correlates of each decision value, the average betas should be compared to 0. However, in Fig. 3 it is stated that the average betas are contrasted with the average effect of the remaining two types of value. Why is this necessary and how is this mean calculated? What are the results if the average betas are compared to 0?

Response

Thank you for the comment regarding our analysis approach in Fig. 3. While comparing individual beta values to 0 is indeed a valid approach, our specific research question focused on identifying brain regions that show preferential encoding of one particular decision value type over others. In our GLM, the three decision values were first entered as separate regressors, and then at the contrast level, one of the three values were assigned as 1 and the remaining two were assigned -0.5. This approach enabled us distinguish areas that are specifically sensitive to one type of decision value.

Major comment 10

How were multiple comparisons corrected for? For example, Fig. 5 contains 27 different comparisons (yielding a Bonferroni-corrected alpha of 0.0019). Personally, I would conduct ANOVA-type analysis with appropriate post-hoc comparisons and corrections applied. This would provide a clearer oversight of the results, such as potential interactions between brain regions and decision values. In Fig. 5, please display individual data-points (e.g. violin-plot) rather than just mean+errorbar.

Response

Thank you for the comment. We have now included a three-way ANOVA with factors of Brain Regions (IPS, ACC and mPFC), Decision (internal exploration, external exploration and accept decisions) and Signal Type (internal exploration, external exploration and accept signals) in our manuscript (Lines 342-346):

To confirm that the three brain regions involved a different code, we performed a three-way ANOVA with factors of Brain Regions (IPS, ACC and mPFC), Decision (internal exploration, external exploration and accept decisions) and Signal type (internal exploration, external exploration and accept signals), and the results showed a significant three-way interaction ($F(8, 621) = 5.26, p < 0.001$).

Minor comment 1

BIC is not needed when comparing models with the same number of factors.

Response

Thank you for the comment. We performed additional analysis on model accuracy for model comparisons and added it to supplementary Fig. S1. Our results reveal that the winning model demonstrates strong predictive accuracy, successfully reproducing participants' choices with an accuracy rate of 81.137% (SD = 5.408%).

Supplementary Figure. S1b. The accuracy of each model's prediction of participants' choices

Minor comment 2

It is problematic to assign specific computations to specific decisions. For example, intuitively, Point(best)-Point(average) is for me what drives an Accept decision. However, the authors argue that the negative (i.e. Point(average)-Point(best)) drives external exploration. Without asking participants directly, it seems problematic to map specific computations to specific types of decision.

Response

Thank you for the comment. As demonstrated in Fig. 2a, our results show that the Accept decision is positively associated with Point(best) and negatively associated with Point(average), which is mathematically equivalent to saying that Point(best)-Point(average) drives Accept decisions. This relationship can be interpreted from two perspectives: either as Point(best)-Point(average) positively correlating with Accept

decisions, or as Point(average)-Point(best) negatively correlating with external exploration. These are essentially two sides of the same coin, representing the same underlying relationship in our data.

Minor comment 3

The authors discuss the impact of uncertainty on exploratory decision making, but surprisingly little uncertainty is included in the models (with actually none for external exploration). Could the authors include some measure of uncertainty reduction / information gain in the models? The number of unknown options seems like a plausible heuristic for external exploration. For internal exploration, the best option is the value multiplied with the variance, which for me indicates a riskier choice rather than a large information gain. Again, what is actually gained in terms of such a decision? Risk? It seems like including measures of information gain that may drive internal and external exploration is highly warranted.

Response

Thank you for the comment. For external exploration, we conducted additional analyses to address this concern by incorporating the number of unknown options as a new external exploration function in the model. However, this approach presented methodological challenges due to a strong multicollinearity between the number of unknown options and cumulative cost variables ($r = -0.986, p < 0.001$). We attempted to resolve this by removing the cumulative cost term to examine the effect of the new external exploration function. Our results showed that there were significant differences among models' BIC scores ($F(24,575) = 4.566, p < 0.001$) and prediction accuracy ($F(24,575) = 21.944, p < 0.001$). Also, our winning model (the light blue bars) demonstrated better performance when compared to all alternatives with a new EE function on BIC score (all $t_{s23} < -5.436, ps < .001$) and prediction accuracy (all $t_{s23} > 6.129, ps < .001$). Our results showed that the new external exploration function may not be the best account for describing external exploration.

For internal exploration, our model has already incorporated uncertainty through its variance term, which relates to potential information gain. This implementation aligns with the fundamental purpose of internal exploration: reducing option uncertainty. The variance term reflects that options with high uncertainty offer greater potential for information gain through internal exploration (options with low uncertainty naturally limit the potential information gain).

Model comparison using the new external exploration function on models' BIC and prediction accuracy.

Minor comment 4

Why is the accumulated cost function included in the decision model, rather than a 'constant' loss function? The accumulated cost is based on previous decisions, while the actual loss-to-be-incurred is what affects the current decision.

Response

Thank you for the comment. We suspect that the "loss-to-be-incurred" parameter suggested by the reviewer is the same as our "cumulated cost". The confusion could be due to ambiguity in our descriptions. In our task, each exploration (internal or external) would cost one point and the number of points used in each trial was updated in real time to participants. In other words, this total cost would be incurred once the trial was terminated by accepting an option. We have now revised our manuscript to clarify this (Lines 130-132):

Each decision phase terminated at the time of participants' decision. Each internal or external exploration costed one point and the cumulated cost incurred by the end of a trial was displayed in real time at the top left-hand corner.

Responses to the reviews

Distinct contributions of prefrontal, parietal, and cingulate signals to exploratory decisions

Victor K. S. Chan, Nicole H. L. Wong, Tsz-Fung Woo, Kei Watanabe, Masahiko Haruno, Chun-Kit Law, Bolton K. H. Chau

Reviewer #1

1. Summary of Revisions

The authors have made a real effort to respond to the comments, and the manuscript is improved in several places. The task itself is thoughtful and well-designed, and some of the new analyses — particularly the expanded behavioral modeling and clearer figure layouts (e.g., Figure 4c) — do help clarify parts of the paper.

That said, I still find the overall manuscript difficult to interpret. The analyses are complex, which is justified by the richness of the design, but the explanations often simplify things in a way that makes the results feel less meaningful or harder to trust. There's a lot of interesting data here, but the narrative doesn't fully do it justice. A more careful, precise framing would help make the findings clearer and more informative.

Response

We thank the reviewer again for the second-round review and for the opportunity to revise the manuscript. To improve clarity, we clarified the ROI definitions, added timing information and a marker for the average reaction time in the relevant figures, and aligned event labels across text and plots. We also simplified and tightened the explanations so that readers can trace each result to its method, for example the whole-brain analysis. We hope these changes can address the concern about interpretability and to make the findings clearer.

Comment 1 (Part 1)

Major Concerns

ROI Definition

The manuscript still doesn't clearly explain how ROIs were defined — whether they were based on peak activation, anatomical labels, or previous studies. This is important for interpreting the regression and time-course analyses. Please explicitly state how each ROI was defined, and include justification where relevant.

Response

Thank you for pointing this out. We have revised the “Region-of-interest (ROI) analysis” paragraph in the Methods to provide an explicit, step-by-step description of how every ROI was defined (Lines 613-619):

Region-of-interest (ROI) analysis. To examine the dynamics of signals during different aspects of decision making, we extracted the time series of brain regions found relevant to internal exploration, external exploration, and accept decisions in the whole brain analysis. All ROIs were first defined based on literature-derived coordinates. Masks were then created to extract activity of ROIs. IPS, which has been implicated in uncertainty driven exploratory behaviour, was defined as a spherical ROI (3 mm rad) centered on MNI coordinates [50, -44 ,43] (Mars et al.⁶²). ACC, a region consistently associated with tracking search or foraging value, was defined similarly [0, 28, 30], coordinates taken from Kolling et al⁸. The mPFC, wherein the value of the chosen option was reportedly encoded, was defined as [0, 51 ,0], coordinates taken from Blair et al⁶³.

Comment 1 (Part 2)

Temporal Alignment and Reporting Consistency

While the authors did clarify some timing definitions in the rebuttal, I found the resulting analysis choices hard to follow — not because they’re wrong, but because the framing doesn’t help make sense of them. The inclusion of many different trial types and decision points leads to results that are difficult to interpret, especially when significance is reported inconsistently or when some effects are emphasized while others are downplayed. This makes it difficult to assess the implications of the time-course analyses. Clarifying these temporal definitions by, for example, making separate figures for whole-brain and ROIs analysis separate when possible for internal exploration, external exploration and accept decisions could help strongly with understanding and feels essential for evaluating the reported effects. It would also be helpful to include the average response time directly on the time-course figures, or to indicate the mean response time per decision type if they differ significantly. This would provide important context for interpreting neural dynamics and help clarify whether observed time-locked activity patterns align meaningfully with decision execution.

To improve clarity:

- Make sure all significant effects are marked consistently and clearly.

Response

We appreciate the reviewer's suggestion and have undertaken a comprehensive consistency sweep of all statistical markings across the manuscript, figures, and supplementary files. All significant effects are flagged with a unified three-level asterisk code. * denotes $p < 0.05$; ** denotes $p < 0.01$; *** denotes $p \leq 0.001$. We have also revised the following paragraphs.

Manuscript (Figure. 2; lines 187-189):

The final results represented the averaged behavioral patterns across all participants. * denotes $p < 0.05$; ** denotes $p < 0.01$; *** denotes $p \leq 0.001$. Error bars represent \pm standard error of the mean (SEM).

Supplementary information (Supplementary Figure. S6; line 122):

These alternative models generated comparable results. * denotes $p < 0.05$, ** denotes $p < 0.01$, *** denotes $p \leq 0.001$. Error bars represent \pm SEM.

Supplementary information (Supplementary Figure. S7; lines 130-131):

Time-locked to the stimulus onset (solid black line). * denotes $p < 0.05$, ** denotes $p < 0.01$, *** denotes $p \leq 0.001$. Error bars represent \pm SEM. Only the trials after internal exploration decision were included.

Comment 1 (Part 3)

- Avoid downplaying or omitting non-significant or contradictory effects without explanation.

Response

Thank you for the helpful comment. We agree that some significant effects lacked sufficient explanation in Figure 5, and we have now added explanatory text in the legend of Supplementary Figure S.8 that is related to Figure 5 in the manuscript. These include explaining the negative external exploration and accept signals during accept decision in IPS; negative internal exploration signal during internal exploration decision in ACC; and negative external exploration and accept signals during internal exploration decision, and positive internal exploration signal during accept decision in mPFC (Lines 134:177):

Supplementary Figure S8. Time courses of the internal exploration, external exploration and accept signals in the IPS, ACC and mPFC during internal exploration, external exploration and accept decisions. (a). In the IPS, the internal exploration signal remained significant in all three types of decision (top panel; internal exploration: $t_{23} = 2.253$, $p = 0.034$; middle panel, external exploration: $t_{23} = 2.285$, $p = 0.032$; bottom panel, accept: $t_{23} = 2.872$, $p = 0.009$). **The IPS signals internal exploration value independent of the impending decision.** Besides, when an accept decision was

made subsequently, the IPS also showed a negative and then positive external exploration signal (bottom panel; negative: $t_{23} = -2.398, p = 0.025$; positive: $t_{23} = 2.320, p = 0.03$) and a negative accept signal ($t_{23} = -3.016, p = 0.006$). **In our task, higher external exploration or accept value indicates that additional internal exploration decision offers limited information gain, and that could be related to the decrease in IPS activity.** **(b).** The ACC reflected the value of external exploration regardless of the type of impending decision, there was a positive external exploration signal when participants made external exploration (middle panel; peaked at 1.22s: $t_{23} = 2.094, p = 0.047$, peaked at 7.39s: $t_{23} = 2.280, p = 0.032$, peaked at 11.83s: $t_{23} = 3.374, p = 0.003$) or accept decisions (bottom panel; $t_{23} = 2.102, p = 0.047$). **Inset (top panel):** During internal exploration, participants often repeated selections of the same option. When all internal exploration decision in each repetition were analyzed, the external exploration signal was initially not significant ($t_{23} = 1.555, p = 0.134$), after the first decision in each repetition was removed, and only the second and later decisions were analysed, the external exploration signal became significant (top panel; $t_{23} = 3.149, p = 0.004$). **Besides,** there was a negative internal exploration signal ($t_{23} = -2.613, p = 0.016$) as well, **suggesting that the ACC compares the relative advantage/disadvantage of external exploration as opposed to the impending internal exploration.** **(c).** The mPFC signalled general decision value. The mPFC reflected a positive accept signal during accept decisions (bottom panel; $t_{23} = 4.205, p > 0.001$). **Besides,** there was a negative internal exploration signal ($t_{23} = -3.373, p = 0.003$), consistent with a value-difference code in which mPFC emphasises the value of the current decision (accept) relative to its alternative (internal exploration). During internal exploration, there were positive accept signals (peaked at -0.43s: $t_{23} = 2.474, p = 0.021$; peaked at 3.51s: $t_{23} = 2.838, p = 0.009$), after that, the mPFC switched to encode an internal exploration signal (top panel; $t_{23} = 2.666, p = 0.014$). In our task, options worth internal exploration are typically high in both uncertainty and average value. This pattern suggests that such options are often evaluated as accept decisions at first, but consideration of their uncertainty leads participants to withhold accept decisions and switch to internal exploration. During external exploration, the mPFC signalled external exploration value (middle panel; peaked at 7.65s: $t_{23} = 3.007, p = 0.006$, peaked at 11.85s: $t_{23} = 3.741, p = 0.001$), **there were also negative internal exploration signals** (peaked at -1.05s: $t_{23} = -2.536, p = 0.018$; peaked at 3.82s: $t_{23} = 2.671, p = 0.014$; peaked at 8.04s: $t_{23} = 2.684, p = 0.013$). Again, these were consistent with a value difference code in which the mPFC emphasizes the value of the current decision (external exploration) compared with its alternative (internal exploration). Time-locked to the stimulus onset (solid black line). A dotted vertical line marks the average reaction time (RT) for the decision type shown in each panel. * denotes $p < 0.05$, ** denotes $p < 0.01$, *** denotes $p \leq 0.001$. Shaded areas represent \pm SEM.

Comment 1 (Part 4)

- add reaction time for each different decision type and maybe include a marker for the average RT those figures of time course analysis?

Response

Thank you for the helpful suggestion. We have now indicated the average reaction time on time-course figures (Fig 6, Supplementary Fig 8, Supplementary Fig 11) and updated the figure legends correspondingly. On those figures, the dotted black line indicates the average reaction time of each decision. Below, we have included the snippets of figure legends that are newly included in the updated manuscript.

Fig. 6. in the main manuscript (Lines 400:402):

Time-locked to the stimulus onset (solid black line). The dotted vertical line indicates the average reaction time (RT) of each decision type: Internal exploration ($M = 3.76$ s, $SD = 1.56$ s), external exploration ($M = 2.75$ s, $SD = 1.19$ s), and Accept ($M = 3.06$ s, $SD = 1.50$ s).

Supplementary Figure. 8 in the Supplementary Information (Lines 173:176):

Time-locked to the stimulus onset (solid black line). A dotted vertical line indicates the average reaction time (RT) for each decision type shown: Internal exploration ($M = 3.76$ s, $SD = 1.56$ s), external exploration ($M = 2.75$ s, $SD = 1.19$ s), and Accept ($M = 3.06$ s, $SD = 1.50$ s).

Supplementary Figure. 11 in the Supplementary Information (Lines 233:235):

There was no significant correlation between the peak signal of each participant and their proportions of internal exploration, external exploration, and accept decision made (right

panel). Time-locked to the stimulus onset (solid black line). A dotted vertical line indicates the average reaction time (RT) for external exploration: Internal exploration ($M = 3.76$ s, $SD = 1.56$ s), external exploration ($M = 2.75$ s, $SD = 1.19$ s), and Accept ($M = 3.06$ s, $SD = 1.50$ s). * denotes $p < 0.05$; ** denotes $p < 0.01$. Shaded areas represent \pm SEM.

Comment 1 (Part 5)

Figure S8 does a good job of laying out decision timing and would help with these issues — see below.

ACC and Unchosen Value

I greatly appreciate their answers about the ACC and conflict, and agree to their take on those results. However, I still believe it could potentially strengthen their result to see (and possibly demonstrate) that the ACC tracks when the second best options requires external exploration for example. This is just an idea but if the authors are defining the unchosen option as the second-best alternative, they could time-lock to the stimulus onset following a switch between chosen and unchosen, and look at ACC activity around that point. Breaking down these trials by what happens next (e.g., repeat explore, accept, or switch type) and showing the distribution across participants would make the analysis more informative and better suited to the task?

Response

Thank you for the thoughtful suggestion regarding ACC and the second-best option. After restricting the analysis to cases where the best unchosen alternative required external exploration, breaking trials down into “repeat exploration,” “accept,” and “switch type” yielded too few observations per participant for reliable estimation. To avoid underpowered analyses while preserving the reviewer’s core contrast, we therefore pooled repeat and switch within each decision family. Under this approach, there was a positive unchosen value signal in ACC during internal exploration ($\beta = 0.661$, $t_{(23)} = 2.255$, $p = 0.034$). During accept, there was a positive chosen value signal ($\beta = 0.087$, $t_{(23)} = 2.206$, $p = 0.038$) and a negative unchosen value signal ($\beta = -0.228$, $t_{(23)} = -2.605$, $p = 0.016$).

Although the analyses were run on pooled trials, as indicated by the red dots in the figure below, the number of trials that satisfy the inclusion criteria (i.e., cases in which the best unchosen alternative required external exploration, and also “repeat explore, accept, or switch type”) is relatively low. Importantly, the chosen value and unchosen value of included trials falls within a narrow range, which limits the generalisability of these analyses. In view of the above reasons, we propose not to include this analysis in the main manuscript or in the supplementary.

Comment 2 (Part 1)

Figures and Supplementary Material

I appreciate the figures the authors did in answer to comment 18 (part 2 3 and 4); I don't think it would strengthen the manuscript to include it in the paper; however this did make me feel like internal exploration, external exploration and accept decision are maybe not as independent as the time series analysis makes it seem by comparing those trial types in the same figure.

Figure S8 is one a very figure in the paper and should probably be moved to the main text. It gives a helpful overview of decision timing, but some aspects are still confusing:

- The inset panel isn't explained and doesn't include stats. If it highlights something specific, that should be stated clearly, with the appropriate statistical test?

Response

Thank you for your comment. We have now revised the manuscript and the legend of Supplementary Figure S8b to explain what the inset shows. We hope these could clear up the confusion.

Manuscript (line 335):

During internal exploration, **the effect of external exploration signal was initially non-significant** (Supplementary Fig. S8b, the inset in the top panel; $\beta = 0.091$, $t_{23} = 1.555$, $p = 0.134$).

Supplementary Figure S8b (lines 149-154):

Inset (top panel): During internal exploration, participants often repeated selections of the same option. When all internal exploration decision in each repetition were analyzed, the external exploration signal was initially not significant ($t_{23} = 1.555$, $p = 0.134$), after the first decision in each repetition was removed, and only the second and later decisions were analysed, the external exploration signal became significant (top panel; $t_{23} = 3.149$, $p = 0.004$).

The effect size for the inset has also been added to Figure S8b.

Comment 2 (Part 2)

- Significance stars need to be used consistently. A p-value of .134 isn't a trend. If trends are to be reported, please define what threshold you're using (e.g., $p < .06$) and use different symbols from actual significance markers; maybe also report the effect size to explain why although it is not significant you believe it is still important to report.

Response

Thank you for your suggestion. We have revised the manuscript so that $p = .134$ is no longer labelled as a trend.

Manuscript (line 335):

During internal exploration, the effect of external exploration signal was initially non-significant (Supplementary Fig. S8b, the inset in the top panel; $\beta = 0.091$, $t_{23} = 1.555$, $p = 0.134$).

Supplementary Figure S8b (lines 149-152):

Inset (top panel): During internal exploration, participants often repeated selections of the same option. When all internal exploration decision in each repetition were analyzed, the external exploration signal was initially not significant ($t_{23} = 1.555$, $p = 0.134$).

Also, the effect size for the inset has also been added to Figure S8b.

The inset was retained to let readers compare the ACC signals before and after the first decision in each repetition was excluded. Presenting the two figures side by side clarifies the signal shift and prevents any impression of selective reporting. We will include a brief justification for retaining the inset if the reviewer believes this would strengthen the manuscript.

Comment 2 (Part 3)

More broadly, the timing information in this figure could be used more in the main narrative to clarify when decisions tend to happen and how different decision types unfold.
?

Response

Thank you for your suggestion. We have added reaction-time markers to Figures 6, S8, and S11 and expanded the related narrative in the manuscript. We hope that these revisions clarify and help readers visualize when each decision typically occurs.

Fig. 6. in the main manuscript (Lines 400:402):

Time-locked to the stimulus onset (solid black line). The dotted vertical line indicates the average reaction time (RT) of each decision type: Internal exploration ($M = 3.76$ s, $SD = 1.56$ s), external exploration ($M = 2.75$ s, $SD = 1.19$ s), and Accept ($M = 3.06$ s, $SD = 1.50$ s).

Supplementary Figure. 8 in the Supplementary Information (Lines 173:176):

Time-locked to the stimulus onset (solid black line). A dotted vertical line indicates the average reaction time (RT) for each decision type shown: Internal exploration (M = 3.76 s, SD = 1.56 s), external exploration (M = 2.75 s, SD = 1.19 s), and Accept (M = 3.06 s, SD = 1.50 s).

Supplementary Figure. 11 in the Supplementary Information (Lines 233:235):

There was no significant correlation between the peak signal of each participant and their proportions of internal exploration, external exploration, and accept decision made (right panel). Time-locked to the stimulus onset (solid black line). A dotted vertical line indicates the average reaction time (RT) for external exploration (M = 2.75 s, SD = 1.19 s). * denotes $p < 0.05$; ** denotes $p < 0.01$. Shaded areas represent \pm SEM.

Minor Clarifications 1

- In the behavioral analysis (Figure 2b), the sliding window method is now clearer, but it would help to state why this transformation was used and what it's meant to reveal.

Response

Thank you for your suggestion. We have now included both the rationale and the interpretive payoff of the 11-rank sliding-window transformation in the legend of Figure 2b (lines 182–185):

A participant-specific analysis was conducted by ranking internal exploration, external exploration, and accept values into 30 probability classes, which were then consolidated into 20 levels using an 11-rank sliding window approach. This transformation increases the number of trials per level, including at extreme values, so the estimates are more stable and less noisy. The resulting smooth curves show the decision threshold, the point at which exploration drops off and acceptance becomes more likely. The choice ratios between decision pairs were analyzed across these levels, excluding trials involving the third decision type.

Minor Clarifications 2

- Please clarify what “trial 0” refers to.

Response

Thank you for the comment. After carefully re-checking the manuscript and all supplementary material, we could not find the term “trial 0”. We believe the confusion may come from Figure 1B, bottom panel, where the x-axis is labelled “0 1 2 3 ...” and the tick 0 might have been misread as trial 0. In this panel the x-axis denotes the number of exploration decisions made within the same trial rather than the ordinal index of the trial itself. We have inserted a brief note in the legend of Fig. 1 (Lines 129–130):

The proportion of trials with different total number of internal exploration and external exploration (bottom panel). **Distribution of the number of exploration decisions per trial (0 = no exploration, 1 = one exploratory decision, etc.).**

We hope this resolves the reviewer’s concern and we are grateful for the opportunity to improve the clarity of the figure.

Minor Clarifications 3

- Specify which trial durations were included in whole-brain analyses (e.g., Figure 3).

Response

Thank you for your comment. We have now highlighted the trial durations in the Methods and added cross-references to aid reading. All information on the trial durations and modelling choices used in our whole-brain contrasts is provided in the following places:

The legend in Fig. 3 (Lines 217:218).

Cluster-based threshold $z > 3.10$, $p < 0.05$. **See more details in the Methods section, “fMRI data analysis: whole-brain analysis”.**

Methods—fMRI data analysis: whole-brain analysis (Lines 596 & 604).

Whole-brain analysis. To identify brain regions associated with internal exploration, external exploration and accept decisions, we performed whole-brain analyses using a univariate GLM approach (Fig. 3).

...trial durations used two types of event constants (duration = 1 and from trial onset to every decision), time-locked to participants’ decisions, were included

These additions make it clear that the full description of trial inclusion criteria is located in the Methods section and that Figure 3 is derived from that analysis. We hope this clarification addresses the reviewer's concern.

Minor Clarifications 4

- Avoid using “trend” for p-values > .10.

Response

Thank you for your suggestion. We have now revised the manuscript and the legend of Supplementary Figure S8b to avoid using the word “trend”.

Manuscript (line 335):

During internal exploration, the effect of external exploration signal was initially non-significant (Supplementary Fig. S8b, the inset in the top panel; $\beta = 0.091$, $t_{23} = 1.555$, $p = 0.134$).

Supplementary Figure S8b (lines 149:152):

Inset (top panel): During internal exploration, participants often repeated selections of the same option. When all internal exploration decision in each repetition were analyzed, the external exploration signal was initially not significant ($t_{23} = 1.555$, $p = 0.134$).

Minor Clarifications 5

- Check that all figures referenced in the text (e.g., “middle panel of Figure 1b”) are present and labeled.

Response

Thank you for your suggestion. We have carried out a line-by-line audit of the manuscript, and supplementary information. Every in-text reference now points to an existing figure or sub-panel, and every panel is explicitly labeled. No discrepancies were found.

Reviewer #2

Comment 1 (Part 1)

I sincerely thank the authors for their extensive work in replying to my concerns. I am satisfied with most responses, but still have some issues with the following replies: Concerning major comment 1:

- The models used are very difficult to understand in terms of how many parameters are fitted in each model. Can the authors please provide a Table with all the models listed together with the number of parameters?

Response

Thank you for pointing this out. We acknowledge that inconsistent terminology between the labels in Figure S1 (e.g. *internal exploration Value*) and the notation used in the text (e.g. $Value_{Internal\ exploration}$) may have caused confusion. Overall, all models involved the same set of parameters, but they only differ by the way each value term was defined. To improve the clarity, we have revised Supplementary Figures S1, S2 and S3 to make the terms consistent.

Revised Supplementary Figure S1:

Revised Supplementary Figure S2:

Revised Supplementary Figure S3:

In addition, we have added a table listing the number of parameters for each model, together with a companion figure that defines the model numbering used in the table. Please note that model 81 is the best fit model and is highlighted in green.

Models 1-20:

Model	Model's parameters									
	β_1 Intercept	β_2 Value _{IE}	β_3 Value _{EE}	β_4 Value _{Acc}	β_5 Cost	β_6 Intercept	β_7 Value _{IE}	β_8 Value _{EE}	β_9 Value _{Acc}	β_{10} Cost
1	-2.837	-15.836*	0.825***	-1.875***	-0.217	0.135	-0.191	1.971***	-4.936***	-1.014***
2	-96.165	-693.27	-0.054	98.276	-0.153	0.397**	-0.424***	-0.247**	-0.445***	-0.706***
3	-2.661*	-10.192*	0.198*	-3.129***	-0.201	0.504***	-0.501***	-0.057	-1.025***	-0.813***
4	-3.722	-19.7	-0.122	-2.041***	-0.293**	0.638***	-0.359***	-0.273**	-0.667***	-0.728***
5	-3.124	-16.216*	0.392	-1.605***	-0.141	0.281	0.018	-0.255	-2.840***	-1.185***
6	-46.278	-317.83	-0.507***	37.117	-0.481***	0.092	-0.064	-2.602***	-0.201**	-1.751***
7	-5.586	-27.027	-0.145	-3.140*	-0.400**	0.14	-0.029	-2.537***	-0.389***	-1.769***
8	-3.025	-16.814	-0.797***	-1.886***	-0.729***	0.282	0.094	-2.591***	-0.666***	-1.858***
9	-0.606	-16.884*	-26.532	22.77	-0.251*	2.908	0.084	-29.255	23.526	-1.325***
10	-4.463*	-7.561*	-0.880***	-11.719**	-0.416***	0.221	0.105	-2.919***	-0.328***	-1.395***
11	-5.62	-28.541	-0.507*	-2.877*	-0.423***	0.254	0.166*	-2.802***	-0.399***	-1.434***
12	-2.942	-17.384	-1.086***	-1.867***	-0.578***	0.411	0.292***	-2.877***	-0.715***	-1.500***
13	-2.957	-16.010*	0.542***	-1.071***	-0.527***	0.132	-0.114	1.975***	-2.607***	-1.749***
14	-4.453**	-7.924*	0.632***	-11.111***	-0.471**	0.255	-0.345***	2.371***	-0.513***	-1.580***
15	-4.691	-22.597	0.608***	-2.637***	-0.551***	0.344	-0.349***	2.362***	-0.970***	-1.701***
16	-1.759	-10.167	0.852***	-1.890***	-0.642***	0.372*	-0.358**	2.445***	-0.480**	-1.643***
17	-2.837	-15.836*	1.137***	-0.579***	-0.217	0.135	-0.191	2.723***	-1.817***	-1.014***
18	-4.714*	-12.134	1.410***	-10.840***	-0.286*	0.448*	-0.328**	3.460***	-0.569***	-0.806***
19	-4.526	-23.147	1.198***	-2.702***	-0.364*	0.500*	-0.357**	3.374***	-0.732***	-0.879***
20	-1.382	-10.168	1.637***	-1.868***	-0.403**	0.543*	-0.308*	3.619***	-0.412*	-0.884***

* denotes $p < 0.05$; ** denotes $p < 0.01$; *** denotes $p \leq 0.001$.

Models 21-40:

Model	Model's parameters									
	β_1 Intercept	β_2 Value _{IE}	β_3 Value _{EE}	β_4 Value _{Acc}	β_5 Cost	β_6 Intercept	β_7 Value _{IE}	β_8 Value _{EE}	β_9 Value _{Acc}	β_{10} Cost
21	0.381	-1.620***	0.863***	-1.829***	-0.659***	0.164	-0.118	2.065***	-5.090***	-1.104***
22	-2.644**	-0.268	-0.149*	-9.613***	-0.566***	0.373**	-1.178***	-0.227*	0.586***	-0.948***
23	-6.923	-98.66	-0.027	85.41	-0.567***	0.392**	0.310**	-0.191*	-1.110***	-0.977***
24	-0.01	-0.546**	-0.138	-2.224***	-0.550***	0.702***	-0.569***	-0.223*	-0.627***	-0.955***
25	0.185	-1.546***	0.345	-1.556***	-0.648***	0.319	-0.087	-0.262	-2.925***	-1.291***
26	-2.182*	-0.35	-0.883***	-8.839***	-0.936***	0.137	-0.372***	-2.755***	0.123	-1.884***
27	-6.415	-95.817	-0.799***	82.786	-0.970***	0.158	0.054	-2.851***	-0.358**	-1.976***
28	0.04	-0.658**	-0.808***	-1.913***	-0.908***	0.276	-0.077	-2.608***	-0.626***	-1.889***
29	1.855	-1.543***	-13.282	10.504	-0.702***	2.524	-0.166	-24.866	19.53	-1.384***
30	-1.828	-0.214	-1.223***	-8.747***	-0.699***	0.288	-0.214	-3.099***	-0.135	-1.382***
31	-5.117	-80.81	-1.121***	69.556	-0.749***	0.346	0.147	-3.156***	-0.420*	-1.452***
32	0.243	-0.622**	-1.079***	-1.867***	-0.724***	0.394	-0.1	-2.818***	-0.592***	-1.459***
33	0.326	-1.609***	0.618***	-0.960***	-0.959***	0.162	-0.162	2.091***	-2.622***	-1.839***
34	-2.436*	-0.405	0.910***	-9.774***	-0.882***	0.329	-0.891***	2.535***	0.377**	-1.773***
35	-3.51	-50.182	0.925***	42.031	-0.936***	0.323	0.09	2.610***	-0.936***	-1.868***
36	0.084	-0.711**	0.844***	-1.881***	-0.868***	0.414*	-0.554***	2.447***	-0.336*	-1.807***
37	0.381	-1.620***	1.191***	-0.471**	-0.659***	0.164	-0.118	2.855***	-1.823***	-1.104***
38	-2.112*	-0.237	1.602***	-10.328***	-0.537***	0.496*	-0.515**	3.595***	-0.007	-0.865***
39	-4.31	-68.371	1.517***	58.151	-0.588***	0.518*	0.187	3.627***	-0.773***	-0.925***
40	0.443	-0.596**	1.591***	-1.847***	-0.593***	0.557*	-0.395***	3.583***	-0.27	-0.985***

* denotes $p < 0.05$; ** denotes $p < 0.01$; *** denotes $p \leq 0.001$.

Models 41-60:

Model	Model's parameters									
	β_1 Intercept	β_2 Value _{IE}	β_3 Value _{EE}	β_4 Value _{Acc}	β_5 Cost	β_6 Intercept	β_7 Value _{IE}	β_8 Value _{EE}	β_9 Value _{Acc}	β_{10} Cost
41	0.272	1.808***	0.731***	-1.998***	-0.777***	0.21	0.469*	2.063***	-4.925***	-1.267***
42	-2.087	1.447***	-0.314***	-5.482*	-0.587***	0.365**	0.1	-0.332***	-0.412***	-0.911***
43	-0.745	1.411***	-0.197*	-1.464***	-0.625***	0.367**	-0.148	-0.172	-0.964***	-0.996***
44	-0.034	0.693**	-0.268**	-2.017***	-0.496***	0.825***	-0.477***	-0.286**	-1.522***	-0.819***
45	0.043	1.829***	0.820**	-2.157***	-0.581***	0.385	0.493***	-0.186	-2.700***	-1.362***
46	-2.372	1.403***	-0.900***	-7.045*	-0.996***	0.139	0.374*	-2.662***	-0.029	-1.945***
47	-0.658	1.418***	-0.769***	-1.724***	-0.983***	0.166	0.328*	-2.628***	-0.157	-1.967***
48	0.027	0.726**	-0.913***	-1.605***	-0.926***	0.348	-0.152	-2.568***	-0.926***	-1.889***
49	1.034	1.792***	-5.408	3.069	-0.777***	1.984	0.482***	-18.582	14.059	-1.497***
50	-1.841	1.452***	-1.239***	-6.382*	-0.796***	0.338	0.420**	-2.944***	-0.106	-1.504***
51	-0.292	1.483***	-1.116***	-1.515**	-0.813***	0.363	0.383**	-2.911***	-0.165*	-1.530***
52	0.235	0.902***	-1.213***	-1.372***	-0.743***	0.449	0.02	-2.863***	-0.740**	-1.492***
53	0.18	1.783***	0.365**	-1.299***	-0.967***	0.183	0.487**	1.970***	-2.493***	-1.935***
54	-1.867	1.225***	0.719***	-6.008**	-0.818***	0.191	-0.066	2.446***	-0.548***	-1.750***
55	-0.494	1.203***	0.716***	-1.686***	-0.879***	0.212	-0.281*	2.457***	-1.041***	-1.838***
56	0.073	0.519*	0.823***	-1.820***	-0.800***	0.533**	-0.662***	2.477***	-1.351***	-1.712***
57	0.272	1.808***	1.003***	-0.849***	-0.777***	0.21	0.469*	2.844***	-1.668***	-1.267***
58	-1.418	1.329***	1.574***	-6.175**	-0.603***	0.479*	0.074	3.622***	-0.490***	-0.956***
59	-0.064	1.333***	1.465***	-1.649***	-0.660***	0.486*	-0.075	3.555***	-0.748***	-1.013***
60	0.429	0.710**	1.628***	-1.558***	-0.558***	0.644**	-0.273	3.654***	-0.822***	-0.940***

* denotes $p < 0.05$; ** denotes $p < 0.01$; *** denotes $p \leq 0.001$.

Models 61-80

Model	Model's parameters									
	β_1 Intercept	β_2 Value _{IE}	β_3 Value _{EE}	β_4 Value _{Acc}	β_5 Cost	β_6 Intercept	β_7 Value _{IE}	β_8 Value _{EE}	β_9 Value _{Acc}	β_{10} Cost
61	0.404	2.098***	0.774***	-2.687***	-0.795***	0.036	0.172	2.155***	-5.067***	-1.197***
62	-1.895	1.477***	-0.585***	-4.923*	-0.577***	-0.036	-0.787***	-0.239**	-0.614***	-0.809***
63	-0.725	1.389***	-0.441***	-1.565***	-0.624***	-0.069	-1.116***	-0.011	-1.161***	-0.924***
64	0.137	-0.521	-0.275**	-3.146***	-0.519***	0.15	-5.337***	0.351***	-5.218***	-0.702***
65	0.235	2.157***	0.09	-2.149***	-0.794***	0.252	0.338	-0.279	-2.703***	-1.360***
66	-1.754	1.573***	-1.485***	-5.665*	-1.166***	-0.009	0.1	-2.722***	-0.095	-1.898***
67	-0.423	1.587***	-1.374***	-1.494***	-1.160***	0.002	0.002	-2.644***	-0.238**	-1.908***
68	0.179	0.563*	-1.203***	-1.754***	-1.037***	0.225	-1.066**	-2.335***	-1.455***	-1.788***
69	2.184	2.106***	-17.077	13.156	-0.812***	2.692	0.34	-27.232	21.848	-1.436***
70	-1.391	1.687***	-1.758***	-5.343*	-0.829***	0.207	0.251	-2.987***	-0.172**	-1.440***
71	-0.108	1.728***	-1.663***	-1.320**	-0.840***	0.213	0.187	-2.931***	-0.214**	-1.461***
72	0.324	1.110***	-1.589***	-1.126***	-0.776***	0.276	-0.308	-2.814***	-0.709**	-1.426***
73	0.326	2.100***	0.528**	-1.891***	-1.038***	0.01	0.316	2.055***	-2.582***	-1.900***
74	-1.929	1.102***	0.996***	-6.309*	-0.845***	-0.142	-0.847**	2.490***	-0.710***	-1.660***
75	-0.497	1.090***	1.001***	-1.971***	-0.924***	-0.137	-1.059***	2.487***	-1.129***	-1.766***
76	0.253	-0.458	0.889***	-2.679***	-0.858***	0.095	-3.486***	2.374***	-3.378***	-1.676***
77	0.404	2.098***	1.067***	-1.464***	-0.795***	0.036	0.172	2.970***	-1.664***	-1.197***
78	-1.358	1.296***	1.965***	-6.122*	-0.530***	0.227	-0.502*	3.702***	-0.659***	-0.887***
79	-0.023	1.296***	1.836***	-1.838***	-0.593***	0.208	-0.655**	3.614***	-0.836***	-0.964***
80	0.534	0.058	1.690***	-2.096***	-0.567***	0.265	-2.406***	3.457***	-2.354***	-0.959***

* denotes $p < 0.05$; ** denotes $p < 0.01$; *** denotes $p \leq 0.001$.

Model 81-100:

	Model's parameters									
Model	β_1 Intercept	β_2 Value _{IE}	β_3 Value _{EE}	β_4 Value _{Acc}	β_5 Cost	β_6 Intercept	β_7 Value _{IE}	β_8 Value _{EE}	β_9 Value _{Acc}	β_{10} Cost
81	0.079	2.558***	0.784***	-3.540***	-0.819***	0.087	0.450*	2.125***	-5.193***	-1.222***
82	-2.691	1.234***	-0.678***	-6.806*	-0.537***	-0.075	-1.026***	-0.086	-0.814***	-0.790***
83	-1.086*	1.179***	-0.493***	-2.227***	-0.596***	-0.026	-1.278***	0.18	-1.285***	-0.916***
84	0.241	-0.321	-0.276*	-3.302***	-0.549***	0.710**	-3.404***	0.439***	-4.183***	-0.759***
85	-0.091	2.522***	0.188	-3.011***	-0.767***	0.27	0.480**	-0.219	-2.859***	-1.338***
86	-2.399	1.549***	-1.766***	-6.909*	-1.204***	0.001	0.07	-2.665***	-0.108	-1.853***
87	-0.747	1.570***	-1.635***	-1.860***	-1.195***	0.028	-0.03	-2.556***	-0.280***	-1.855***
88	0.106	0.625**	-1.368***	-1.823***	-1.048***	0.338	-0.980***	-2.161***	-1.511***	-1.726***
89	1.874	2.472***	-17.403	12.69	-0.817***	2.679	0.477**	-26.97	21.488	-1.438***
90	-1.938	1.877***	-2.259***	-6.038	-0.821***	0.217	0.385*	-3.119***	-0.095	-1.420***
91	-0.415	1.917***	-2.157***	-1.468**	-0.830***	0.23	0.332*	-3.056***	-0.149*	-1.436***
92	0.14	1.572***	-2.159***	-0.735**	-0.798***	0.263	0.138	-3.055***	-0.303	-1.442***
93	-0.013	2.515***	0.490**	-2.707***	-1.035***	0.032	0.519*	2.027***	-2.727***	-1.894***
94	-2.621	0.803***	0.956***	-8.263*	-0.788***	-0.115	-0.970***	2.456***	-0.851***	-1.628***
95	-0.791	0.886***	0.967***	-2.593***	-0.887***	-0.065	-1.061***	2.460***	-1.139***	-1.740***
96	0.31	-0.355*	0.885***	-2.688***	-0.858***	0.378	-2.529***	2.363***	-2.834***	-1.693***
97	0.079	2.558***	1.081***	-2.302***	-0.819***	0.087	0.450*	2.929***	-1.837***	-1.222***
98	-1.981	1.159***	2.074***	-7.695*	-0.477***	0.259	-0.441*	3.653***	-0.708***	-0.888***
99	-0.305	1.217***	1.943***	-2.401***	-0.562***	0.261	-0.521**	3.589***	-0.867***	-0.978***
100	0.49	0.259	1.796***	-1.969***	-0.558***	0.408	-1.527***	3.455***	-1.779***	-0.996***

* denotes $p < 0.05$; ** denotes $p < 0.01$; *** denotes $p \leq 0.001$.

Model 101-120:

Model	Model's parameters									
	β_1 Intercept	β_2 Value _{IF}	β_3 Value _{EF}	β_4 Value _{Acc}	β_5 Cost	β_6 Intercept	β_7 Value _{IF}	β_8 Value _{EF}	β_9 Value _{Acc}	β_{10} Cost
101	0.036	2.464***	0.696***	-3.459***	-0.703***	0.106	0.653**	2.004***	-5.180***	-1.226***
102	-2.96	1.186***	-0.677***	-7.473*	-0.487***	0.004	-0.863***	-0.112	-0.779***	-0.826***
103	-1.163*	1.144***	-0.497***	-2.329***	-0.549***	0.056	-1.103***	0.149	-1.261***	-0.958***
104	0.145	-0.117	-0.318*	-3.123***	-0.527***	0.812**	-2.532***	0.291**	-3.444***	-0.838***
105	-0.154	2.450***	0.605*	-3.411***	-0.559***	0.304	0.606***	-0.174	-2.958***	-1.304***
106	-2.661	1.438***	-1.679***	-7.706*	-1.095***	0.043	0.193	-2.710***	-0.074	-1.863***
107	-0.823	1.459***	-1.529***	-2.095***	-1.087***	0.072	0.107	-2.613***	-0.243**	-1.871***
108	0.041	0.674**	-1.363***	-1.748***	-0.998***	0.338	-0.507*	-2.334***	-1.131***	-1.796***
109	1.293	2.402***	-10.563	6.348	-0.690***	2.356	0.600***	-23.333	18.113	-1.434***
110	-2.199	1.801***	-2.245***	-6.792	-0.729***	0.247	0.540**	-3.224***	-0.043	-1.413***
111	-0.495	1.842***	-2.126***	-1.663**	-0.739***	0.264	0.503**	-3.170***	-0.091	-1.429***
112	0.135	1.381***	-2.056***	-0.982***	-0.724***	0.311	0.351	-3.121***	-0.257	-1.447***
113	-0.057	2.420***	0.366*	-2.732***	-0.871***	0.06	0.694**	1.945***	-2.820***	-1.856***
114	-2.829	0.742***	0.884***	-8.902*	-0.724***	-0.023	-0.792***	2.431***	-0.820***	-1.655***
115	-0.852	0.850***	0.900***	-2.708***	-0.822***	0.031	-0.871***	2.427***	-1.114***	-1.766***
116	0.23	-0.151	0.887***	-2.501***	-0.820***	0.479*	-1.851***	2.362***	-2.308***	-1.736***
117	0.036	2.464***	0.957***	-2.364***	-0.703***	0.106	0.653**	2.760***	-2.018***	-1.226***
118	-2.234	1.073***	1.948***	-8.479*	-0.424**	0.309	-0.296	3.562***	-0.662***	-0.898***
119	-0.387	1.170***	1.843***	-2.584***	-0.507***	0.326	-0.350*	3.515***	-0.840***	-0.990***
120	0.457	0.380**	1.829***	-1.896***	-0.509***	0.513*	-0.969**	3.506***	-1.373***	-1.008***

* denotes $p < 0.05$; ** denotes $p < 0.01$; *** denotes $p \leq 0.001$.

We hope the table could help avoid the confusions. We will include it in the Supplementary Information if the reviewer believes it would strengthen the manuscript.

Comment 1 (Part 2)

- Why was the cost only included in the full model? This still leaves the question whether a simpler model with the cost function(e.g. Ac+cost) provides a more parsimonious fit.

Response

Thank you for the comment. We have now tested whether the cost would provide a more parsimonious fit. To test this more extensively, we added a cumulative cost regressor to models of each possible combinations of value parameters (e.g., Ac + cost, In + Ac + cost, Ex + Ac + cost, etc.), and then to compare their goodness-of-fit (please refer to the figure below). A repeated-measures ANOVA revealed significant differences in BIC scores ($F(6,161) = 10.870, p < .001$), and in prediction accuracies ($F(6,161) = 41.640, p < .001$). Pairwise comparisons also showed that our original winning model (In + Ex + Ac + cost; the red bars in the figures below) still outperformed every alternative that included the cost term, both in BIC (all $t_{s23} < -6.241, p_s < .001$) and in prediction accuracy (all $t_{s23} > 6.260, p_s < .001$). These new results again support our original model selection that, despite its greater complexity, the full model still provides the best account of the behavioural data, which justifies retaining all parameters. For parameter labels, "In" represents Internal exploration value, "Ex" represents External exploration value, and "Ac" represents Accept value.

Comment 2

Concerning major comment 2:

- There are two identical models in the VBA analyses (EEV1, IEV1 and EEV5, IEV1), yet only one of them is considered the best. How is this possible?

Response

Thank you for the comment. We think the confusion could be related to the observation that the models using IEV5 instead of IEV1, as model (EEV1, IEV5, ACCV1) and model (EEV5, IEV5, ACCV1) are evaluated differently in the Variational Bayesian Analysis (VBA) even though our general linear model (GLM) treats them as equivalent. The divergence stems from the operational definition that $EEV5 \equiv EEV1 - ACCV1$ (the operational definition from Supplementary Figure S1):

$$EEV5: Value_{External\ exploration(5)} = \overline{Point}_{(all\ options)} - Point_{(the\ best\ option)}$$

$$EEV1: Value_{External\ exploration(1)} = \overline{Point}_{(all\ options)}$$

$$ACCV1: Value_{Accept(1)} = Point_{(the\ best\ option)}$$

In the GLM framework, this identity means the two design matrices span the same space, resulting in identical log-likelihoods, BIC values, and thus identical exceedance probabilities. VBA, however, imposes a modest complexity penalty on the redundant regressor in EEV5, which slightly reduces its average log-likelihood (model using EEV5: -210.29 versus model using EEV1: -209.71). Consequently, the model (EEV5, IEV5, ACCV1) receives a lower group-level exceedance probability, whereas the more parsimonious model (EEV1, IEV5, ACCV1) emerges as the preferred model.

We have inserted a brief note in the legend of Supplementary Figure S2 and hope this could clear up the confusion (Lines 82:88):

The same model showed the highest exceedance probability (the Best-fit model). This analysis corroborated our initial findings, with the same model emerging as the best-fit, thus strengthening our conclusions through converging evidence from multiple analytical approaches. **Note that the model with type 1 external exploration value, type 5 internal exploration value and type 1 accept value (EEV1, IEV5, ACCV1) and the model (EEV5, IEV5, ACCV1) are statistically equivalent under the GLM because $EEV5 \equiv EEV1 - ACCV1$ (Please see the operational definition from Supplementary Figure S1). In the VBA framework, a modest complexity penalty is applied to the redundant term in EEV5, resulting in a slightly lower mean log-likelihood (model using EEV5: -210.29 versus model using EEV1: -209.71). The more parsimonious specification therefore attains higher probability.**

Comment 3 (Part 1)

Concerning major comment 6:

- The model descriptions are still not good enough. For example, in Equation 1 there are 10 beta values listed and I understand that each model uses a subset of them? Still, it would be really valuable to have a Table with each mode and its parameters (i.e. betas).

Response

Thank you for pointing this out and we are sure that this reflects room for further improving the clarity of the corresponding section. In all our models, they use the full set of ten beta values, i.e. not a subset of them. Instead, each model differed by the operational definitions of the regressors that are entered to Equation 1. These definitions are listed in Supplementary Figure S1, while the linear form and number of coefficients remain the same across the models. We acknowledge that inconsistent terminology between the labels in Figure S1 (e.g. *internal exploration Value*) and the notation used in the text (e.g. $Value_{Internal\ exploration}$) may have caused confusion. To avoid it, we have revised Supplementary Figure S1, S2 and S3 to make the terms consistent. We have also prepared a table showing each model's parameters, as suggested by the reviewer.

Revised Supplementary Figure S1:

Revised Supplementary Figure S2:

Revised Supplementary Figure S3:

In addition, we have added a table listing the beta values for each model, together with a companion figure that defines the model numbering used in the table. Please note that model 81 is the best fit model and is highlighted in green.

Models 1-20:

Model	Model's parameters									
	β_1 Intercept	β_2 Value _{IE}	β_3 Value _{EE}	β_4 Value _{Acc}	β_5 Cost	β_6 Intercept	β_7 Value _{IE}	β_8 Value _{EE}	β_9 Value _{Acc}	β_{10} Cost
1	-2.837	-15.836*	0.825***	-1.875***	-0.217	0.135	-0.191	1.971***	-4.936***	-1.014***
2	-96.165	-693.27	-0.054	98.276	-0.153	0.397**	-0.424***	-0.247**	-0.445***	-0.706***
3	-2.661*	-10.192*	0.198*	-3.129***	-0.201	0.504***	-0.501***	-0.057	-1.025***	-0.813***
4	-3.722	-19.7	-0.122	-2.041***	-0.293**	0.638***	-0.359***	-0.273**	-0.667***	-0.728***
5	-3.124	-16.216*	0.392	-1.605***	-0.141	0.281	0.018	-0.255	-2.840***	-1.185***
6	-46.278	-317.83	-0.507***	37.117	-0.481***	0.092	-0.064	-2.602***	-0.201**	-1.751***
7	-5.586	-27.027	-0.145	-3.140*	-0.400**	0.14	-0.029	-2.537***	-0.389***	-1.769***
8	-3.025	-16.814	-0.797***	-1.886***	-0.729***	0.282	0.094	-2.591***	-0.666***	-1.858***
9	-0.606	-16.884*	-26.532	22.77	-0.251*	2.908	0.084	-29.255	23.526	-1.325***
10	-4.463*	-7.561*	-0.880***	-11.719**	-0.416***	0.221	0.105	-2.919***	-0.328***	-1.395***
11	-5.62	-28.541	-0.507*	-2.877*	-0.423***	0.254	0.166*	-2.802***	-0.399***	-1.434***
12	-2.942	-17.384	-1.086***	-1.867***	-0.578***	0.411	0.292***	-2.877***	-0.715***	-1.500***
13	-2.957	-16.010*	0.542***	-1.071***	-0.527***	0.132	-0.114	1.975***	-2.607***	-1.749***
14	-4.453**	-7.924*	0.632***	-11.111***	-0.471**	0.255	-0.345***	2.371***	-0.513***	-1.580***
15	-4.691	-22.597	0.608***	-2.637***	-0.551***	0.344	-0.349***	2.362***	-0.970***	-1.701***
16	-1.759	-10.167	0.852***	-1.890***	-0.642***	0.372*	-0.358**	2.445***	-0.480**	-1.643***
17	-2.837	-15.836*	1.137***	-0.579***	-0.217	0.135	-0.191	2.723***	-1.817***	-1.014***
18	-4.714*	-12.134	1.410***	-10.840***	-0.286*	0.448*	-0.328**	3.460***	-0.569***	-0.806***
19	-4.526	-23.147	1.198***	-2.702***	-0.364*	0.500*	-0.357**	3.374***	-0.732***	-0.879***
20	-1.382	-10.168	1.637***	-1.868***	-0.403**	0.543*	-0.308*	3.619***	-0.412*	-0.884***

* denotes $p < 0.05$; ** denotes $p < 0.01$; *** denotes $p \leq 0.001$.

Models 21-40:

Model	Model's parameters									
	β_1 Intercept	β_2 Value _{IE}	β_3 Value _{EE}	β_4 Value _{Acc}	β_5 Cost	β_6 Intercept	β_7 Value _{IE}	β_8 Value _{EE}	β_9 Value _{Acc}	β_{10} Cost
21	0.381	-1.620***	0.863***	-1.829***	-0.659***	0.164	-0.118	2.065***	-5.090***	-1.104***
22	-2.644**	-0.268	-0.149*	-9.613***	-0.566***	0.373**	-1.178***	-0.227*	0.586***	-0.948***
23	-6.923	-98.66	-0.027	85.41	-0.567***	0.392**	0.310**	-0.191*	-1.110***	-0.977***
24	-0.01	-0.546**	-0.138	-2.224***	-0.550***	0.702***	-0.569***	-0.223*	-0.627***	-0.955***
25	0.185	-1.546***	0.345	-1.556***	-0.648***	0.319	-0.087	-0.262	-2.925***	-1.291***
26	-2.182*	-0.35	-0.883***	-8.839***	-0.936***	0.137	-0.372***	-2.755***	0.123	-1.884***
27	-6.415	-95.817	-0.799***	82.786	-0.970***	0.158	0.054	-2.851***	-0.358**	-1.976***
28	0.04	-0.658**	-0.808***	-1.913***	-0.908***	0.276	-0.077	-2.608***	-0.626***	-1.889***
29	1.855	-1.543***	-13.282	10.504	-0.702***	2.524	-0.166	-24.866	19.53	-1.384***
30	-1.828	-0.214	-1.223***	-8.747***	-0.699***	0.288	-0.214	-3.099***	-0.135	-1.382***
31	-5.117	-80.81	-1.121***	69.556	-0.749***	0.346	0.147	-3.156***	-0.420*	-1.452***
32	0.243	-0.622**	-1.079***	-1.867***	-0.724***	0.394	-0.1	-2.818***	-0.592***	-1.459***
33	0.326	-1.609***	0.618***	-0.960***	-0.959***	0.162	-0.162	2.091***	-2.622***	-1.839***
34	-2.436*	-0.405	0.910***	-9.774***	-0.882***	0.329	-0.891***	2.535***	0.377**	-1.773***
35	-3.51	-50.182	0.925***	42.031	-0.936***	0.323	0.09	2.610***	-0.936***	-1.868***
36	0.084	-0.711**	0.844***	-1.881***	-0.868***	0.414*	-0.554***	2.447***	-0.336*	-1.807***
37	0.381	-1.620***	1.191***	-0.471**	-0.659***	0.164	-0.118	2.855***	-1.823***	-1.104***
38	-2.112*	-0.237	1.602***	-10.328***	-0.537***	0.496*	-0.515**	3.595***	-0.007	-0.865***
39	-4.31	-68.371	1.517***	58.151	-0.588***	0.518*	0.187	3.627***	-0.773***	-0.925***
40	0.443	-0.596**	1.591***	-1.847***	-0.593***	0.557*	-0.395***	3.583***	-0.27	-0.985***

* denotes $p < 0.05$; ** denotes $p < 0.01$; *** denotes $p \leq 0.001$.

Models 41-60:

Model	Model's parameters									
	β_1 Intercept	β_2 Value _{IE}	β_3 Value _{EE}	β_4 Value _{Acc}	β_5 Cost	β_6 Intercept	β_7 Value _{IE}	β_8 Value _{EE}	β_9 Value _{Acc}	β_{10} Cost
41	0.272	1.808***	0.731***	-1.998***	-0.777***	0.21	0.469*	2.063***	-4.925***	-1.267***
42	-2.087	1.447***	-0.314***	-5.482*	-0.587***	0.365**	0.1	-0.332***	-0.412***	-0.911***
43	-0.745	1.411***	-0.197*	-1.464***	-0.625***	0.367**	-0.148	-0.172	-0.964***	-0.996***
44	-0.034	0.693**	-0.268**	-2.017***	-0.496***	0.825***	-0.477***	-0.286**	-1.522***	-0.819***
45	0.043	1.829***	0.820**	-2.157***	-0.581***	0.385	0.493***	-0.186	-2.700***	-1.362***
46	-2.372	1.403***	-0.900***	-7.045*	-0.996***	0.139	0.374*	-2.662***	-0.029	-1.945***
47	-0.658	1.418***	-0.769***	-1.724***	-0.983***	0.166	0.328*	-2.628***	-0.157	-1.967***
48	0.027	0.726**	-0.913***	-1.605***	-0.926***	0.348	-0.152	-2.568***	-0.926***	-1.889***
49	1.034	1.792***	-5.408	3.069	-0.777***	1.984	0.482***	-18.582	14.059	-1.497***
50	-1.841	1.452***	-1.239***	-6.382*	-0.796***	0.338	0.420**	-2.944***	-0.106	-1.504***
51	-0.292	1.483***	-1.116***	-1.515**	-0.813***	0.363	0.383**	-2.911***	-0.165*	-1.530***
52	0.235	0.902***	-1.213***	-1.372***	-0.743***	0.449	0.02	-2.863***	-0.740**	-1.492***
53	0.18	1.783***	0.365**	-1.299***	-0.967***	0.183	0.487**	1.970***	-2.493***	-1.935***
54	-1.867	1.225***	0.719***	-6.008**	-0.818***	0.191	-0.066	2.446***	-0.548***	-1.750***
55	-0.494	1.203***	0.716***	-1.686***	-0.879***	0.212	-0.281*	2.457***	-1.041***	-1.838***
56	0.073	0.519*	0.823***	-1.820***	-0.800***	0.533**	-0.662***	2.477***	-1.351***	-1.712***
57	0.272	1.808***	1.003***	-0.849***	-0.777***	0.21	0.469*	2.844***	-1.668***	-1.267***
58	-1.418	1.329***	1.574***	-6.175**	-0.603***	0.479*	0.074	3.622***	-0.490***	-0.956***
59	-0.064	1.333***	1.465***	-1.649***	-0.660***	0.486*	-0.075	3.555***	-0.748***	-1.013***
60	0.429	0.710**	1.628***	-1.558***	-0.558***	0.644**	-0.273	3.654***	-0.822***	-0.940***

* denotes $p < 0.05$; ** denotes $p < 0.01$; *** denotes $p \leq 0.001$.

Models 61-80

Model	Model's parameters									
	β_1 Intercept	β_2 Value _{IE}	β_3 Value _{EE}	β_4 Value _{Acc}	β_5 Cost	β_6 Intercept	β_7 Value _{IE}	β_8 Value _{EE}	β_9 Value _{Acc}	β_{10} Cost
61	0.404	2.098***	0.774***	-2.687***	-0.795***	0.036	0.172	2.155***	-5.067***	-1.197***
62	-1.895	1.477***	-0.585***	-4.923*	-0.577***	-0.036	-0.787***	-0.239**	-0.614***	-0.809***
63	-0.725	1.389***	-0.441***	-1.565***	-0.624***	-0.069	-1.116***	-0.011	-1.161***	-0.924***
64	0.137	-0.521	-0.275**	-3.146***	-0.519***	0.15	-5.337***	0.351***	-5.218***	-0.702***
65	0.235	2.157***	0.09	-2.149***	-0.794***	0.252	0.338	-0.279	-2.703***	-1.360***
66	-1.754	1.573***	-1.485***	-5.665*	-1.166***	-0.009	0.1	-2.722***	-0.095	-1.898***
67	-0.423	1.587***	-1.374***	-1.494***	-1.160***	0.002	0.002	-2.644***	-0.238**	-1.908***
68	0.179	0.563*	-1.203***	-1.754***	-1.037***	0.225	-1.066**	-2.335***	-1.455***	-1.788***
69	2.184	2.106***	-17.077	13.156	-0.812***	2.692	0.34	-27.232	21.848	-1.436***
70	-1.391	1.687***	-1.758***	-5.343*	-0.829***	0.207	0.251	-2.987***	-0.172**	-1.440***
71	-0.108	1.728***	-1.663***	-1.320**	-0.840***	0.213	0.187	-2.931***	-0.214**	-1.461***
72	0.324	1.110***	-1.589***	-1.126***	-0.776***	0.276	-0.308	-2.814***	-0.709**	-1.426***
73	0.326	2.100***	0.528**	-1.891***	-1.038***	0.01	0.316	2.055***	-2.582***	-1.900***
74	-1.929	1.102***	0.996***	-6.309*	-0.845***	-0.142	-0.847**	2.490***	-0.710***	-1.660***
75	-0.497	1.090***	1.001***	-1.971***	-0.924***	-0.137	-1.059***	2.487***	-1.129***	-1.766***
76	0.253	-0.458	0.889***	-2.679***	-0.858***	0.095	-3.486***	2.374***	-3.378***	-1.676***
77	0.404	2.098***	1.067***	-1.464***	-0.795***	0.036	0.172	2.970***	-1.664***	-1.197***
78	-1.358	1.296***	1.965***	-6.122*	-0.530***	0.227	-0.502*	3.702***	-0.659***	-0.887***
79	-0.023	1.296***	1.836***	-1.838***	-0.593***	0.208	-0.655**	3.614***	-0.836***	-0.964***
80	0.534	0.058	1.690***	-2.096***	-0.567***	0.265	-2.406***	3.457***	-2.354***	-0.959***

* denotes $p < 0.05$; ** denotes $p < 0.01$; *** denotes $p \leq 0.001$.

Model 81-100:

	Model's parameters									
Model	β_1 Intercept	β_2 Value _{IE}	β_3 Value _{EE}	β_4 Value _{Acc}	β_5 Cost	β_6 Intercept	β_7 Value _{IE}	β_8 Value _{EE}	β_9 Value _{Acc}	β_{10} Cost
81	0.079	2.558***	0.784***	-3.540***	-0.819***	0.087	0.450*	2.125***	-5.193***	-1.222***
82	-2.691	1.234***	-0.678***	-6.806*	-0.537***	-0.075	-1.026***	-0.086	-0.814***	-0.790***
83	-1.086*	1.179***	-0.493***	-2.227***	-0.596***	-0.026	-1.278***	0.18	-1.285***	-0.916***
84	0.241	-0.321	-0.276*	-3.302***	-0.549***	0.710**	-3.404***	0.439***	-4.183***	-0.759***
85	-0.091	2.522***	0.188	-3.011***	-0.767***	0.27	0.480**	-0.219	-2.859***	-1.338***
86	-2.399	1.549***	-1.766***	-6.909*	-1.204***	0.001	0.07	-2.665***	-0.108	-1.853***
87	-0.747	1.570***	-1.635***	-1.860***	-1.195***	0.028	-0.03	-2.556***	-0.280***	-1.855***
88	0.106	0.625**	-1.368***	-1.823***	-1.048***	0.338	-0.980***	-2.161***	-1.511***	-1.726***
89	1.874	2.472***	-17.403	12.69	-0.817***	2.679	0.477**	-26.97	21.488	-1.438***
90	-1.938	1.877***	-2.259***	-6.038	-0.821***	0.217	0.385*	-3.119***	-0.095	-1.420***
91	-0.415	1.917***	-2.157***	-1.468**	-0.830***	0.23	0.332*	-3.056***	-0.149*	-1.436***
92	0.14	1.572***	-2.159***	-0.735**	-0.798***	0.263	0.138	-3.055***	-0.303	-1.442***
93	-0.013	2.515***	0.490**	-2.707***	-1.035***	0.032	0.519*	2.027***	-2.727***	-1.894***
94	-2.621	0.803***	0.956***	-8.263*	-0.788***	-0.115	-0.970***	2.456***	-0.851***	-1.628***
95	-0.791	0.886***	0.967***	-2.593***	-0.887***	-0.065	-1.061***	2.460***	-1.139***	-1.740***
96	0.31	-0.355*	0.885***	-2.688***	-0.858***	0.378	-2.529***	2.363***	-2.834***	-1.693***
97	0.079	2.558***	1.081***	-2.302***	-0.819***	0.087	0.450*	2.929***	-1.837***	-1.222***
98	-1.981	1.159***	2.074***	-7.695*	-0.477***	0.259	-0.441*	3.653***	-0.708***	-0.888***
99	-0.305	1.217***	1.943***	-2.401***	-0.562***	0.261	-0.521**	3.589***	-0.867***	-0.978***
100	0.49	0.259	1.796***	-1.969***	-0.558***	0.408	-1.527***	3.455***	-1.779***	-0.996***

* denotes $p < 0.05$; ** denotes $p < 0.01$; *** denotes $p \leq 0.001$.

Model 101-120:

Model	Model's parameters									
	β_1 Intercept	β_2 Value _{IE}	β_3 Value _{EE}	β_4 Value _{ACC}	β_5 Cost	β_6 Intercept	β_7 Value _{IE}	β_8 Value _{EE}	β_9 Value _{ACC}	β_{10} Cost
101	0.036	2.464***	0.696***	-3.459***	-0.703***	0.106	0.653**	2.004***	-5.180***	-1.226***
102	-2.96	1.186***	-0.677***	-7.473*	-0.487***	0.004	-0.863***	-0.112	-0.779***	-0.826***
103	-1.163*	1.144***	-0.497***	-2.329***	-0.549***	0.056	-1.103***	0.149	-1.261***	-0.958***
104	0.145	-0.117	-0.318*	-3.123***	-0.527***	0.812**	-2.532***	0.291**	-3.444***	-0.838***
105	-0.154	2.450***	0.605*	-3.411***	-0.559***	0.304	0.606***	-0.174	-2.958***	-1.304***
106	-2.661	1.438***	-1.679***	-7.706*	-1.095***	0.043	0.193	-2.710***	-0.074	-1.863***
107	-0.823	1.459***	-1.529***	-2.095***	-1.087***	0.072	0.107	-2.613***	-0.243**	-1.871***
108	0.041	0.674**	-1.363***	-1.748***	-0.998***	0.338	-0.507*	-2.334***	-1.131***	-1.796***
109	1.293	2.402***	-10.563	6.348	-0.690***	2.356	0.600***	-23.333	18.113	-1.434***
110	-2.199	1.801***	-2.245***	-6.792	-0.729***	0.247	0.540**	-3.224***	-0.043	-1.413***
111	-0.495	1.842***	-2.126***	-1.663**	-0.739***	0.264	0.503**	-3.170***	-0.091	-1.429***
112	0.135	1.381***	-2.056***	-0.982***	-0.724***	0.311	0.351	-3.121***	-0.257	-1.447***
113	-0.057	2.420***	0.366*	-2.732***	-0.871***	0.06	0.694**	1.945***	-2.820***	-1.856***
114	-2.829	0.742***	0.884***	-8.902*	-0.724***	-0.023	-0.792***	2.431***	-0.820***	-1.655***
115	-0.852	0.850***	0.900***	-2.708***	-0.822***	0.031	-0.871***	2.427***	-1.114***	-1.766***
116	0.23	-0.151	0.887***	-2.501***	-0.820***	0.479*	-1.851***	2.362***	-2.308***	-1.736***
117	0.036	2.464***	0.957***	-2.364***	-0.703***	0.106	0.653**	2.760***	-2.018***	-1.226***
118	-2.234	1.073***	1.948***	-8.479*	-0.424**	0.309	-0.296	3.562***	-0.662***	-0.898***
119	-0.387	1.170***	1.843***	-2.584***	-0.507***	0.326	-0.350*	3.515***	-0.840***	-0.990***
120	0.457	0.380**	1.829***	-1.896***	-0.509***	0.513*	-0.969**	3.506***	-1.373***	-1.008***

* denotes $p < 0.05$; ** denotes $p < 0.01$; *** denotes $p \leq 0.001$.

We hope the table could help avoid the confusion. We will include it in the Supplementary Information if the reviewer believes it would strengthen the manuscript.

Comment 3 (Part 2)

- Your multinomial approach to modeling behavior is highly interesting and other researchers may profit from understanding it better. Currently, your description of the modeling presumes that the reader is familiar with the approach, which most are not (I asked around). I would insist that you elaborate on the whole process, and describes it more clearly. For example, why does one decision need to be used as a ‘reference’?

Response

Thank you for your suggestion. We have expanded the Methods section to give a clearer, step-by-step account of our multinomial-logistic model (Methods, Lines 548–555):

...the choice probability of accept, internal exploration, and external exploration decisions, respectively. With three choice probabilities that sum to one, only two independent comparisons are identifiable that a reference category is fixed for estimation. We set accept as the reference, which provides a non-exploratory baseline common to both exploratory branches. The model estimates two log-odds contrasts (internal exploration versus accept and external exploration versus accept). Coefficients indicate how a one-unit increase in a predictor changes the odds of selecting the corresponding exploratory action rather than accepting. The choice of reference does not affect fitted probabilities or overall model fit; it only alters coefficient labeling and interpretation.

We have also revised the corresponding section of the Results section (Lines 154:156):

Next, to illustrate that participants used internal exploration, external exploration and accept values to guide their choices, we performed a multinomial logistic regression. This approach estimated the effects of these value terms on the odds-ratio of internal exploration versus accept decisions and their effects on external exploration versus accept decisions in a single analysis. The results suggested that the choice ratio between internal exploration and accept ...”

We hope these additions make the modelling procedures clearer.

Comment 4

Concerning major comment 9:

- Using a contrast to identify brain regions has a big disadvantage in that a significant contrast $A > B$ may be due to either A being more activated than B, or because B is more de-activated than A. Because you look at correlations, it means that significant $A > B$ contrast may wrongly be interpreted as A showing a positive correlation with parameter X, while in reality it may be that it shows no correlation, and B instead shows a strong negative

correlation with X. It is therefore critical to show that the correlation between A and X is also larger than zero.

Response

Thank you for your comment. In our whole-brain GLM, the three value regressors share a common intercept, which without sum-to-zero coding would render the design matrix rank deficient and the parameters non-identifiable. We therefore apply sum-to-zero contrasts ([+1 -0.5 -0.5]) that compare one condition to the average of the other two while remaining orthogonal to the intercept. Each regressor is z-scored within participant, placing all β -weights on a common scale.

To resolve the activation-vs-deactivation ambiguity of A>B contrasts, we verified the results by the ROI time-course analyses (Fig. 4b), without reusing contrast-defined voxels to avoid double dipping. Value signals were positive for the corresponding, highlighted regions (e.g., IPS internal-exploration: $t_{23} = 3.275, p = .003$; ACC external-exploration: multiple positive peaks, $t_{23} \geq 2.074, ps \leq .049$; mPFC accept: positive peaks, $t_{23} \geq 3.337, ps = .003$), indicating that the whole-brain contrasts reflect positive tracking in the target ROIs rather than deactivation in the comparison conditions.

Fig. 4b:

We have revised the manuscript accordingly and hope these additions make the whole-brain analysis procedures and the results clearer.

Manuscript (Line 223:226):

First, it illustrates the dynamics of the signals across time without the need of assuming the shape of their waveforms (as in typical whole-brain analyses). Second, because it is free from any waveform assumptions, it allows identification of signals that deviate from a canonical haemodynamic response function. **Third, it verifies that the whole-brain contrasts in Fig. 3 reflect positive value signals in the target regions rather than relative deactivation in the compared conditions (see Fig. 3 for contrast definitions and Fig. 4b for independent verification).**

Manuscript (Line 286:288):

The mPFC showed a positive external exploration value signal, a positive accept value signal, and a negative internal exploration value signal. **Together, these positive signals verified that the whole-brain contrasts (Fig. 3) reflect positive tracking in the highlighted ROIs (IPS: internal exploration; ACC: external exploration; mPFC: Accept), rather than deactivation in the comparison conditions.**

Comment 5

Concerning major 10:

- The ANOVA is ok, but only shows that there are differences somewhere in your data. To find out where these differences are, you need to run post-hoc tests, which need to be corrected for multiple comparisons. This is a fundamental statistical approach which is highly relevant for your data, given all the uncontrolled comparisons made in Figure 5. Without the proper corrections, these results are not convincing at all.

Response

Thank you for your comment. In addition to the three-way ANOVA, we now report FDR-corrected post-hoc pairwise tests that localize the effects. These analyses are provided in Supplementary Fig. S9. We have also revised the manuscript to highlight this additional analysis.

Manuscript (Lines 349:350):

To confirm that the three brain regions involved a different code, we performed a three-way ANOVA with factors of Brain Regions (IPS, ACC and mPFC), Decision (internal exploration, external exploration and accept decisions) and Signal type (internal exploration, external exploration and accept signals), and the results showed a significant

three-way interaction ($F(8, 621) = 5.26, p < 0.001$; full False Discovery Rate (FDR)-corrected post-hoc contrasts and for each Region \times Decision \times Signal cell are provided in Supplementary Fig. S9).

Supplementary Figure S9 (Lines 179:205):

A

B

Supplementary Figure S9. Post-hoc contrasts by Region × Decision × Signal (False Discovery Rate (FDR) -corrected). (A). We first examined the internal exploration value signal in IPS (left panel). During internal exploration decisions, the internal exploration signal was greater in IPS than ACC (IPS: $Mean = 0.103$, $SD = 0.225$; ACC: $Mean = -0.323$, $SD = 0.605$; post-hoc FDR-corrected $p = 0.0013$). During external exploration decisions, the internal exploration signal was greater in IPS ($Mean = 0.080$, $SD = 0.171$) than ACC ($Mean = -0.058$, $SD = 0.178$, $p = 0.013$) and mPFC (ACC: $Mean = -0.089$, $SD = 0.162$, $p = 0.045$). During accept exploration decisions, the internal exploration signal was greater in IPS ($Mean = 0.075$, $SD = 0.128$) than ACC ($Mean = -0.050$, $SD = 0.138$, $p = 0.036$) and mPFC ($Mean = -0.091$, $SD = 0.133$, $p = 0.002$). Second, we examined the external exploration value signal with a focus on ACC (middle panel). During internal exploration decisions, the external exploration signal was greater in ACC ($Mean = 0.209$, $SD = 0.327$) than mPFC ($Mean = -0.070$, $SD = 0.196$, $p = 0.002$). During accept exploration decisions, the external exploration signal was greater in ACC (ACC: $Mean = 0.058$, $SD = 0.136$) than IPS ($Mean = -0.059$, $SD = 0.115$, $p = 0.007$) and mPFC ($Mean = -0.056$, $SD = 0.140$, $p = 0.007$). (B). Third, we examined the value signals in mPFC (right panel). During internal exploration decisions, the internal exploration signal ($Mean = 0.166$, $SD = 0.305$) was greater than the external exploration signal ($Mean = -0.070$, $SD = 0.196$; $p = 0.0078$). During external exploration decisions, the external exploration signal ($Mean = 0.071$, $SD = 0.121$) was greater than the internal exploration signal (IE: $Mean = -0.089$, $SD = 0.162$; $p = 0.0073$). During accept decisions, the accept signal ($Mean = 0.179$, $SD = 0.213$) was greater than the internal exploration signal ($Mean = -0.091$, $SD = 0.133$; $p < 0.001$) and the external exploration signal (EE: $Mean = 0.043$, $SD = 0.213$; $p < 0.001$). Taken together, these post-hoc contrasts strengthen the role in exploratory decisions across regions. IPS generally encodes the value of internal exploration regardless the nature of the decisions, while ACC encodes the value of external exploration, and mPFC shows decision general coding. * denotes $p < 0.05$; ** denotes $p < 0.01$; *** denotes $p \leq 0.001$. Error bars represent \pm SEM.

Responses to the reviews

Distinct contributions of prefrontal, parietal, and cingulate signals to exploratory decisions

Victor K. S. Chan, Nicole H. L. Wong, Tsz-Fung Woo, Kei Watanabe, Masahiko Haruno, Chun-Kit Law, Bolton K. H. Chau

Reviewer #1

Comment 1

I find the paper much better and clearer ! Thank you for thoroughly answering all of the comments with so much attention.

I have 3 final minor suggestion to help with clarity but do not require any further analysis or explanation:

1. you say the whole brain analysis was time locked at stimulus onset, given the complexity of task please specify which stimulus onset with more precision line 596: 599:

a. "time-locked to each decision phase stimulus onset (or offset following a internal exploration decision) ” or "time-locked to stimulus onset at the beginning of each trial ”.

Response

Thank you for the suggestion to clarify the timing reference in our whole-brain analysis. We now specify that all regressors were time-locked to the stimulus onset at the beginning of each trial (Lines 603-606):

...We entered the internal exploration value, external exploration value, accept value, cumulative cost that were time-locked to stimulus onset **at the beginning of each trial** as regressors in the GLM, convolved with a canonical hemodynamic function (two-gamma model)⁶¹ to provide idealised hemodynamic responses.

Comment 2 (Part 1)

2. I still find figure 5 and its description disorganized/lacking in consistency about the results that you report (i.e. some significant results are not discussed).

a. Explaining the reasoning behind the trials selection in the ACC/exploratory signal analysis at the beginning of that paragraph would help with clarity. (Unless it was only done for the exploratory signal analysis in which case this should be made a lot more clearer and the negative internal exploration signal during an internal exploration decision should

not be reported at the same time as the positive external exploration signal during an internal exploration decision as they are not done on the same trials) also being consistent in order in which you report really helps (internal, external accept. And consistently specify signal or decision (and not switching terms half way to accept value for example)

Response

Thank you for the suggestions on clarity and consistency. We have now revised manuscript (Lines 311-321) to state the trial-selection logic in general:

...To test whether each brain region signals exploration value or general decision value, next we ran time course analyses **within each trial subset defined by the impending decision (i.e., the internal exploration, external exploration or accept decision to be executed), and the three value regressors (internal exploration, external exploration and accept values) were entered simultaneously. Thus, all effects were reported from the same trials in that decision context. We then extracted the peak signal for each of the three decision types and carried out three one sample t tests against zero. p values reported for Fig. 5 are false discovery rate corrected (FDR-adjusted p; FDR procedure described in the Multiple-comparisons section in Methods).**

...**In line with the literature which predominantly reports positive scaling of value signals ^{27,35,40-42}, we focus on positive signals here to facilitate visualization of the signal patterns and cross-ROI comparisons. While all negative signals are reported in Supplementary Fig. S8.**

References:

27. Levy, D. J. & Glimcher, P. W. The root of all value: A neural common currency for choice. *Curr Opin Neurobiol* **22**, 1027–1038 (2012).
35. Law, C. K., Kolling, N., Chan, C. C. H. & Chau, B. K. H. Frontopolar cortex represents complex features and decision value during choice between environments. *Cell Rep* **42**, (2023).
40. Boorman, E. D., Behrens, T. E. J., Woolrich, M. W. & Rushworth, M. F. S. How Green Is the Grass on the Other Side? Frontopolar Cortex and the Evidence in Favor of Alternative Courses of Action. *Neuron* **62**, 733–743 (2009).
41. Bartra, O., McGuire, J. T. & Kable, J. W. The valuation system: A coordinate-based meta-analysis of BOLD fMRI experiments examining neural correlates of subjective value. *Neuroimage* **76**, 412–427 (2013).
42. Shadlen, M. N. & Newsome, W. T. Neural Basis of a Perceptual Decision in the Parietal Cortex (Area LIP) of the Rhesus Monkey. *J Neurophysiol* **86**, 1916–1936 (2001).

manuscript (Lines 342-349) to state the trial-selection logic in the ACC signal:

... Previous studies suggest that when individuals repeated the same exploitative decision before switching to an exploratory decision, the ACC signal also ramps up gradually^{7,24}. In our study, we also noticed that participants often made internal exploration decisions repeatedly with the same option. Hence, **we ran an additional analysis while the trials were defined by leaving** out the first occasion when participants made an internal exploration decision with an option, **and** we hypothesized that the ACC signal should be more robust. Indeed, we found a significant external exploration signal (Fig. 5b, middle panel; $\beta = 0.209$, $t_{23} = 3.135$, **FDR-adjusted $p = 0.014$**).

manuscript (Lines 358-366) to keep reporting order consistent (internal, external, accept):

... **mPFC showed general decision signal, instead of accept signal**. Interestingly, during internal exploration, the mPFC showed a significant internal exploration signal (Fig. 5c, top panel; $\beta = 0.166$, $t_{23} = 2.666$, FDR-adjusted $p = 0.014$). Furthermore, during external exploration, the mPFC showed external exploration signal (Fig. 5c, middle panel; $\beta = 0.071$, $t_{23} = 2.872$, FDR-adjusted $p = 0.026$). Based on the results in Figures 2 and 3, we expected that the mPFC should also pass this test by showing an accept signal in all three types of decision. In the mPFC, we could only find positive accept signals during internal exploration decisions ($\beta = 0.155$, $t_{23} = 2.838$, FDR-adjusted $p = 0.014$), and accept decisions (Fig. 5c, bottom panel; $\beta = 0.179$, $t_{23} = 4.133$, FDR-adjusted $p = 0.001$).

Supplementary Figure S8 (Lines 146-172) to keep reporting order consistent (internal, external, accept):

... **(b)**. The ACC showed a negative internal exploration signal in ACC ($t_{23} = -2.613$, $p = 0.016$), suggesting that the ACC compares the relative advantage or disadvantage of external exploration as opposed to the impending internal exploration. Critically, the ACC showed external exploration signal regardless of the type of impending decision, there was a positive external exploration signal when participants made external exploration (middle panel; peaked at 1.22s: $t_{23} = 2.094$, $p = 0.047$, peaked at 7.39s: $t_{23} = 2.280$, $p = 0.032$, peaked at 11.83s: $t_{23} = 3.374$, $p = 0.003$) or accept decisions (bottom panel; $t_{23} = 2.102$, $p = 0.047$). **Inset (top panel)**: During internal exploration, participants often repeated selections of the same option. When all internal exploration decision in each repetition were analyzed, the external exploration signal was initially not significant ($t_{23} = 1.555$, $p = 0.134$), after the first decision in each repetition was removed, and only the second and later decisions were analysed, the external exploration signal became significant (top panel; $t_{23} = 3.149$, $p = 0.004$). **(c)**. The mPFC showed general decision signal. During internal exploration, there were positive accept signals (peaked at -0.43s: $t_{23} = 2.474$, $p = 0.021$; peaked at 3.51s: $t_{23} = 2.838$, $p = 0.009$), after that, the mPFC switched to encode an internal exploration signal (top panel; $t_{23} = 2.666$, $p = 0.014$). In

our task, options worth internal exploration are typically high in both uncertainty and average value. This pattern suggests that such options are often evaluated as accept decisions at first, but consideration of their uncertainty leads participants to withhold accept decisions and switch to internal exploration. During external exploration, the mPFC showed external exploration signal (middle panel; peaked at 7.65s: $t_{23} = 3.007$, $p = 0.006$, peaked at 11.85s: $t_{23} = 3.741$, $p = 0.001$), there were also negative internal exploration signals (peaked at -1.05s: $t_{23} = -2.536$, $p = 0.018$; peaked at 3.82s: $t_{23} = 2.671$, $p = 0.014$; peaked at 8.04s: $t_{23} = 2.684$, $p = 0.013$). During accept decisions, the mPFC reflected a positive accept signal (bottom panel; $t_{23} = 4.205$, $p > 0.001$). Besides, there was a negative internal exploration signal ($t_{23} = -3.373$, $p = 0.003$), consistent with a value-difference code in which mPFC emphasises the value of the current decision (accept) relative to its alternative (internal exploration).

Comment 2 (Part 2)

b. similarly, when reporting the mPFC effect specify the direction of the effect line 359 "and a positive accept signal during internal exploration decisions" and you need to discuss or explain why don't you mention the negative significant effect of internal exploration signal in the mPFC for external decision and accept decision?.

Response

Thank you for the suggestion regarding clarity in reporting the mPFC effects. We have revised the manuscript to state the mPFC effects wherever they are reported.

Manuscript (Lines 319-321):

...In line with the literature which predominantly reports positive scaling of value signals ^{27,35,40-42}, we focus on positive signals here to facilitate visualization of the signal patterns and cross-ROI comparisons. While all negative signals are reported in Supplementary Fig. S8.

Supplementary Figure S8 (Lines 166-172):

... there were also negative internal exploration signals (peaked at -1.05s: $t_{23} = -2.536$, $p = 0.018$; peaked at 3.82s: $t_{23} = 2.671$, $p = 0.014$; peaked at 8.04s: $t_{23} = 2.684$, $p = 0.013$). During accept decisions, the mPFC reflected a positive accept signal (bottom panel; $t_{23} = 4.205$, $p > 0.001$). Besides, there was a negative internal exploration signal ($t_{23} = -3.373$, $p = 0.003$), consistent with a value-difference code in which mPFC emphasises the value of the current decision (accept) relative to its alternative (internal exploration).

Comment 2 (Part 3)

c. finally, stay consistent with the vocabulary that you use rather than use a new term (use accept signal or accept decision rather than accept "value"). "mPFC signalled external exploration value signal [...]" (Fig. 5c, middle panel; $\beta = 0.071$, $t_{23} = 2.872$, $p = 0.009$), but not accept value"

Response

Thank you so much for your suggestion. We have now revised the manuscript and supplementary Fig. S8 to stay consistent with the terms we used.

Manuscript (Lines 231-233):

... In contrast, there was an absence of an external exploration signal (Fig. 4b, left panel; $t_{23} = 1.858$, $p = 0.076$) and an absence of an accept **signal** (Fig. 4b, left panel; $t_{23} = -1.463$, $p = 0.157$).

Manuscript (Lines 265-269):

... Finally, a between-participant analysis showed that those with stronger accept **signal** had a greater proportion of accept decisions (Fig. 4c, right panel; $r = 0.618$, $p = 0.022$), those with stronger internal exploration **signals** also had greater proportions of internal exploration decisions (Fig. 4c, right panel; $r = 0.431$, $p = 0.045$).

Manuscript (Lines 280-284):

...The IPS showed a positive internal exploration **signal**, but the absence of both an external exploration **signal** and an accept **signal**. The ACC showed a positive external exploration **signal**, a negative internal exploration **signal** and an absence of accept **signal**. The mPFC showed a positive external exploration **signal**, a positive accept **signal**, and a negative internal exploration **signal**.

Manuscript (Lines 289-290):

... Individuals showing stronger mPFC accept **signal** tended to make more accept decisions...

Manuscript (Lines 358-366):

...**mPFC showed general decision signal, instead of accept signal**. Interestingly, during internal exploration, the mPFC showed a significant internal exploration signal (Fig. 5c, top panel; $\beta = 0.166$, $t_{23} = 2.666$, $p = 0.014$). Furthermore, during external exploration, the mPFC showed external exploration signal (Fig. 5c, middle panel; $\beta = 0.071$, $t_{23} = 2.872$, $p = 0.009$). Based on the results in Figures 2 and 3, we expected that the mPFC should also pass this test by showing an accept signal in all three types of decision. In the mPFC, we could only find positive accept signals during internal exploration decisions ($\beta = 0.155$, t_{23}

= 2.838, $p = 0.009$), and accept decisions (Fig. 5c, bottom panel; $\beta = 0.179$, $t_{23} = 4.133$, $p < 0.001$).

Supplementary Figure S8 (Lines 140-141; 146-172):

... The IPS showed internal exploration signals independent of the impending decision...

... **(b)**. The ACC showed a negative internal exploration signal ($t_{23} = -2.613$, $p = 0.016$), suggesting that the ACC compares the relative advantage or disadvantage of external exploration as opposed to the impending internal exploration. Critically, the ACC showed external exploration signal regardless of the type of impending decision, there was a positive external exploration signal when participants made external exploration (middle panel; peaked at 1.22s: $t_{23} = 2.094$, $p = 0.047$, peaked at 7.39s: $t_{23} = 2.280$, $p = 0.032$, peaked at 11.83s: $t_{23} = 3.374$, $p = 0.003$) or accept decisions (bottom panel; $t_{23} = 2.102$, $p = 0.047$). **Inset (top panel)**: During internal exploration, participants often repeated selections of the same option. When all internal exploration decision in each repetition were analyzed, the external exploration signal was initially not significant ($t_{23} = 1.555$, $p = 0.134$), after the first decision in each repetition was removed, and only the second and later decisions were analysed, the external exploration signal became significant (top panel; $t_{23} = 3.149$, $p = 0.004$). **(c)**. The mPFC showed general decision signal. During internal exploration, there were positive accept signals (peaked at -0.43s: $t_{23} = 2.474$, $p = 0.021$; peaked at 3.51s: $t_{23} = 2.838$, $p = 0.009$), after that, the mPFC switched to encode an internal exploration signal (top panel; $t_{23} = 2.666$, $p = 0.014$). In our task, options worth internal exploration are typically high in both uncertainty and average value. This pattern suggests that such options are often evaluated as accept decisions at first, but consideration of their uncertainty leads participants to withhold accept decisions and switch to internal exploration. During external exploration, the mPFC showed external exploration signal (middle panel; peaked at 7.65s: $t_{23} = 3.007$, $p = 0.006$, peaked at 11.85s: $t_{23} = 3.741$, $p = 0.001$), there were also negative internal exploration signals (peaked at -1.05s: $t_{23} = -2.536$, $p = 0.018$; peaked at 3.82s: $t_{23} = 2.671$, $p = 0.014$; peaked at 8.04s: $t_{23} = 2.684$, $p = 0.013$). During accept decisions, the mPFC reflected a positive accept signal (bottom panel; $t_{23} = 4.205$, $p > 0.001$). Besides, there was a negative internal exploration signal ($t_{23} = -3.373$, $p = 0.003$), consistent with a value-difference code in which mPFC emphasises the value of the current decision (accept) relative to its alternative (internal exploration).

Supplementary Figure S9 (Lines 181; 189-190; 194-195):

... We first examined the internal exploration signal in IPS (left panel).

... Second, we examined the external exploration signal with a focus on ACC (middle panel).

... Third, we examined the **signals** in mPFC (right panel).

Comment 3

4. Supp Figure S9: please make sure that the * are not overlapping each other for visibility. If their size difference is not meaningful, please make them all the same size. if it is, please explain the difference.

Response

Thank you so much for your suggestion. We have now revised Supplementary Figure S9 to keep the * are not overlapping each other and the sizes are the same.

Revised Supplementary Figure S9:

Reviewer #2

Comment 1 (Part 1)

I sincerely thank the authors for their work on my behalf. Many concerns have been removed, except for a minor one and a (potentially) major one.

Minor 1: Thank you for clarifying the modeling. However, now that I understand the procedure better, I realize that you are forcing the models to contain all your terms (i.e. each model contains IEV + EEV + ACCV terms), while less complex models (e.g. IEV only, EEV + ACCV, IEV + EEV, ...) are left out of the analyses and model comparisons. While I would like to see an analysis of the full model space to make sure simpler models are not more parsimonious (which is usually conducted), this limitation should at the very least be mentioned in the manuscript.

Response

Thank you for raising this point. We have run these analyses. Specifically, we compared the full model that includes Internal exploration value (In/IEV), External exploration value (Ex/EEV), Accept value (Ac/ACCV), and the cost against all parsimonious models (e.g., In/Ex/Ac only; Ex + Ac; In + Ex; etc.). Across participants, there were significant differences among models in both BIC and predictive performance (BIC: $F(12, 311) = 9.242, p < .001$; prediction accuracy: $F(12, 311) = 36.606, p < .001$). Critically, the winning model (the “red bars” in our figure) outperformed every alternative in BIC (all $t_{s23} < -6.241$, all $ps < .001$) and in prediction accuracy (all $t_{s23} > 6.260$, all $ps < .001$). These results support our model selection: despite higher complexity, the full model provides the best account of behavior and justifies retaining all three value parameters. We are happy to preform extra analyses If we have missed any specific variant you had in mind. Thank you.

Comment 1 (Part 2)

Major 1: Concerning the multiple comparisons issue, thank you for adding Supp. Fig 9. However, I was mainly concerned about the 27 uncorrected tests reported in Fig. 5. I see only 7 tests with p -values $< .01$, which makes me suspicious of the correction method used. In my experience, using FDR-correction methods for so many tests (with low uncorrected significance) is unlikely to yield so many reported significant results. Until FDR-correction has been applied to the tests reported in Fig. 5, those results are hard to believe.

Response

Thank you for highlighting the multiple-comparisons issue in Fig. 5. We have now revised both the analysis and the results by including a correction of the statistical threshold due to multiple comparison. For each combination of region and signal (e.g., internal exploration signal in IPS), we extracted the peak signal for the three decision types and ran three one-sample t -tests against zero. These three tests jointly address a single, hypothesis-driven question (whether that region encodes that signal across decisions), so we treat them as one family and apply the Benjamini–Hochberg FDR procedure within that family. The p values reported in the Results and the asterisks in Fig. 5 now reflect these FDR-adjusted p -values. In contrast, Supplementary Fig. S8 is explicitly labelled as showing raw p -values for descriptive illustration of the time courses.

We have now revised the manuscript and supplementary Fig. S8.

Manuscript (Lines 311-318):

...To test whether each brain region signals exploration value or general decision value, next we ran time course analyses **within each trial subset defined by the impending decision (i.e., the internal exploration, external exploration or accept decision to be executed), and the three value regressors (internal exploration, external exploration and accept values) were entered simultaneously. Thus, all effects were reported from the same trials in that decision context. We then extracted the peak signal for each of the three decision types and carried out three one sample t tests against zero. p values reported for Fig. 5 are false discovery rate corrected (FDR-adjusted p; FDR procedure described in the Multiple-comparisons section in Methods).**

Manuscript (Lines 323-326):

... In particular, we found that, in the IPS, the internal exploration signal remained significant in all three types of decision (Fig. 5a, top panel; internal exploration: $\beta = 0.103$, $t_{23} = 2.253$, **FDR-adjusted $p = 0.034$** , external exploration: $\beta = 0.080$, $t_{23} = 2.285$, **FDR-adjusted $p = 0.034$** ; accept: $\beta = 0.075$, $t_{23} = 2.872$, **FDR-adjusted $p = 0.026$**).

Manuscript (Lines 336-341):

... There was a positive external exploration signal when participants made external exploration (Fig. 5b, middle panel; $\beta = 0.039$, $t_{23} = 2.094$, **FDR-adjusted $p = 0.047$**) or accept decisions ($\beta = 0.058$, $t_{23} = 2.102$, **FDR-adjusted $p = 0.047$**). During internal exploration, the effect of external exploration signal was initially non-significant (Supplementary Fig. S8b, the inset in the top panel; $\beta = 0.091$, $t_{23} = 1.555$, **FDR-adjusted $p = 0.134$**).

Manuscript (Lines 348-349):

...Indeed, we found a significant external exploration signal (Fig. 5b, middle panel; $\beta = 0.209$, $t_{23} = 3.135$, **FDR-adjusted $p = 0.014$**).

Manuscript (Lines 358-366):

... **mPFC showed general decision signal, instead of accept signal. Interestingly, during internal exploration, the mPFC showed a significant internal exploration signal (Fig. 5c, top panel; $\beta = 0.166$, $t_{23} = 2.666$, FDR-adjusted $p = 0.014$). Furthermore, during external exploration, the mPFC showed external exploration signal (Fig. 5c, middle panel; $\beta = 0.071$, $t_{23} = 2.872$, FDR-adjusted $p = 0.026$). Based on the results in Figures 2**

and 3, we expected that the mPFC should also pass this test by showing an accept signal in all three types of decision. In the mPFC, we could only find positive accept signals during internal exploration decisions ($\beta = 0.155$, $t_{23} = 2.838$, FDR-adjusted $p = 0.014$), and accept decisions (Fig. 5c, bottom panel; $\beta = 0.179$, $t_{23} = 4.133$, FDR-adjusted $p = 0.001$).

Revised Figure.5:

Figure.5 (Lines 371-373):

... A Benjamini and Hochberg false-discovery-rate procedure (FDR, $q = 0.05$) was applied to the three decision-wise tests within each combination of region and signal, and the stars reflect these corrected p values (FDR-adjusted p).

Figure.5 (Lines 382-384):

... * denotes FDR-adjusted $p < 0.05$, ** denotes FDR-adjusted $p < 0.01$, *** denotes FDR-adjusted $p \leq 0.001$. Error bars represent \pm SEM.

Manuscript (Lines 639-645):

... *Multiple-comparisons correction.* To test whether each region encodes a given signal independently of the impending decision while controlling for multiple comparisons in the tests shown in Fig. 5, we first obtained the peak signal for the internal exploration, external exploration and accept decisions and performed three one-sample t tests against zero (one per decision type). We then controlled the false discovery rate across these three tests within each combination of region and signal using the Benjamini and Hochberg procedure (FDR, $q = 0.05$). In the Results, we refer to these adjusted values as FDR-adjusted p .

Supplementary Figure S8 (Lines 136-137):

... **Supplementary Figure S8. Time courses of the internal exploration, external exploration and accept signals in the IPS, ACC and mPFC during internal exploration, external exploration and accept decisions.** Statistics for this figure are reported with raw p values for descriptive illustration of the time courses.